# PER-ETD: A Polynomially Efficient Emphatic Temporal Difference Learning Method

**Ziwei Guan, Tengyu Xu & Yingbin Liang**
Department of Electrical and Computer Engineering
Ohio State University
Columbus, OH 43210, USA
`{guan.283, xu.3260}@buckeyemail.osu.edu` & `liang.889@osu.edu`

## Abstract

Emphatic temporal difference (ETD) learning (Sutton et al., 2016) is a successful method to conduct the off-policy value function evaluation with function approximation. Although ETD has been shown to converge asymptotically to a desirable value function, it is well-known that ETD often encounters a large variance so that its sample complexity can increase exponentially fast with the number of iterations. In this work, we propose a new ETD method, called PER-ETD (i.e., **PE**riodically **R**estarted-ETD), which restarts and updates the follow-on trace only for a finite period for each iteration of the evaluation parameter. Further, PER-ETD features a design of the logarithmical increase of the restart period with the number of iterations, which guarantees the best trade-off between the variance and bias and keeps both vanishing sublinearly. We show that PER-ETD converges to the same desirable fixed point as ETD, but improves the exponential sample complexity of ETD to be polynomials. Our experiments validate the superior performance of PER-ETD and its advantage over ETD.

## 1 Introduction

As a major value function evaluation method, temporal difference (TD) learning (Sutton, 1988; Dayan, 1992) has been widely used in various planning problems in reinforcement learning. Although TD learning performs successfully in the on-policy settings, where an agent can interact with environments under the target policy, it can perform poorly or even diverge under the off-policy settings when the agent only has access to data sampled by a behavior policy (Baird, 1995; Tsitsiklis & Van Roy, 1997; Mahmood et al., 2015). To address such an issue, the gradient temporal-difference (GTD) (Sutton et al., 2008) and least-squares temporal difference (LSTD) (Yu, 2010) algorithms have been proposed, which have been shown to converge in the off-policy settings. However, since GTD and LSTD consider an objective function based on the behavior policy, which adjusts only the distribution mismatch of the action and does not adjust the distribution mismatch of the state, their converging points can be largely biased from the true value function due to the distribution mismatch between the target and behavior policies, even when the express power of the function approximation class is arbitrarily large (Kolter, 2011).

In order to provide a more accurate evaluation, Sutton et al. (2016) proposed the emphatic temporal difference (ETD) algorithm, which introduces the *follow-on trace* to address the distribution mismatch issue and thus adjusts both state and action distribution mismatch. The stability of ETD was then shown in Sutton et al. (2016); Mahmood et al. (2015), and the asymptotic convergence guarantee for ETD was established in Yu (2015), it has also achieved great success in many tasks (Ghiassian et al., 2016; Ni, 2021). However, although ETD can address the distribution mismatch issue to yield a more accurate evaluation, it often suffers from very large variance error due to the follow-on trace estimation over a long or infinite time horizon (Hallak et al., 2016). Consequently, the convergence of ETD can be unstable. It can be shown that the variance of ETD can grow exponentially fast as the number of iterations grow so that ETD requires **exponentially** large number of samples to converge. Hallak et al. (2016) proposed an ETD method to keep the follow-on trace bounded but at the cost of a possibly large bias error. This thus poses the following intriguing question:

*Can we design a new ETD method, which overcomes its large variance without introducing a large bias error, and improves its exponential sample complexity to be polynomial at the same time?*

In this work, we provide an affirmative answer.

## 1.1 MAIN CONTRIBUTIONS

We propose a novel ETD approach, called PER-ETD (i.e., **PE**riodically **R**estarted-ETD), in which for each update of the value function parameter we restart the follow-on trace iteration and update it only for $b$ times (where we call $b$ as the **period length**). Such a periodic restart effectively reduces the variance of the follow-on trace. More importantly, with the design of the period length $b$ to increase logarithmically with the number of iterations, PER-ETD attains the *polynomial* rather than *exponential* sample complexity required by ETD.

We provide the theoretical guarantee of the sample efficiency of PER-ETD via the finite-time analysis. We show that PER-ETD (both PER-ETD(0) and PER-ETD($\lambda$)) converges to the same fixed points of ETD(0) and ETD($\lambda$), respectively, but with only *polynomial* sample complexity (whereas ETD takes exponential sample complexity). Our analysis features the following key insights. (a) The period length $b$ plays the role of trading off between the variance (of the follow-on trace) and bias error (with respect to the fixed point of ETD), and its optimal choice of logarithmical increase with the number of iterations achieves the best tradeoff and keeps both errors vanishing sublinearly. (b) Our analysis captures how the mismatch between the behavior and target policies affects the convergence rate of PER-ETD. Interestingly, the mismatch level determines a phase-transition phenomenon of PER-ETD: as long as the mismatch is below a certain threshold, then PER-ETD achieves the same convergence rate as the on-policy TD algorithm; and if the mismatch is above the threshold, the converge rate of PER-ETD gradually decays as the level of mismatch increases.

Experimentally, we demonstrate that PER-ETD converges in the case that neither TD nor ETD converges. Further, our experiments provide the following two interesting observations. (a) There does exist a choice of the period length for PER-ETD, which attains the best tradeoff between the variance and bias errors. Below such a choice, the bias error is large so that evaluation is not accurate, and above it the variance error is large so that the convergence is unstable. (b) Under a small period length $b$, it is not always the case that PER-ETD($\lambda$) with $\lambda = 1$ attains the smallest error with respect to the ground truth value function. The best $\lambda$ depends on the geometry of the locations of fixed points of PER-ETD($\lambda$) for $0 \le \lambda \le 1$, which is determined by chosen features.

## 1.2 RELATED WORKS

**TD learning and GTD:** The asymptotic convergence of TD learning was established by Sutton (1988); Jaakkola et al. (1994); Dayan & Sejnowski (1994); Tsitsiklis & Van Roy (1997), and its non-asymptotic convergence rate was further characterized recently in Dalal et al. (2018a); Bhandari et al. (2018); Kotsalis et al. (2020); Chen et al. (2019); Kaledin et al. (2020); Hu & Syed (2019); Srikant & Ying (2019). The gradient temporal-difference (GTD) was proposed in Sutton et al. (2008) for off-policy evaluation and was shown to converge asymptotically. Then, Dalal et al. (2018b); Gupta et al. (2019); Wang et al. (2018); Xu et al. (2019); Xu & Liang (2021) provided the finite-time analysis of GTD and its variants.

**Emphatic Temporal Difference (ETD) Learning:** The ETD approach was originally proposed in the seminal work Sutton et al. (2016), which introduced the follow-on trace to overcome the distribution mismatch between the behavior and target policies. Yu (2015) provided the asymptotic convergence guarantee for ETD. Hallak et al. (2016) showed that the variance of the follow-on trace may be unbounded. They further proposed an ETD method with a variable decay rate to keep the follow-on trace bounded but at the cost of a possibly large bias error. Our approach is different and keeps both the variance and bias vanishing sublinearly with the number of iterations. Imani et al. (2018) developed a new policy gradient theorem, where the emphatic weight is used to correct the distribution shift. Zhang et al. (2020b) provided a new variant of ETD, where the emphatic weights are estimated through function approximation. Van Hasselt et al. (2018); Jiang et al. (2021) studied ETD with deep neural function class.

**Comparison to concurrent work:** During our preparation of this paper, a concurrent work (Zhang & Whiteson, 2021) was posted on arXiv, and proposed a truncated ETD (which we refer to as

T-ETD for short here), which truncates the update of the follow-on trace to reduce the variance of ETD. While T-ETD and our PER-ETD share a similar design idea, there are several critical differences between our work from Zhang & Whiteson (2021). (a) Our PER-ETD features a design of the logarithmical increase of the restart period with the number of iterations, which guarantees the convergence to the original fixed point of ETD, with both the variance and bias errors vanishing sublinearly. However, T-ETD is guaranteed to converge only to a *truncation-length-dependent* fixed point, where the convergence is obtained by treating the truncation length as a constant. A careful review of the convergence proof indicates that the variance term scales exponentially fast with the truncation length, and hence the polynomial efficiency is not guaranteed as the truncation length becomes large. (b) Our convergence rate for PER-ETD does not depend on the cardinality of the state space and has only polynomial dependence on the mismatch parameter of the behavior and target policies. However, the convergence rate in Zhang & Whiteson (2021) scales with the cardinality of the state space, and increases exponentially fast with the mismatch parameter of behavior and target policies. (c) This paper further studies PER-ETD($\lambda$) and the impact of $\lambda$ on the converge rate and variance and bias errors, whereas Zhang & Whiteson (2021) considers further the application of T-ETD to the control problem.

## 2 BACKGROUND AND PRELIMINARIES

### 2.1 MARKOV DECISION PROCESS

We consider the infinite-horizon Markov decision process (MDP) defined by the five tuple $(\mathcal{S}, \mathcal{A}, r, \mathsf{P}, \gamma)$. Here, $\mathcal{S}$ and $\mathcal{A}$ denote the state and action spaces respectively, which are both assumed to be finite sets, $r : \mathcal{S} \times \mathcal{A} \to \mathbb{R}$ denotes the reward function, $\mathsf{P} : \mathcal{S} \times \mathcal{A} \to \Delta(\mathcal{S})$ denotes the transition kernel, where $\Delta(\mathcal{S})$ denotes the probability simplex over the state space $\mathcal{S}$, and $\gamma \in (0, 1)$ is the discount factor.

A policy $\pi : \mathcal{S} \to \Delta(\mathcal{A})$ of an agent maps from the state space to the probability simplex over the action space $\mathcal{A}$, i.e., $\pi(a|s)$ represents the probability of taking the action $a$ under the state $s$. At any time $t$, given that the system is at the state $s_t$, the agent takes an action $a_t$ with the probability $\pi(a_t|s_t)$, and receives a reward $r(s_t, a_t)$. The system then takes a transition to the next state $s_{t+1}$ at time $t + 1$ with the probability $\mathsf{P}(s_{t+1}|s_t, a_t)$.

For a given policy $\pi$, we define the value function corresponding to an initial state $s_0 = s \in \mathcal{S}$ as $V_\pi(s) = \mathbb{E}\left[\sum_{t=0}^{\infty} \gamma^t r(s_t, a_t)|s_0 = s, \pi\right]$. Then the value function over the state space can be expressed as a vector $V_\pi = (V_\pi(1), V_\pi(2), \ldots, V_\pi(|\mathcal{S}|))^\top \in \mathbb{R}^{|\mathcal{S}|}$. Here, $V_\pi$ is a deterministic function of the policy $\pi$. We use capitalized characters to be consistent with the literature.

When the state space is large, we approximate the value function $V_\pi$ via a linear function class as $V_\theta(s) = \phi^\top(s)\theta$, where $\phi(s) \in \mathbb{R}^d$ denotes the feature vector, and $\theta \in \mathbb{R}^d$ denotes the parameter vector to be learned. We further let $\Phi = [\phi(1), \phi(2), \ldots, \phi(|\mathcal{S}|)]^\top$ denote the feature matrix, and then $V_\theta = \Phi\theta$. We assume that the feature matrix $\Phi$ has linearly independent columns and each feature vector has bounded $\ell_2$-norm, i.e., $\|\phi(s)\|_2 \leq B_\phi$ for all $s \in \mathcal{S}$.

### 2.2 TEMPORAL DIFFERENCE (TD) LEARNING FOR ON-POLICY EVALUATION

In order to evaluate the value function for a given target policy $\pi$ (i.e., find the linear function approximation parameter $\theta$), the temporal difference (TD) learning can be employed based on a sampling trajectory, which takes the following update rule at each time $t$:

$$\theta_{t+1} = \theta_t + \eta_t \left( r(s_t, a_t) + \gamma \theta_t^\top \phi(s_{t+1}) - \theta_t^\top \phi(s_t) \right) \phi(s_t), \tag{1}$$

where $\eta_t$ is the stepsize at time $t$. The main idea here is to follow the Bellman operation update to approach its fixed point, and the above sampled version update can be viewed as the so-called semi-gradient descent update. If the trajectory is sampled by the target policy $\pi$, then the above TD algorithm can be shown to converge to the fixed point solution, where the convergence is guaranteed by the negative definiteness of the so-called *key matrix* $A := \lim_{t \to \infty} \mathbb{E}\left[(\gamma\phi(s_t + 1) - \phi(s_t))\phi^\top(s_t)\right].$

## 2.3 Emphatic TD (ETD) Learning for Off-policy Evaluation

Consider the off-policy setting, where the goal is still to evaluate the value function for a given target policy $\pi$, but the agent has access only to trajectories sampled under a behavior policy $\mu$. Namely, at each time $t$, the probability of taking an action $a_t$ given $s_t$ is $\mu(a_t|s_t)$. Let $d_\mu$ denote the stationary distribution of the Markov chain induced by the behavior policy $\mu$, i.e., $d_\mu$ satisfies $d_\mu^\top = d_\mu^\top P_\pi$. We assume that $d_\mu(s) > 0$ for all states. The mismatch between the target and behavior policies can be addressed by incorporating the importance sampling factor $\rho(s,a) := \frac{\pi(a|s)}{\mu(a|s)}$ into eq. (1) to adjust the TD learning update direction. However, with such modification, the key matrix $A$ may not be negative definite so that the algorithm is no longer guaranteed to converge.

In order to address this divergence issue, the emphatic temporal difference (ETD) algorithm has been proposed by Sutton et al. (2016), which takes the following update

$$\theta_{t+1} = \theta_t + \eta_t \rho(s_t, a_t) F_t \left( r(s_t, a_t) + \gamma \theta_t^\top \phi(s_{t+1}) - \theta_t^\top \phi(s_t) \right) \phi(s_t). \tag{2}$$

In eq. (2), in addition to the importance sampling factor $\rho$, a follow-on trace coefficient $F_t$ is introduced as a calibration factor, which is updated as

$$F_t = \gamma \rho(s_{t-1}, a_{t-1}) F_{t-1} + 1, \tag{3}$$

with initialization $F_0 = 1$. With such a follow-on trace factor, the key matrix becomes negative definite, and ETD has been shown to converge asymptotically in Yu (2015) to the fixed point

$$\theta^* = \left( \Phi^\top F(I - \gamma P_\pi) \Phi \right)^{-1} \Phi^\top F r_\pi, \tag{4}$$

where $F = \text{diag}(f(1), f(2), \ldots, f(|\mathcal{S}|))$ and $f(i) = d_\mu(i) \lim_{t \to \infty} \mathbb{E}[F_t | s_t = i]$.

Similarly, the ETD($\lambda$) algorithm can be further derived, which has the following update

$$\theta_{t+1} = \theta_t + \eta_t \rho(s_t, a_t) \left( r(s_t, a_t) + \gamma \theta_t^\top \phi(s_{t+1}) - \theta_t^\top \phi(s_t) \right) e_t,$$

where $e_t$ is updated as $e_t = \gamma \lambda \rho(s_{t-1}, a_{t-1}) e_{t-1} + M_t \phi(s_t)$ and $M_t = \lambda + (1-\lambda) F_t$, where $M_0 = 1$ and $e_0 = \phi(s_0)$. It has been shown that with a diminishing stepsize (Yu, 2015), ETD($\lambda$) converges to the fixed point given by $\theta_\lambda^* = \left( \Phi^\top M(I - \gamma \lambda P_\pi)^{-1}(I - \gamma P_\pi)\Phi\theta \right)^{-1} \Phi^\top M(I - \gamma \lambda P_\pi)^{-1} r_\pi$, where $M = \text{diag}(m(1), m(2), \ldots, m(|\mathcal{S}|))$ and $m(i) = d_\mu(i) \lim_{t \to \infty} \mathbb{E}[M_t | s_t = i]$.

## 2.4 Notations

For the simplicity of expression, we adopt the following shorthand notations. For a fixed integer $b$, let $s_t^\tau := s_{t(b+1)+\tau}, a_t^\tau := a_{t(b+1)+\tau}, \rho_t^\tau = \frac{\pi(a_t^\tau | s_t^\tau)}{\mu(a_t^\tau | s_t^\tau)}$ and $\phi_t^\tau = \phi(s_t^\tau)$. We also define the filtration $\mathcal{F}_t = \sigma \left( s_0, a_0, s_1, a_1, \ldots, s_{t(b+1)+b}, a_{t(b+1)+b}, s_{t(b+1)+b+1} \right)$. Further, let $r_\pi \in \mathbb{R}^{|\mathcal{S}|}$, where $r_\pi(s) = \sum_{a \in \mathcal{A}} r(s, a) \pi(a|s)$. Let $P_\pi \in \mathbb{R}^{|\mathcal{S}| \times |\mathcal{S}|}$, where $P_\pi(s'|s) = \sum_{a \in \mathcal{A}} \pi(a|s) \mathsf{P}(s'|s, a)$. For a matrix $M \in \mathbb{R}^{N \times N}$, $M_{(s, \cdot)}$ denotes its $s$-th row and $M_{(\cdot, s)}$ denotes its $s$-th column. We define $B_\phi := \max_s \|\phi(s)\|_2$ as the upper bound on the feature vectors, and define $\rho_{max} := \max_{s,a} \frac{\pi(a|s)}{\mu(a|s)}$ as the maximum of the distribution mismatch over all state-action pairs.

## 3 Proposed PER-ETD Algorithms

**Drawbacks of ETD:** In the original design of ETD (Sutton et al., 2016) described in Section 2.3, the follow-on trace coefficient $F_t$ is updated throughout the execution of the algorithm. As a result, its variance can increase exponentially with the number of iterations, which causes the algorithm to be unstable and diverge, as observed in Hallak et al. (2016) (also see our experiment in Section 5).

In order to overcome the divergence issue of ETD, we propose to **PE**riodically **R**estart the follow-on trace update for ETD, which we call as the PER-ETD algorithm (see Algorithm 1). At iteration $t$, PER-ETD reinitiates the follow-on trace $F$ and update it for $b$ iterations to obtain an estimate $F_t^b$, where we call $b$ as the period length. The emphatic update operator at $t$ is then given by

$$\widehat{\mathcal{T}}_t(\theta) = F_t^b \rho_t^b \phi_t^b (\phi_t^b - \gamma \phi_t^{b+1})^\top \theta - F_t^b \rho_t^b \phi_t^b r_t^b, \tag{5}$$

---

**Algorithm 1** PER-ETD(0)

---

1: **Input:** Parameters $T$, $b$, and $\eta_t$.
2: Initialize: $\theta_0 = 0$.
3: **for** $t = 0, 1, ..., T$ **do**
4:     $F$ update: $F_t^{\tau+1} = \gamma \rho_t^\tau F_t^\tau + 1$, where $\tau = 0, 1, \ldots, b-1$ and $F_t^0 = 1$;
5:     $\theta$ update: $\theta_{t+1} = \Pi_\Theta \left( \theta_t + \eta_t F_t^b \rho_t^b (r_t^b + \gamma \theta_t^\top \phi_t^{b+1} - \theta_t^\top \phi_t^b) \phi_t^b \right)$
6: **end for**

---

and PER-ETD updates the value function parameter $\theta_t$ as $\theta_{t+1} = \Pi_\Theta \left( \theta_t - \eta_t \widehat{\mathcal{T}}_t(\theta_t) \right)$, where the projection onto an bounded closed convex set $\Theta$ helps to stabilize the algorithm. It can be shown that $\lim_{b \to \infty} \mathbb{E}[\widehat{\mathcal{T}}_t(\theta) | \mathcal{F}_{t-1}] = \mathcal{T}(\theta)$ where $\mathcal{T}(\theta) := \left( \Phi^\top F(I - \gamma P_\pi) \Phi \right) \theta - \Phi^\top F r_\pi$. The fixed point of the operator $\mathcal{T}(\theta)$ is $\theta^*$ defined in eq. (4), which is exactly the fixed point of original ETD.

**Definition 1** (Optimal point and $\epsilon$-accurate convergence). *We call the unique fixed point $\theta^*$ of $\mathcal{T}(\theta)$ as the optimal point (which is the same as the fixed point of ETD). The algorithm attains an $\epsilon$-accurate optimal point if its output $\theta_T$ satisfies $\|\theta_T - \theta^*\|_2^2 \leq \epsilon$.*

The goal of PER-ETD is to find the original optimal point $\theta^*$ of ETD, which is independent from the period length $b$. Our analysis will provide a guidance to choose the period length $b$ in order for PER-ETD to keep both the variance and bias errors below the target $\epsilon$-accuracy with polynomial sample efficiency.

---

**Algorithm 2** PER-ETD($\lambda$)

---

1: **Input:** Parameters $T$, $b$, and $\eta_t$.
2: Initialize: $\theta_0 = 0$.
3: **for** $t = 0, 1, \ldots, T$ **do**
4:     Set $F_t^0 = M_t^0 = 1$ and $e_t^0 = \phi_t^0$
5:     **for** $\tau = 1, \ldots, b$ **do**
6:         $F_t^\tau = \rho_t^{\tau-1} \gamma F_t^{\tau-1} + 1$,    $M_t^\tau = \lambda + (1-\lambda) F_t^\tau$,    $e_t^\tau = \gamma \lambda \rho_t^{\tau-1} e_t^{\tau-1} + M_t^\tau \phi_t^\tau$
7:     **end for**
8:     $\theta$ update: $\theta_{t+1} = \Pi_\Theta \left( \theta_t + \eta_t \rho_t^b \left( r_t^b + \gamma \theta_t^\top \phi_t^{b+1} - \theta_t^\top \phi_t^b \right) e_t^b \right)$
9: **end for**

---

We then extend PER-ETD(0) to PER-ETD($\lambda$) (see Algorithm 2), which incorporates the eligible trace. Specifically, at each iteration $t$, PER-ETD($\lambda$) reinitiates the follow-on trace $F_t$ and updates it together with $M_t$ and the eligible trace $e_t$ for $b$ iterations to obtain an estimate $e_t^b$. Then the emphatic update operator at $t$ is given by

$$\widehat{\mathcal{T}}_t^\lambda(\theta) = \rho_t^b e_t^b \left( \phi_t^b - \gamma \phi_t^{b+1} \right)^\top \theta - \rho_t^b r_t^b e_t^b, \tag{6}$$

and the value function parameter $\theta_t$ is updated as $\theta_{t+1} = \Pi_\Theta \left( \theta_t - \eta_t \widehat{\mathcal{T}}_t^\lambda(\theta_t) \right)$. It can be shown that $\lim_{b \to \infty} \mathbb{E} \left[ \widehat{\mathcal{T}}_t^\lambda(\theta) \Big| \mathcal{F}_{t-1} \right] = \mathcal{T}^\lambda(\theta)$, where $\mathcal{T}^\lambda(\theta) = \Phi^\top M(I - \gamma \lambda P_\pi)^{-1}(I - \gamma P_\pi) \Phi \theta - \Phi^\top M(I - \gamma \lambda P_\pi)^{-1} r_\pi$, which takes a unique fixed point $\theta_\lambda^*$ as the original ETD($\lambda$). The optimal point and the $\epsilon$-accurate convergence can be defined in the same fashion as in Definition 1. It has been shown in Hallak et al. (2016) that $\theta_\lambda^*$ is exactly the orthogonal projection of $V_\pi$ to the function space when $\lambda = 1$, and thus is the optimal approximation to the value function.

## 4   Finite-Time Analysis of PER-ETD Algorithms

### 4.1   Technical Assumptions

We take the following standard assumptions for analyzing the TD-type algorithms in the literature (Jiang et al., 2021; Zhang & Whiteson, 2021; Yu, 2015).

**Assumption 1** (Coverage of behavior policy). *For all $s \in \mathcal{S}$ and $a \in \mathcal{A}$, the behavior policy $\mu$ satisfies $\mu(a|s) > 0$ as long as $\pi(a|s) > 0$.*

**Assumption 2.** *The Markov chain induced by the behavior policy $\mu$ is irreducible and recurrent.*

The following lemma on the geometric ergodicity has been established.

**Lemma 1** (Geometric ergodicity). *(Levin & Peres, 2017, Thm. 4.9) Suppose Assumption 2 holds. Then the Markov chain induced by the behavior policy $\mu$ has a unique stationary distribution $d_\mu$ over the state space $\mathcal{S}$. Moreover, the Markov chain is uniformly geometric ergodic, i.e., there exist constants $C_M \geq 0$ and $0 < \chi < 1$ such that for every initial state $s_0 \in \mathcal{S}$, the state distribution $d_{\mu,t}(s) = \mathbb{P}(s_t = s | s_0)$ after $t$ transitions satisfies $\|d_{\mu,t} - d_\mu\|_1 \leq C_M \chi^t$.*

### 4.2 FINITE-TIME ANALYSIS OF PER-ETD(0)

In PER-ETD(0), the update of the value function parameter is fully determined by the empirical emphatic operator $\widehat{\mathcal{T}}_t(\theta)$ defined in eq. (5). Thus, we first characterize the bias and variance errors of $\widehat{\mathcal{T}}_t(\theta)$, which serve the central role in establishing the convergence rate for PER-ETD(0).

**Proposition 1** (Bias bound). *Suppose Assumptions 1 and 2 hold. Then we have*

$$\mathbb{E}\left[\left\|\mathcal{T}(\theta_t) - \mathbb{E}\left[\widehat{\mathcal{T}}_t(\theta_t)\Big|\mathcal{F}_{t-1}\right]\right\|_2\right] \leq C_b \left(B_\phi \|\theta_t - \theta^*\|_2 + \epsilon_{approx}\right)\xi^b,$$

*where $\epsilon_{approx} = \|\Phi\theta^* - V_\pi\|_\infty$ is the approximation error of the fixed point, $\xi = \max\{\gamma, \chi\} < 1$, $B_\phi = \max_s \|\phi(s)\|_2$, and $C_b > 0$ is a constant whose exact form can be found in the proof.*

Proposition 1 characterizes the conditional expectation of the bias error of the empirical emphatic operator $\widehat{\mathcal{T}}_t(\theta)$. Since $\xi = \max\{\gamma, \chi\} < 1$, such a bias error decays exponentially fast as $b$ increases.

**Proposition 2** (Variance bound). *Suppose Assumptions 1 and 2 hold. Then we have*

$$\mathbb{E}\left[\left\|\widehat{\mathcal{T}}_t(\theta_t)\right\|_2^2\Big|\mathcal{F}_{t-1}\right] \leq \sigma^2, \quad \text{where} \quad \sigma^2 = \begin{cases} \mathcal{O}(1), & \text{if} \quad \gamma^2\rho_{max} < 1, \\ \mathcal{O}(b), & \text{if} \quad \gamma^2\rho_{max} = 1, \\ \mathcal{O}\left((\gamma^2\rho_{max})^b\right), & \text{if} \quad \gamma^2\rho_{max} > 1, \end{cases} \quad (7)$$

*where $\mathcal{O}(\cdot)$ is with respect to the scaling of $b$, and $\rho_{max} = \max_{s,a} \frac{\pi(a|s)}{\mu(a|s)}$.*

Proposition 2 captures the variance bound of the empirical emphatic operator. It can be seen that if the distribution mismatch is large (i.e., $\gamma^2\rho_{max} > 1$), the variance bound grows exponentially large as $b$ increases, which is consistent with the finding in Hallak et al. (2016). However, as we show below, as long as $b$ is controlled to grow only *logarithmically* with the number of iterations, such a variance error will decay sublinearly with the number of iterations. At the same time, the bias error can also be controlled to decay sublinearly, so that the overall convergence of PER-ETD can be guaranteed with *polynomial* sample complexity efficiency.

**Theorem 1.** *Suppose Assumptions 1 and 2 hold. Consider PER-ETD(0) specified in Algorithm 1. Let the stepsize $\eta_t = \mathcal{O}\left(\frac{1}{t}\right)$ and suppose the period length $b$ and the projection set $\Theta$ are properly chosen (see Appendix D.3 for the precise conditions). Then the output $\theta_T$ of PER-ETD(0) falls into the following two cases.*

*(a) If $\gamma^2\rho_{max} \leq 1$, then $\mathbb{E}\left[\|\theta_T - \theta^*\|_2^2\right] \leq \tilde{\mathcal{O}}\left(\frac{1}{T}\right)$.*

*(b) If $\gamma^2\rho_{max} > 1$, then $\mathbb{E}\left[\|\theta_T - \theta^*\|_2^2\right] \leq \mathcal{O}\left(\frac{1}{T^a}\right)$, where $a = 1/(\log_{1/\xi}(\gamma^2\rho_{max}) + 1) < 1$.*

*Thus, PER-ETD(0) attains an $\epsilon$-accurate solution with $\tilde{\mathcal{O}}\left(\frac{1}{\epsilon}\right)$ samples if $\gamma^2\rho_{max} \leq 1$, and with $\tilde{\mathcal{O}}\left(\frac{1}{\epsilon^{1/a}}\right)$ samples if $\gamma^2\rho_{max} > 1$.*

Theorem 1 captures how the convergence rate depends on the mismatch between the behavior and target policies via the parameter $\rho_{max}$ (where $\rho_{max} \geq 1$). (a) If $\gamma^2\rho_{max} \leq 1$, i.e., the mismatch is less than a threshold, then PER-ETD(0) converges at the rate of $\tilde{\mathcal{O}}\left(\frac{1}{T}\right)$, which is the same as that of on-policy TD learning (Bhandari et al., 2018). This result indicates that even under a mild mismatch $1 < \rho_{max} \leq 1/\gamma^2$, PER-ETD achieves the same convergence rate as on-policy TD learning. (b) If $\gamma^2\rho_{max} \geq 1$, i.e., the mismatch is above the threshold, then PER-ETD(0) converges at a slower

rate of $\tilde{\mathcal{O}}\left(\frac{1}{T^a}\right)$ because $a < 1$. Further, as the mismatch parameter $\rho_{max}$ gets larger, the converge becomes slower, because $a$ becomes smaller.

**Bias and variance tradeoff:** Theorem 1 also indicates that although PER-ETD(0) updates the follow-on trace only over a finite period length $b$, it still converges to the optimal fixed point $\theta^*$. This benefits from the proper choice of the period length, which achieves the best bias and variance tradeoff as we explain as follows. The proof of Theorem 1 shows that the output $\theta_T$ of PER-ETD(0) satisfies the following convergence rate:

$$\mathbb{E}\left[\|\theta_T - \theta^*\|_2^2\right] \leq \mathcal{O}\left(\frac{\|\theta_0 - \theta^*\|_2^2}{T^2}\right) + \underbrace{\mathcal{O}\left(\frac{\sigma^2}{T}\right)}_{\text{variance}} + \underbrace{\mathcal{O}\left(\frac{\xi^{2b}}{T}\right) + \mathcal{O}\left(\xi^b\right)}_{\text{bias}}. \tag{8}$$

If $\gamma^2 \rho_{max} \leq 1$, then $\sigma^2$ in the variance term in eq. (8) satisfies $\sigma^2 \leq \mathcal{O}(b)$ as given in eq. (7), which increases at most linearly fast with $b$. Then we set $b = \mathcal{O}\left(\frac{\log T}{\log(1/\xi)}\right)$ so that both the variance and the bias terms in eq. (8) achieve the same order of $\mathcal{O}\left(\frac{1}{T}\right)$, which dominates the overall convergence.

If $\gamma^2 \rho_{max} > 1$, then $\sigma^2$ in the variance term in eq. (8) satisfies $\sigma^2 = \mathcal{O}\left((\gamma^2 \rho_{max})^b\right)$ as given in eq. (7), Now, we need to set $b$ as $b = \mathcal{O}\left(\frac{\log(T)}{\log(\gamma^2 \rho_{max}) + \log(1/\xi)}\right)$, where the increase with $\log T$ has a smaller coefficient than the previous case, so that both the variance and the bias terms in eq. (8) achieve the same order of $\mathcal{O}\left(\frac{1}{T^a}\right)$. Such a choice of $b$ balances the exponentially increasing variance and exponentially decaying bias to achieve the same rate.

### 4.3 FINITE-TIME ANALYSIS OF PER-ETD($\lambda$)

In PER-ETD($\lambda$), the update of the value function parameter is determined by the empirical emphatic operator $\widehat{\mathcal{T}}_t^\lambda(\theta)$ defined in eq. (6). Thus, we first obtain the bias and variance errors of $\widehat{\mathcal{T}}_t^\lambda(\theta)$, which facilitate the analysis of the convergence rate for PER-ETD($\lambda$).

**Proposition 3.** *Suppose Assumptions 1 and 2 hold. Then we have*

$$\left\|\mathbb{E}\left[\widehat{\mathcal{T}}_t^\lambda(\theta_t)\Big|\mathcal{F}_{t-1}\right] - \mathcal{T}^\lambda(\theta_t)\right\|_2 \leq C_{b,\lambda}\left(B_\phi\|\theta_t - \theta_\lambda^*\|_2 + \epsilon_{approx}\right)\xi^b,$$

*where $\epsilon_{approx} = \|\Phi\theta_\lambda^* - V_\pi\|_\infty$ is the approximation error of the fixed point, $\xi = \max\{\chi, \gamma\} < 1$, $B_\phi = \max_s \|\phi(s)\|_2$, and $C_{b,\lambda}$ is a constant given a fixed $\lambda$ whose exact form can be found in proof.*

The above proposition shows that the bias error of the empirical emphatic operator $\widehat{\mathcal{T}}_t^\lambda(\theta)$ in PER-ETD($\lambda$) decays exponentially fast as $b$ increases, because $\xi = \max\{\gamma, \chi\} < 1$.

**Proposition 4.** *Suppose Assumptions 1 and 2 hold. Then we have $\mathbb{E}\left[\left\|\widehat{\mathcal{T}}_t^\lambda(\theta_t)\right\|_2^2\Big|\mathcal{F}_{t-1}\right] \leq \sigma_\lambda^2$, where $\sigma_\lambda^2 = \mathcal{O}\left(\rho_{max}^b\right)$.*

Compared with Proposition 2 of PER-ETD(0), Proposition 4 indicates that ETD($\lambda$) has a larger variance, which always increases exponentially with $b$ when $\rho_{max} > 1$. This is due to the fact that the eligible trace $e_t^b$ carries the historical information and is less stable than $\phi_t^b$.

**Theorem 2.** *Suppose Assumptions 1 and 2 hold. Consider PER-ETD($\lambda$) specified in Algorithm 2. Let the stepsize $\eta_t = \mathcal{O}\left(\frac{1}{t}\right)$ and suppose the period length $b$ and the projection set $\Theta$ are properly chosen (see Appendix E.3 for the precise conditions). Then the output $\theta_T$ of PER-ETD($\lambda$) satisfies $\mathbb{E}\left[\|\theta_T - \theta_\lambda^*\|_2^2\right] \leq \mathcal{O}\left(\frac{1}{T^{a_\lambda}}\right)$, where $a_\lambda = \frac{1}{\log_{1/\xi}(\rho_{max})+1}$. PER-ETD($\lambda$) attains an $\epsilon$-accurate solution with $\tilde{\mathcal{O}}\left(\frac{1}{\epsilon^{1/a_\lambda}}\right)$ samples.*

Theorem 2 indicates that PER-ETD($\lambda$) converges to the optimal fixed point $\theta_\lambda^*$ determined by the infinite-length update of the follow-on trace. Furthermore, PER-ETD($\lambda$) converges at the rate of $\tilde{\mathcal{O}}\left(\frac{1}{T^{a_\lambda}}\right)$ which is slower than PER-ETD(0) (as $a_\lambda < a$) due to the larger variance of PER-ETD($\lambda$).

**Bias and variance tradeoff:** We next explain how the period length $b$ achieves the best tradeoff between the bias and variance errors and thus yields polynomial sample efficiency. The proof of

Theorem 2 shows that the output $\theta_T$ of PER-ETD($\lambda$) satisfies the following convergence rate:

$$\mathbb{E}\left[\|\theta_T - \theta_\lambda^*\|_2^2\right] \leq \mathcal{O}\left(\frac{\|\theta_0 - \theta_\lambda^*\|_2^2}{T^2}\right) + \underbrace{\mathcal{O}\left(\frac{\sigma_\lambda^2}{T}\right)}_{\text{variance}} + \underbrace{\mathcal{O}\left(\frac{\xi^{2b}}{T}\right) + \mathcal{O}\left(\xi^b\right)}_{\text{bias}}. \tag{9}$$

In eq. (9), $\sigma_\lambda^2$ in the variance term takes the form $\sigma_\lambda^2 = \mathcal{O}\left(\rho_{max}^b\right)$ as given in Proposition 4. We need to set $b = \mathcal{O}\left(\frac{\log(T)}{\log(\rho_{max}) + \log(1/\xi)}\right)$ so that both the variance and the bias terms in eq. (9) achieve the same order of $\mathcal{O}\left(\frac{1}{T^{a_\lambda}}\right)$. Thus, such a choice of $b$ balances the exponentially increasing variance and exponentially decaying bias to achieve the same rate.

**Impact of the eligible trace (via the parameter $\lambda$) on error bound:** It has been shown that with the aid of eligible trace, both TD and ETD achieve smaller error bounds (Sutton & Barto, 2018; Hallak et al., 2016). However, this is not always the case for PER-ETD. Since PER-ETD applies a finite period length $b$, the fixed point of PER-ETD(1) is generally not the same as the projection of the ground truth to the function approximation space. Thus, as $\lambda$ changes from 0 to 1, depending on the geometrical locations of the fixed points of PER-ETD($\lambda$) for all $\lambda$ (determined by chosen features) with respect to the ground truth projection, any value $0 \leq \lambda \leq 1$ may achieve the smallest bias error. We illustrate this further by experiments in Section 5.2.

## 5 EXPERIMENTS

### 5.1 PERFORMANCE OF PER-ETD(0)

We consider the BAIRD counter-example. The details of the MDP setting and behavior and target policies could be found in Appendix A.1. We adopt a constant learning rate for both PER-ETD(0) and PER-ETD($\lambda$) and all experiments take an average over 20 random initialization. We set the stepsize $\eta = 2^{-9}$ for all algorithms for fair comparison. For PER-ETD(0), we adopt one-dimensional features $\Phi_1 = (0.35, 0.35, 0.35, 0.35, 0.35, 0.35, 0.37)^\top$. The ground truth value function $V_\pi = (10, 10, 10, 10, 10, 10, 10)^\top$ and does not lie inside the linear function class.

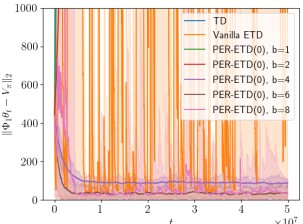 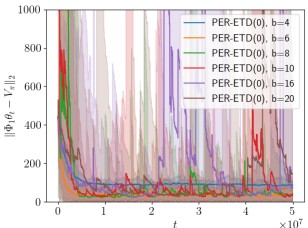

(a) Comparison of TD, ETD, PER-ETD(0)  (b) Tradeoff between bias and variance by $b$

Figure 1: Performance of PER-ETD(0) and comparison

In Figure 1(a), we compare the performance of of TD, vanilla ETD(0) and PER-ETD(0) with $b = 2, 4, 8$ in terms of the distance between the ground truth and the learned value functions. It can be observed that our proposed PER-ETD(0) converges close to the ground truth at a properly chosen period length such as $b = 4$ and $b = 8$, whereas TD diverges due to no treatment on off-policy data historically, and ETD (0) also diverges due to the very large variance.

In Figure 1(b), we plot how the bias and the variance of PER-ETD(0) change as the period length $b$ changes. Clearly, small $b$ (e.g., $b = 4$) yields a small variance but a large bias. Then as $b$ increases from 4 to 6, bias is substantially reduced. As $b$ continues to increase from 8 to 20, there is a significant increase in variance. This demonstrates a clear tradeoff between the bias and variance as we capture in our theory.

### 5.2 PERFORMANCE OF PER-ETD($\lambda$)

We next focus on PER-ETD($\lambda$) under the same experiment setting as in Section 5.1 and study how $\lambda$ affects the performance. We conduct our experiments under three features $\Phi_1$, $\Phi_2$, and $\Phi_3$ specified

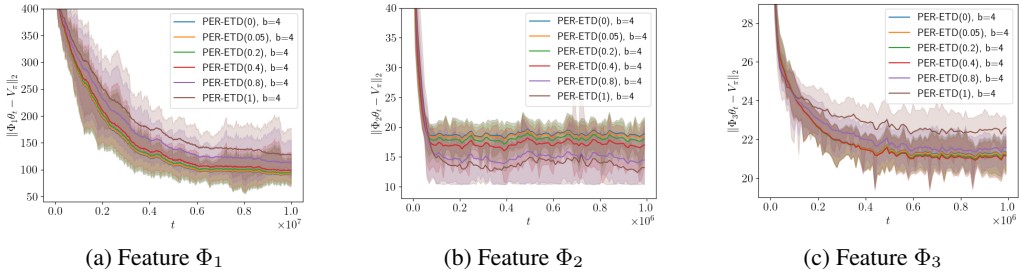

Figure 2: Performance of PER-ETD($\lambda$) and dependence on features

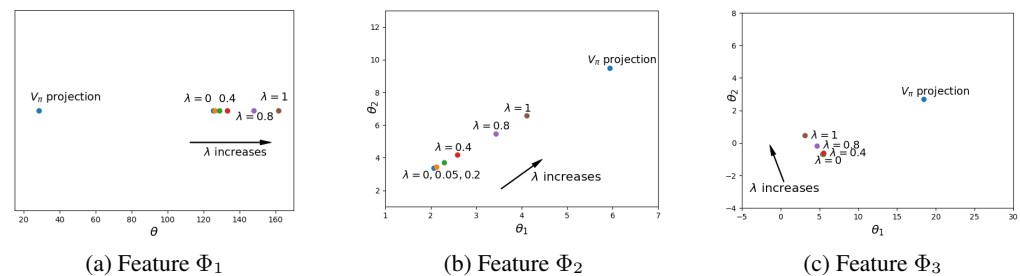

Figure 3: Fixed points of PER-ETD($\lambda$) and project of the value function. (a): $\theta$ lies in 1-dimensional Euclidean space $\mathbb{R}^1$ along horizontal direction; (b), (c): $\theta$ lies in 2-dimensional Euclidean space $\mathbb{R}^2$.

in Appendix A.2. Figure 2 shows how the bias error with respect to the ground truth changes as $\lambda$ increases under the three chosen features. As shown in Figure 2 (a), (b), and (c), $\lambda = 0, 1$, and some value between 0 and 1 respectively achieve the smallest error under the corresponding feature. This is in contrast to the general understanding that $\lambda = 1$ typically achieves the smallest error. In fact, each case can be explained by the plot in Figure 3 under the same feature. Each plot in Figure 3 illustrates how the fixed points of PER-ETD($\lambda$) are located with respect to the ground truth projection (as $V_\pi$ projection) for $b = 4$. Since the period length $b$ is finite, the fixed point of PER-ETD(1) is not located at the same point as the ground truth projection. The geometric locations of the fixed points of PER-ETD($\lambda$) for $0 \le \lambda \le 1$ are determined by chosen features. The bias error corresponds to the distance between the fixed point of PER-ETD($\lambda$) and the $V_\pi$ projection. Then under each feature, the value of $\lambda$ that attains the smallest error with respect to the $V_\pi$ projection can be readily seen from the plot in Figure 3. For example, under the feature $\Phi_3$, Figure 3 (c) suggests that neither $\lambda = 0$ nor $\lambda = 1$, but some $\lambda$ between 0 and 1 achieves the smallest error. This explains the result in Figure 2 (c) that $\lambda = 0.4$ achieves the smallest error among other curves.

As a summary, our experiment suggests that the best $\lambda$, under which PER-ETD($\lambda$) attains the smallest error, depends on the geometry of the problem determined by chosen features. In practice, if PER-ETD($\lambda$) is used as a critic in policy optimization problems, $\lambda$ may be tuned via the final reward achieved by the algorithm.

## 6 CONCLUSION

In this paper, we proposed a novel PER-ETD algorithm, which uses a periodic restart technique to control the variance of follow-on trace update. Our analysis shows that by selecting the period length properly, both bias and variance of PER-ETD vanishes sublinearly with the number of iterations, leading to the polynomial sample efficiency to the desired unique fixed point of ETD, whereas ETD requires exponential sample complexity. Our experiments verified the advantage of PER-ETD against both TD and ETD. Moreover, our experiments of PER-ETD($\lambda$) illustrated that under the finite period length in practice, the best $\lambda$ that achieves the smallest bias error is feature dependent. We anticipate that PER-ETD can be applied to various off-policy optimal control algorithms such as actor-critic algorithms and multi-agent reinforcement learning algorithms.

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

# SUPPLEMENTARY MATERIALS

## A    SPECIFICATION OF EXPERIMENTS IN SECTION 5

### A.1    EXPERIMENT SETTINGS

The BAIRD counter-example is illustrated in Figure 4, which has 7 states and 2 actions. If the first action (illustrated as dashed lines) is taken, then the environment transitions from the current state to states 1 to 6 following the uniform distribution and returns a reward 0; and if the second action (illustrated as solid lines) is taken, the environment transitions from the current state to state 7 with probability 1 and returns a reward 1. We choose the target policy as $\pi(0|s) = 0.1$ and $\pi(1|s) = 0.9$ for all states; and choose the behavior policy as $\mu(0|s) = 6/7$ and $\mu(1|s) = 1/7$ for all states. Moreover, we specify the discount factor $\gamma = 0.99$.

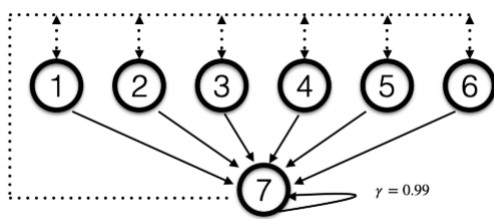

Figure 4: BAIRD example (Sutton & Barto, 2018)

### A.2    FEATURES FOR EXPERIMENTS IN SECTION 5.2

In the experiment in Section 5.2, we choose the following features:

$$
\begin{aligned}
\Phi_1 =& (0.35, 0.35, 0.35, 0.35, 0.35, 0.35, 0.37)^\top; \\
\Phi_2 =& ((0.3425, 0.0171)^\top, (0.1902, 0.4248)^\top, (0.1354, 0.76)^\top, (0.1357, 0.7973)^\top, \\
& (0.8674, 0.8774)^\top, (0.5166, 0.9493)^\top, (0.3094, 0.8535)^\top)^\top; \\
\Phi_3 =& ((0.5162, 0.9013)^\top, (0.5128, 0.5999)^\top, (0.289, 0.4649)^\top, (0.3399, 0.5334)^\top, \\
& (0.315, 0.2278)^\top, (0.667, 0.461)^\top, (0.3706, 0.1457)^\top)^\top.
\end{aligned}
$$

### A.3    COMPUTATION OF THE FIXED POINT OF PER-ETD($\lambda$)

In this section, we provide the steps to compute the fixed point of PER-ETD($\lambda$) in Figure 3. We first define the matrix $A$ and $c$ as follows

$$
\begin{aligned}
A &:= \lim_{t \to \infty} \mathbb{E}\left[ A_t \left( := \left( \rho_t^b e_t^b (\phi_t^b - \gamma \phi_t^{b+1})^\top \right) \right) \right], \\
c &:= \lim_{t \to \infty} \mathbb{E}\left[ c_t \left( := \rho_t^b r_t^b e_t^b \right) \right].
\end{aligned}
$$

It can be shown that, the fixed point of PER-ETD($\lambda$) algorithm is $\theta^* = A^{-1}c$.

We next show how to derive the formulation of the matrix $A$ and vector $c$. As we will show later in eqs. (51), (53) and (55), we have

$$
A = \lim_{t \to \infty} \mathbb{E}\left[ A_t | \mathcal{F}_{t-1} \right] = \bar{\beta}_b (I - \gamma P_\pi)\Phi, \tag{10}
$$

$$
c = \lim_{t \to \infty} \mathbb{E}\left[ c_t | \mathcal{F}_{t-1} \right] = \bar{\beta}_b r_\pi, \tag{11}
$$

where $\bar{\beta}_b := \lim_{t \to \infty} \mathbb{E}\left[ \beta_b \right]$ and

$$
\beta_b(s) = \lambda \Phi^\top D_{\mu,b} + (1-\lambda)\Phi^\top F_b + \gamma\lambda\beta_{b-1}P_\pi, \tag{12}
$$

where $D_{\mu,\tau} = \mathrm{diag}(d_{\mu,\tau})$, $d_{\mu,\tau}(s) = \mathbb{P}(s_t^\tau = s | \mathcal{F}_{t-1})$, $F_b = \mathrm{diag}(f_b)$, and $f_b$ is determined iteratively by eq. (25) as follows

$$f_b = d_{\mu,b} + \gamma P_\pi^\top f_{b-1}, \quad \text{with} \quad f_0 = d_{\mu,0}. \tag{13}$$

Taking expectation on both sides of eqs. (12) and (13) with respect to $\mathcal{F}_{t-1}$ and letting $t \to \infty$ yield

$$\bar{f}_b = d_\mu + \gamma P_\pi^\top \bar{f}_{b-1}, \quad \text{with} \quad \bar{f}_0 = d_\mu, \tag{14}$$

$$\bar{\beta}_b = \lambda \Phi^\top D_\mu + (1-\lambda)\Phi^\top \bar{F}_b + \gamma\lambda\bar{\beta}_{b-1}P_\pi, \quad \text{with} \quad \bar{\beta}_0 = \Phi^\top D_\mu \tag{15}$$

where $\bar{f}_b := \lim_{t\to\infty} \mathbb{E}[f_b]$ and $\bar{F}_b = \mathrm{diag}(\bar{f}_b)$. The explicit formulation of $\bar{\beta}_b$ can be derived by applying eqs. (14) and (15) iteratively. We can then obtain $A$ and $c$ by substituting the obtained formulation of $\bar{\beta}_b$ into eq. (10) and eq. (11), respectively.

### A.4 Replotted Figures 1 and 2 with Variance Bars

In this subsection, we replotted Figures 1 and 2 with variance bars (rather than error bands) in Figures 5 and 6, respectively.

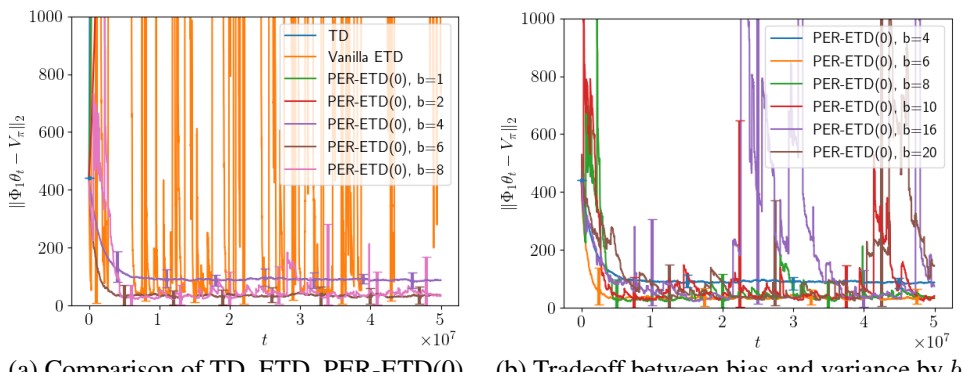

(a) Comparison of TD, ETD, PER-ETD(0)  (b) Tradeoff between bias and variance by $b$

Figure 5: Performance of PER-ETD(0) and comparison

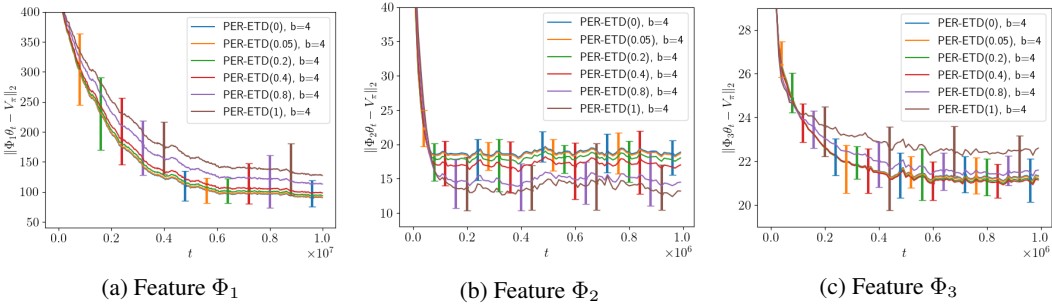

(a) Feature $\Phi_1$  (b) Feature $\Phi_2$  (c) Feature $\Phi_3$

Figure 6: Performance of PER-ETD($\lambda$) and dependence on features

## B More Experiments

In this section, we conduct further experiments to answer the following two intriguing questions:

- If the distribution mismatch parameter $\rho_{max}$ changes, how will different approaches perform and compare with each other?
- Focusing on our algorithm PER-ETD, how do the choices of behavior policy and target policy affect its convergence?

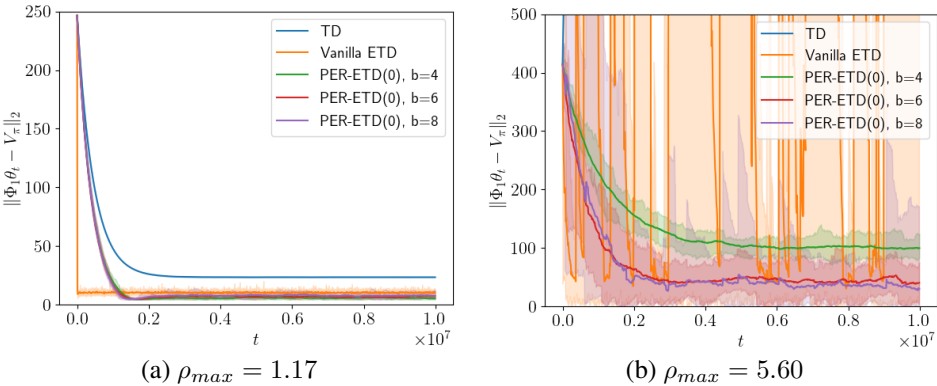

(a) $\rho_{max} = 1.17$           (b) $\rho_{max} = 5.60$

Figure 7: Comparisons of TD, ETD, PER-ETD(0) with different target policies

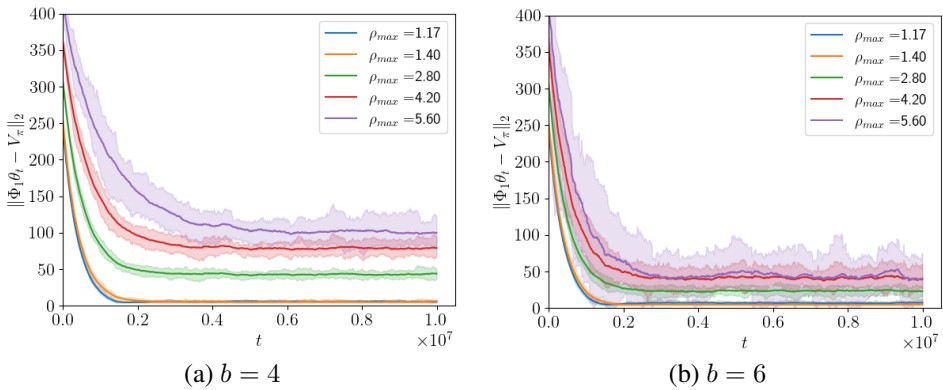

(a) $b = 4$           (b) $b = 6$

Figure 8: Performance of PER-ETD(0) under different target policies (marked by their different resulting distribution mismatch parameter $\rho_{max}$). The behavior policy is kept the same.

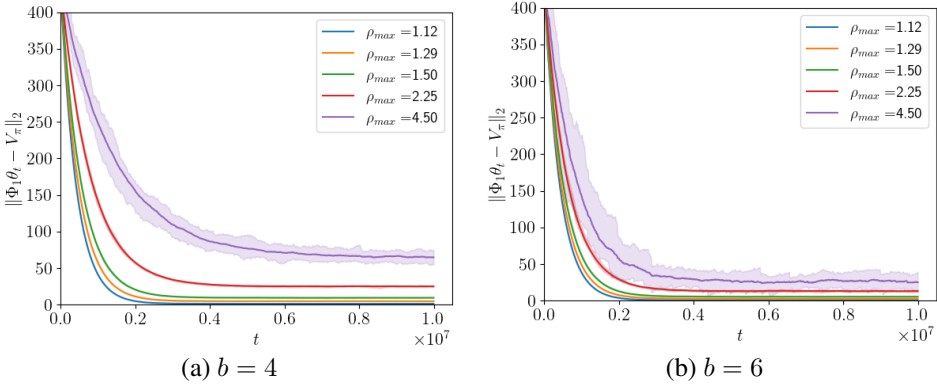

(a) $b = 4$           (b) $b = 6$

Figure 9: Performance of PER-ETD(0) under different behavior policies (marked by their different resulting distribution mismatch parameter $\rho_{max}$). The target policy is kept the same.

We focus on the MDP environment in Appendix A.1. In our experiments, the performance of every algorithm is averaged over 20 random initializations and the error band in our plots captures the actual variation of the performance during these experimental runs (which can be viewed as the variance of the algorithms).

In Figure 7, we consider two settings with the distribution mismatch parameter $\rho_{max} = 1.17$ and 5.60, respectively, and compare the performance of three off-policy algorithms: TD, vanilla ETD and our PER-ETD. More specifically, we choose the behavior policy as $\mu(1|s) = \frac{6}{7}$ and $\mu(2|s) = \frac{1}{7}$

for all states, and choose two target polices, whose probabilities to take the second action are $0.167$ and $0.8$, respectively, on all states. (Then their probabilities to take the first action are determined automatically through $\pi(1|s) + \pi(2|s) = 1$). Therefore, the maximum distribution mismatch $\rho_{max}$ for the these two target policies are $1.17$ and $5.60$, respectively. Figure 7 (a) shows that under only slightly mismatch (i.e., $\rho_{max} = 1.17$), TD suffers from a large convergence error (i.e., the error with respect to the ground truth value function at the point of convergence) and converges slowly. Vanilla ETD converges with the fastest rate and achieves a smaller convergence error than TD, but suffers from a relatively large variance. Our PER-ETD achieves a better tradeoff between the convergence rate and the variance (faster rate than TD, almost the same convergence error as ETD but with smaller variance). Figure 7 (b) shows that under a large distribution mismatch (i.e., $\rho_{max} = 5.6$), TD does not converge, and vanilla ETD experiences a substantially large variance. However, our PER-ETD still convergences fast as long as the period length $b$ is chosen properly, e.g., $b = 4, 6, 8$. Further note that PER-ETD has a smaller convergence error as the period length $b$ increases, but the variance gets larger; which are consistent with our theorem.

In Figure 8, we focus on our PER-ETD, and study how different target policies affect the performance. We choose the same behavior policy as the above experiment, i.e., $\mu(1|s) = \frac{6}{7}$ and $\mu(2|s) = \frac{1}{7}$ for all states. We choose 5 target polices, whose probabilities to take the second action are $0.167, 0.2, 0.4, 0.6, 0.8$ for all states, respectively. These different target policies affect the performance via their resulting distribution mismatch $\rho_{max} = 1.17, 1.40, 2.80, 4.20, 5.60$, respectively. For both $b = 4$ and $b = 6$, Figure 8 indicates that larger mismatch causes slower convergence rate, larger convergence error and larger variance, which agrees with our theorem.

In Figure 9, we also focus on our PER-ETD, and study how different behavior policies affect the performance. We pick the target policy to be $\pi(1|s) = 0.1$ and $\pi(2|s) = 0.9$ for all states, and 5 different behavior policies with $\mu(2|s) = 0.2, 0.4, 0.6, 0.7$ and $0.8$ for all state, respectively. These different behavior policies affect the performance via their different resulting distribution mismatch $\rho_{max} = 1.12, 1.29, 1.5, 2.25, 4.5$, respectively. Figure 9 clearly demonstrates that larger distribution mismatch results in slower convergence rate, larger convergence error and larger variance, which is in the same nature as changing the target policy shown in Figure 8 and is consistent with our theorem.

## C    SUPPORTING LEMMAS

The following lemma is well-known. We include it for the convenience of our proof.

**Lemma 2.** *Consider a transition matrix $P \in \mathbb{R}^{N \times N}$, where $\sum_j P_{(i,j)} = 1$ for all $i$ and $0 \leq P_{(i,j)} \leq 1$ for all $j$. We have for any $n \in \mathbb{N}$, $\|P^n\|_\infty = 1$, and $\|(P^\top)^n\|_1 = 1$.*

**Lemma 3.** *Consider $0 < p, q < 1$, with $p \neq q$. We have $\sum_{k=0}^{n-1} p^k q^{n-k} \leq \frac{1}{|p-q|} \xi^n$, where $\xi = \max\{p, q\}$.*

*Proof of Lemma 3.* If $p > q$, we have

$$\sum_{k=0}^{n-1} p^k q^{n-k} \leq \sum_{k=0}^{n-1} p^{k+1} q^{n-1-k} \leq \sum_{m=0}^{n-1} p^{n-m} q^m.$$

Without loss of generality, we assume $p < q$ and $\xi = q$. We have

$$\sum_{k=0}^{n-1} p^k q^{n-k} = q^n \cdot \sum_{k=0}^{n-1} (\tfrac{p}{q})^k = q^n \cdot \frac{1 - (p/q)^n}{1 - p/q} \leq \frac{1}{q-p} \cdot q^n = \frac{1}{|p-q|} \cdot \xi^n.$$

$\square$

**Lemma 4.** *Consider a matrix $P \in \mathbb{R}^{N \times N}$ where $\sum_j P_{(i,j)} \leq C$ for all $i$ and $0 \leq P_{(i,j)} \leq 1$ for all $j$, and a positive vector $x$ where $x \in \mathbb{R}^N$ and $x_i \geq 0$ for all $i$. We have*

$$\mathbf{1}^\top P x \leq C \mathbf{1}^\top x.$$

*Proof of Lemma 4.* We have

$$\mathbf{1}^\top P x = \sum_{1 \le i,j \le N} P_{i,j} x_i = \sum_{1 \le i \le N} x_i \left( \sum_{1 \le j \le N} P_{i,j} \right) \le C \sum_{1 \le i \le N} x_i = C \mathbf{1}^\top x.$$

$\square$

**Lemma 5.** *Consider a diagonal matrix $D \in \mathbb{R}^{|\mathcal{S}| \times |\mathcal{S}|}$, where $D = \mathrm{diag}(d(1), d(2), \ldots, d(|\mathcal{S}|))$. a transition matrix $P \in \mathbb{R}^{|\mathcal{S}| \times |\mathcal{S}|}$ where $\sum_j P_{i,j} = 1$ for all $i = 1, 2, \ldots, |\mathcal{S}|$ and $P_{i,j} \ge 0$ for all $j$, and an arbitrary vector $x \in \mathbb{R}^{|\mathcal{S}|}$. We have*

$$\|\Phi^\top D P x\|_2 \le B_\phi \|d\|_1 \|x\|_\infty.$$

*Proof of Lemma 5.* We have

$$\Phi^\top D = (d(1)\phi(1), d(2)\phi(2), \ldots, d(|\mathcal{S}|)\phi(|\mathcal{S}|)),$$

and

$$\left( \Phi^\top D P \right)_{(\cdot, s)} = \sum_{\tilde{s}} P_{\tilde{s},s} d(\tilde{s}) \phi(\tilde{s}),$$

which implies

$$\Phi^\top D P x = \sum_s x(s) \left( \sum_{\tilde{s}} P_{\tilde{s},s} d(\tilde{s}) \phi(\tilde{s}) \right) = \sum_{\tilde{s}} \left( \sum_s x(s) P_{\tilde{s},s} \right) d(\tilde{s}) \phi(\tilde{s}).$$

Taking $\ell_2$ norm on both sides of the above equality yields

$$
\begin{aligned}
\left\| \Phi^\top D P x \right\|_2 &= \left\| \sum_{\tilde{s}} \left( \sum_s x(s) P_{\tilde{s},s} \right) d(\tilde{s}) \phi(\tilde{s}) \right\|_2 \\
&\le \sum_{\tilde{s}} \left| \sum_s x(s) P_{\tilde{s},s} \right| \cdot |d(\tilde{s})| \cdot \|\phi(\tilde{s})\|_2 \\
&= B_\phi \|d\|_1 \cdot \max_{\tilde{s}} \left| \sum_s x(s) P_{\tilde{s},s} \right| \\
&\le B_\phi \|d\|_1 \|x\|_\infty \max_{\tilde{s}} \left| \sum_s P_{\tilde{s},s} \right| \\
&= B_\phi \|d\|_1 \|x\|_\infty.
\end{aligned}
$$

$\square$

**Lemma 6.** *Consider a diagonal matrix $D \in \mathbb{R}^{|\mathcal{S}| \times |\mathcal{S}|}$ where $D = \mathrm{diag}(d(1), d(2), \ldots, d(|\mathcal{S}|))$, a transition matrix $P \in \mathbb{R}^{|\mathcal{S}| \times |\mathcal{S}|}$ where $\sum_j P_{i,j} = 1$ for all $i = 1, 2, \ldots, |\mathcal{S}|$ and $P_{i,j} \ge 0$ for all $j$, a matrix $Q \in \mathbb{R}^{|\mathcal{S}| \times |\mathcal{S}|}$ that satisfies $0 \le Q_{i,j} \le C P_{i,j}$, where $C > 1$ is a constant, and an arbitrary vector $x \in \mathbb{R}^{|\mathcal{S}|}$. We have*

$$\mathrm{trace}\left( Q^m P^n \Phi \Phi^\top D \right) \le C^m B_\phi^2 \|d\|_1,$$

*and,*

$$\mathrm{trace}\left( P^n Q^m \Phi \Phi^\top D \right) \le C^m B_\phi^2 \|d\|_1,$$

*for any $m$ and $n \in \mathbb{N}_{\ge 0}$.*

*Proof of Lemma 6.* For any given $\tau \ge m \ge 0$ and $n \ge 0$,

$$\left| (Q^m P^n \Phi \Phi^\top)_{(i,j)} \right| = \left| \langle (Q^m)_{(i,\cdot)}, (P^n \Phi \Phi^\top)_{(\cdot,j)} \rangle \right| \overset{(i)}{\le} \|(Q^m)_{(i,\cdot)}\|_1 \|(P^n \Phi \Phi^\top)_{(\cdot,j)}\|_\infty, \quad (16)$$

where $(i)$ follows from the Hölder's inequality.

Furthermore, for the term of $(P^n \Phi \Phi^\top)_{(\cdot, j)}$, we have,

$$\|(P^n \Phi \Phi^\top)_{(\cdot, j)}\|_\infty \leq \|P^n\|_\infty \|(\Phi \Phi^\top)_{(\cdot, j)}\|_\infty \overset{(i)}{=} \|(\Phi \Phi^\top)_{(\cdot, j)}\|_\infty,$$

where $(i)$ follows from Lemma 2.

Moreover, for the $i$th entry of $(\Phi \Phi^\top)_{(\cdot, j)}$, we have

$$\left|(\Phi \Phi^\top)_{(i,j)}\right| = \left|\phi(i)^\top \phi(j)\right| \leq \|\phi(i)\|_2 \|\phi(j)\|_2 \leq B_\phi^2$$

The above uniform bounds over all $i$ imply that $\|(\Phi \Phi^\top)_{(\cdot, j)}\|_\infty \leq B_\phi^2$. Hence,

$$\|(P^n \Phi \Phi^\top)_{(\cdot, j)}\|_\infty \leq B_\phi^2.$$

Substituting the above inequality back into eq. (16), we obtain

$$\left|(Q^m P^n \Phi \Phi^\top)_{(i,j)}\right| \leq B_\phi^2 \|(Q^m)_{(i,\cdot)}\|_1 \overset{(i)}{\leq} C^m \|(P^{\tau-m})_{(i,\cdot)}\|_1 B_\phi^2$$

$$\leq C^m B_\phi^2 \sum_j P^{\tau-m}(j|i) = C^m B_\phi^2, \tag{17}$$

where $(i)$ follows by the condition of $Q$, $Q_{(i,j)} \leq C P_{(i,j)}$ for all $i, j$.

Finally, we have

$$\text{trace}\left(Q^m P^n \Phi \Phi^\top D\right) = \sum_i d(i)(Q^m P^n \Phi \Phi^\top)_{(i,i)} \leq \sum_i |d(i)| \left|(Q^m P^n \Phi \Phi^\top)_{(i,i)}\right|$$

$$\leq \|d\|_1 \max_i \left|(Q^m P^n \Phi \Phi^\top)_{(i,i)}\right| \overset{(i)}{\leq} C^m B_\phi^2 \|d\|_1, \tag{18}$$

where $(i)$ follows from eq. (17).

Following steps similar to those in eqs. (16) to (18), we can obtain

$$\text{trace}\left(P^n Q^m \Phi \Phi^\top D\right) \leq C^m B_\phi^2 \|d\|_1.$$

$\square$

**Lemma 7.** *The operators $\mathcal{T}(\theta)$ and $\mathcal{T}^\lambda(\theta)$ satisfy the generalized monotone variational inequality. There exist $\mu_0, \mu_\lambda > 0$, s.t., $\langle \mathcal{T}(\theta), \theta - \theta^* \rangle \geq \mu_0 \|\theta - \theta^*\|_2^2$, and $\langle \mathcal{T}^\lambda(\theta), \theta - \theta^* \rangle \geq \mu_\lambda \|\theta - \theta^*\|_2^2$.*

*Proof of Lemma 7.* We have

$$\langle \mathcal{T}(\theta), \theta - \theta^* \rangle \overset{(i)}{=} \left\langle \left(\Phi^\top F(I - \gamma P_\pi)\Phi\right)(\theta - \theta^*), \theta - \theta^* \right\rangle$$

$$= (\theta - \theta^*)^\top \left(\Phi^\top F(I - \gamma P_\pi)\Phi\right)(\theta - \theta^*)$$

$$\geq \lambda_{min}\left(\Phi^\top F(I - \gamma P_\pi)\Phi\right) \|\theta - \theta^*\|_2^2, \tag{19}$$

where $(i)$ follows from the definition of the $\mathcal{T}$ and $\theta^*$.

Recall that $F(I - \gamma P_\pi)$ is positive definite(Sutton et al., 2016; Mahmood et al., 2015) and $\Phi$ has linearly independent columns. For any $x \in \mathbb{R}^d$ with $x \neq 0$, we have $\Phi x \neq 0$ and

$$x^\top \Phi^\top F(I - \gamma P_\pi)\Phi x = (\Phi x)^\top F(I - \gamma P_\pi)(\Phi x) > 0.$$

The above inequality shows that $\Phi^\top F(I - \gamma P_\pi)\Phi$ is positive definite and thus, there exists $\mu_0 > 0$ such that $\mu_0 = \lambda_{min}\left(\Phi^\top F(I - \gamma P_\pi)\Phi\right)$.

Following steps similar to those in eq. (19) and applying the positive definiteness of $M(I - \gamma \lambda P_\pi)^{-1}(I - \gamma P_\pi)$ (Sutton et al., 2016; Mahmood et al., 2015) yield

$$\left\langle \mathcal{T}^\lambda(\theta), \theta - \theta^* \right\rangle \geq \mu_\lambda \|\theta - \theta^*\|_2^2.$$

$\square$

**Lemma 8.** *The operators $\mathcal{T}(\theta)$ and $\mathcal{T}^\lambda(\theta)$ satisfy the Lipschitz condition. There exist $L_0, L_\lambda > 0$, such that, $\|\mathcal{T}(\theta_1) - \mathcal{T}(\theta_2)\|_2 \le L_0 \|\theta_1 - \theta_2\|_2$, and $\left\|\mathcal{T}^\lambda(\theta_1) - \mathcal{T}^\lambda(\theta_2)\right\|_2 \le L_\lambda \|\theta_1 - \theta_2\|_2$.*

*Proof of lemma 8.* We have

$$
\begin{aligned}
\|\mathcal{T}(\theta_1) - \mathcal{T}(\theta_2)\|_2 &= \left\| \left( \Phi^\top F(I - \gamma P_\pi) \Phi \right) (\theta_1 - \theta_2) \right\|_2 \\
&\le \left\| \Phi^\top F(I - \gamma P_\pi) \Phi \right\|_2 \|\theta_1 - \theta_2\|_2.
\end{aligned} \tag{20}
$$

Let $L_0 := \left\| \Phi^\top F(I - \gamma P_\pi) \Phi \right\|_2$, eq. (20) completes the proof of the first inequality in the Lemma. Let $L_\lambda := \|\Phi^\top M(I - \gamma \lambda P_\pi)^{-1}(I - \gamma P_\pi)\Phi\|_2$, the steps similar to those in eq. (20) finalizes the proof of the second inequality in the Lemma. $\square$

**Lemma 9** (Three point lemma). *Suppose $\Theta$ is a closed and bounded subset of $\mathbb{R}^d$, and $\theta^*$ is the solution of the following maximization problem, $\max_{\theta \in \Theta} \eta \langle G, \theta \rangle + \frac{1}{2}\|\theta - \theta_0\|_2^2$, where $G \in \mathbb{R}^d$ is a vector. Then, we have, for any $\theta \in \Theta$,*

$$
\eta \langle G, \theta^* - \theta \rangle + \frac{1}{2}\|\theta_0 - \theta^*\|_2^2 \le \frac{1}{2}\|\theta_0 - \theta\|_2^2 - \frac{1}{2}\|\theta^* - \theta\|_2^2.
$$

*Proof.* The proof can be found in Lan (2020). $\square$

## D PROOFS OF PROPOSITIONS AND THEOREM FOR PER-ETD(0)

### D.1 PROOF OF PROPOSITION 1

First, by the definition of $\widehat{\mathcal{T}}_t(\theta_t)$, we have

$$
\mathbb{E}\left[ \widehat{\mathcal{T}}_t(\theta_t) \middle| \mathcal{F}_{t-1} \right]
$$

$$
\overset{(i)}{=} \sum_{s \in \mathcal{S}} \sum_{a \in \mathcal{A}} \sum_{s' \in \mathcal{S}}
$$

$$
\cdot \mathbb{P}\left( s_t^b = s, a_t^b = a, s_t^{b+1} = s' \middle| \mathcal{F}_{t-1} \right) \mathbb{E}\left[ \widehat{\mathcal{T}}_t(\theta_t) \middle| \mathcal{F}_{t-1}, s_t^b = s, a_t^b = a, s_t^{b+1} = s' \right]
$$

$$
\overset{(ii)}{=} \sum_{s \in \mathcal{S}} \sum_{a \in \mathcal{A}} \sum_{s' \in \mathcal{S}} \mathbb{P}\left( s_t^b = s \middle| \mathcal{F}_{t-1} \right) \mu(a|s) \mathsf{P}(s'|s, a)
$$

$$
\cdot \mathbb{E}\left[ \rho_t^b F_t^b \phi(s)[\phi^\top(s)\theta_t - r(s,a) - \gamma\phi^\top(s')\theta_t] \middle| \mathcal{F}_{t-1}, s_t^b = s, a_t^b = a, s_t^{b+1} = s' \right]
$$

$$
\overset{(iii)}{=} \sum_{s \in \mathcal{S}} \sum_{a \in \mathcal{A}} \sum_{s' \in \mathcal{S}} \mathbb{P}\left( s_t^b = s \middle| \mathcal{F}_{t-1} \right) \pi(a|s) \mathsf{P}(s'|s,a)\phi(s)[\phi^\top(s)\theta_t - r(s,a) - \gamma\phi^\top(s')\theta_t]
$$

$$
\cdot \mathbb{E}\left[ F_t^b \middle| \mathcal{F}_{t-1}, s_t^b = s \right]
$$

$$
= \sum_{s \in \mathcal{S}} \mathbb{P}\left( s_t^b = s \middle| \mathcal{F}_{t-1} \right) \mathbb{E}\left[ F_t^b \middle| \mathcal{F}_{t-1}, s_t^b = s \right] \phi(s) \left[ \phi^\top(s)\theta_t - \gamma[P_\pi\Phi]_{(s,\cdot)}\theta_t - r_\pi(s) \right], \tag{21}
$$

where $(i)$ follows from the law of total probability, $(ii)$ follows from rewriting $\mathbb{P}\left( s_t^b = s, a_t^b = a, s_t^{b+1} = s' \middle| \mathcal{F}_{t-1} \right)$, and $(iii)$ follows from the facts that $F_t^b$ only depends on $(s_0, a_0, s_1, a_1, \ldots, s_{t(b+1)+b-1}, a_{t(b+1)+b-1})$ and the chain is Markov.

Recall the definition of $\mathcal{T}(\theta_t)$, we have

$$
\begin{aligned}
\mathcal{T}(\theta_t) &= \Phi^\top F\left[ (I - \gamma P_\pi)\Phi\theta_t - r_\pi \right] \\
&= \sum_{s \in \mathcal{S}} f(s) \left( \phi^\top(s)\theta_t - \gamma[P_\pi\Phi]_{(s,\cdot)}\theta_t - r_\pi(s) \right) \phi(s).
\end{aligned} \tag{22}
$$

Equations (21) and (22) together imply the following,

$$
\mathcal{T}(\theta_t) - \mathbb{E}\left[ \widehat{\mathcal{T}}_t(\theta_t) \middle| \mathcal{F}_{t-1} \right] = \sum_{s \in \mathcal{S}} \left( f(s) - \mathbb{P}\left( s_t^b = s \middle| \mathcal{F}_{t-1} \right) \mathbb{E}\left[ F_t^b \middle| \mathcal{F}_{t-1}, s_t^b = s \right] \right)
$$

$$\cdot \left( \phi^\top(s)\theta_t - \gamma \left[P_\pi \Phi\right]_{(s,\cdot)} \theta_t - r_\pi(s) \right) \phi(s)$$

$$\stackrel{(i)}{=} \sum_{s \in \mathcal{S}} (f(s) - f_b(s)) \left( \phi^\top(s)\theta_t - \gamma \left[P_\pi \Phi\right]_{(s,\cdot)} \theta_t - r_\pi(s) \right) \phi(s)$$

$$= \sum_{s \in \mathcal{S}} (f(s) - f_b(s)) \left( (I - \gamma P_\pi) \Phi\theta_t - r_\pi \right)_s \phi(s),$$

where in $(i)$ We define $f_b(s) := \mathbb{P}\left(s_t^b = s | \mathcal{F}_{t-1}\right) \mathbb{E}\left[F_t^b | \mathcal{F}_{t-1}, s_t^b = s\right]$. Taking $\ell_2$ norm on both sides of the above equality yields

$$\left\| \mathcal{T}(\theta_t) - \mathbb{E}\left[ \widehat{\mathcal{T}}_t(\theta_t) \middle| \mathcal{F}_{t-1} \right] \right\|_2$$

$$\leq \left\| \sum_{s \in \mathcal{S}} (f(s) - f_b(s)) \left( (I - \gamma P_\pi) \Phi\theta_t - r_\pi \right)_s \phi(s) \right\|_2$$

$$\leq \sum_{s \in \mathcal{S}} |f(s) - f_b(s)| \cdot |((I - \gamma P_\pi)\Phi\theta_t - r_\pi)_s| \cdot \|\phi(s)\|_2$$

$$\leq \max_{s \in \mathcal{S}} \{\|\phi(s)\|_2\} \max_{s \in \mathcal{S}} \{|((I - \gamma P_\pi)\Phi\theta_t - r_\pi)_s|\} \sum_{s \in \mathcal{S}} |f(s) - f_b(s)|$$

$$= B_\phi \|(I - \gamma P_\pi)\Phi\theta_t - r_\pi\|_\infty \|f - f_b\|_1. \tag{23}$$

We next proceed to bound $\|f - f_b\|_1$. Consider $f_b(s)$, we have

$$f_b(s) = \mathbb{P}\left(s_t^b = s | \mathcal{F}_{t-1}\right) \mathbb{E}\left[F_t^b | s_t^b = s, \mathcal{F}_{t-1}\right]$$

$$\stackrel{(i)}{=} \mathbb{P}\left(s_t^b = s | \mathcal{F}_{t-1}\right)$$
$$\sum_{\tilde{s} \in \mathcal{S}, \tilde{a} \in \mathcal{A}} \mathbb{P}\left(s_t^{b-1} = \tilde{s}, a_t^{b-1} = \tilde{a} | s_t^b = s, \mathcal{F}_{t-1}\right)$$
$$\cdot \mathbb{E}\left[\gamma \rho_t^{b-1} F_t^{b-1} + 1 | \mathcal{F}_{t-1}, s_t^b = s, s_t^{b-1} = \tilde{s}, a_t^{b-1} = \tilde{a}\right]$$

$$\stackrel{(ii)}{=} \mathbb{P}\left(s_t^b = s | \mathcal{F}_{t-1}\right)$$
$$\cdot \left( 1 + \sum_{\tilde{s} \in \mathcal{S}, \tilde{a} \in \mathcal{A}} \frac{\mathbb{P}\left(s_t^{b-1} = \tilde{s} | \mathcal{F}_{t-1}\right) \mu(\tilde{a}|\tilde{s}) \mathsf{P}(s|\tilde{s}, \tilde{a})}{\mathbb{P}\left(s_t^b = s | \mathcal{F}_{t-1}\right)} \mathbb{E}\left[ \gamma \frac{\pi(\tilde{a}|\tilde{s})}{\mu(\tilde{a}|\tilde{s})} F_t^{b-1} \middle| \mathcal{F}_{t-1}, s_t^{b-1} = \tilde{s} \right] \right)$$

$$= \mathbb{P}\left(s_t^b = s | \mathcal{F}_{t-1}\right) + \gamma \sum_{\tilde{s} \in \mathcal{S}} \mathbb{P}(s_t^{b-1} = \tilde{s} | \mathcal{F}_{t-1}) P_\pi(s|\tilde{s}) \mathbb{E}\left[F_t^{b-1} | \mathcal{F}_{t-1}, s_t^{b-1} = s\right],$$

where $(i)$ follows from the law of total probability and $(ii)$ follows from the Bayes rule and the facts that $F_t^{b-1}$ only depends on the chain elements $(s_t^0, a_t^0, s_t^1, \ldots, s_t^{b-1}, a_t^{b-1})$ and the Markov property.

Define $d_{\mu,b}(s) = \mathbb{P}\left(s_t^b = s | \mathcal{F}_{t-1}\right)$, the above equality can be rewritten as

$$f_b(s) = d_{\mu,b}(s) + \gamma \sum_{\tilde{s}} P_\pi(s|\tilde{s}) f_{b-1}(\tilde{s}). \tag{24}$$

Since eq. (24) holds for all $s \in \mathcal{S}$, we have

$$f_b = d_{\mu,b} + \gamma P_\pi^\top f_{b-1}. \tag{25}$$

Note that for $f$ we have the following holds (Sutton et al., 2016; Zhang et al., 2020a)

$$f = d_\mu + \gamma P_\pi^\top f. \tag{26}$$

Equations (25) and (26) imply

$$f - f_b = d_\mu - d_{\mu,b} + \gamma P_\pi^\top (f - f_{b-1}).$$

Applying the above equality recursively yields

$$f - f_b = \sum_{\tau=0}^{b-1} (\gamma P_\pi^\top)^\tau (d_\mu - d_{\mu,b-\tau}) + \gamma^b (P_\pi^\top)^b (f - f_0).$$

Take $\ell_1$ norm on both sides of the above equality, we have

$$
\begin{aligned}
\|f - f_b\|_1 &= \left\| \sum_{\tau=0}^{b-1} \gamma^\tau (P_\pi^\top)^\tau (d_\mu - d_{\mu,b-\tau}) + \gamma^b (P_\pi^\top)^b (f - f_0) \right\| \\
&\leq \sum_{\tau=0}^{b-1} \gamma^\tau \left\| (P_\pi^\top)^\tau (d_\mu - d_{\mu,b-\tau}) \right\|_1 + \gamma^b \left\| (P_\pi^\top)^b (f - f_0) \right\|_1 \\
&\leq \sum_{\tau=0}^{b-1} \gamma^\tau \left\| (P_\pi^\top)^\tau \right\|_1 \|d_\mu - d_{\mu,b-\tau}\|_1 + \gamma^b \left\| (P_\pi^\top)^b \right\| \|f - f_0\|_1 \\
&\overset{(i)}{\leq} \sum_{\tau=0}^{b-1} \gamma^\tau \|d_\mu - d_{\mu,b-\tau}\|_1 + \gamma^b \|f - f_0\|_1 \\
&\overset{(ii)}{\leq} \sum_{\tau=0}^{b-1} C_M \gamma^\tau \chi^{b-\tau} + \gamma^b \|f - f_0\|_1 \\
&\overset{(ii)}{\leq} \frac{1}{|\chi - \gamma|} \cdot C_M \xi^b + \gamma^b (1 + \|f\|_1),
\end{aligned}
\tag{27}
$$

where $(i)$ follows from Lemma 2, $(ii)$ follows from Lemma 1, and $(iii)$ follows from Lemma 3 and defining $\xi := \max\{\chi, \gamma\}$.

To bound the term $\|(I - \gamma P_\pi)\Phi\theta_t - r_\pi\|_\infty$, we proceed as following

$$
\begin{aligned}
\|(I - \gamma P_\pi)\Phi\theta_t - r_\pi\|_\infty &\overset{(i)}{=} \|(I - \gamma P_\pi)(\Phi\theta_t - V_\pi)\|_\infty \\
&= \|(I - \gamma P_\pi)(\Phi\theta_t - \Phi\theta^* + \Phi\theta^* - V_\pi)\|_\infty \\
&\leq \|I - \gamma P_\pi\|_\infty (\|\Phi\theta_t - \Phi\theta^*\|_\infty + \|\Phi\theta^* - V_\pi\|_\infty) \\
&\overset{(ii)}{\leq} (1 + \gamma) B_\phi \|\theta_t - \theta^*\|_2 + (1 + \gamma) \epsilon_{approx},
\end{aligned}
\tag{28}
$$

where $(i)$ follows from the fact $V_\pi = (I - \gamma P_\pi)^{-1} r_\pi$ and $(ii)$ follows from the facts that $\|I - \gamma P_\pi\|_\infty = \max_i \{1 - (P_\pi)_{(i,i)} + \gamma \sum_{j \neq i} (P_\pi)_{(i,j)}\} \leq 1 + \gamma$, $\epsilon_{approx} := \|\Phi\theta^* - V_\pi\|_\infty$, and

$$
\|\Phi\theta_t - \Phi\theta^*\|_\infty = \max_{s \in \mathcal{S}} \phi^\top(s)(\theta_t - \theta^*) \leq B_\phi \|\theta_t - \theta^*\|_2.
$$

Substituting eqs. (27) and (28) into eq. (23) yields

$$
\begin{aligned}
\left\| \mathcal{T}(\theta_t) - \mathbb{E}\left[ \widehat{\mathcal{T}}_t(\theta_t) \middle| \mathcal{F}_{t-1} \right] \right\|_2 &\leq \left( B_\phi^2 \|\theta_t - \theta^*\|_2 + B_\phi \epsilon_{approx} \right) (1 + \gamma) \left( \frac{C_M \xi^b}{|\chi - \gamma|} + \gamma^b (1 + \|f\|_1) \right) \\
&\leq C_b \left( B_\phi \|\theta_t - \theta^*\|_2 + \epsilon_{approx} \right) \xi^b,
\end{aligned}
$$

where $\xi = \max\{\chi, \gamma\}$ and $C_b = B_\phi (1 + \gamma) \left( \frac{C_M}{|\chi - \gamma|} + (1 + \|f\|_1) \right)$.

## D.2 PROOF OF PROPOSITION 2

According to the definition of $\widehat{\mathcal{T}}_t(\theta_t)$, we have

$$
\begin{aligned}
&\mathbb{E}\left[ \left\| \widehat{\mathcal{T}}_t(\theta_t) \right\|^2 \middle| \mathcal{F}_{t-1} \right] \\
&= \mathbb{E}\left[ \left( \rho_t^b F_t^b \right)^2 \left( \theta_t^\top \phi_t^b - r_t^b - \gamma \theta_t^\top \phi_t^{b+1} \right)^2 \|\phi_t^b\|_2^2 \middle| \mathcal{F}_{t-1} \right] \\
&\overset{(i)}{=} \sum_{s \in \mathcal{S}} \sum_{a \in \mathcal{A}} \sum_{s' \in \mathcal{S}} \mathbb{P}\left( s_t^b = s, a_t^b = a, s_t^{b+1} = s' \middle| \mathcal{F}_{t-1} \right) \cdot \left( \theta_t^\top \phi(s) - r(s,a) - \gamma \theta_t^\top \phi(s') \right)^2 \\
&\qquad \cdot \|\phi(s)\|_2^2 \cdot \mathbb{E}\left[ (\rho_t^b F_t^b)^2 \middle| \mathcal{F}_{t-1}, s_t^b = s, a_t^b = a, s_t^{b+1} = s' \right] \\
&\overset{(ii)}{=} \sum_{s \in \mathcal{S}} \sum_{a \in \mathcal{A}} \sum_{s' \in \mathcal{S}} \mathbb{P}\left( s_t^b = s \middle| \mathcal{F}_{t-1} \right) \mu(a|s) P(s'|s,a) \cdot \left( \theta_t^\top \phi(s) - r(s,a) - \gamma \theta_t^\top \phi(s') \right)^2
\end{aligned}
$$

$$\cdot \|\phi(s)\|_2^2 \cdot \frac{\pi^2(a|s)}{\mu^2(a|s)} \mathbb{E}\left[(F_t^b)^2 \big| \mathcal{F}_{t-1}, s_t^b = s, a_t^b = a, s_t^{b+1} = s'\right]$$

$$= \sum_{s \in \mathcal{S}} \mathbb{P}\left(s_t^b = s \big| \mathcal{F}_{t-1}\right) \mathbb{E}\left[(F_t^b)^2 \big| \mathcal{F}_{t-1}, s_t^b = s\right] \|\phi(s)\|_2^2$$

$$\cdot \sum_{a \in \mathcal{A}} \sum_{s' \in \mathcal{S}} \frac{\pi^2(a|s)}{\mu(a|s)} \mathsf{P}(s'|s,a)(\phi^\top(s)\theta_t - r(s,a) - \gamma\phi^\top(s')\theta_t)^2, \tag{29}$$

where $(i)$ follows from the law of total probability and $(ii)$ follows from the fact that $F_t^b$ is independent from previous states and actions given $s_t^b$.

Note that $\|\phi(s)\|_2 \leq B_\phi$ for all $s \in \mathcal{S}$, $r(s,a) \leq r_{max}$ for all $(s,a) \in \mathcal{S} \times \mathcal{A}$, and $\|\theta_t\|_2 \leq B_\theta$ for all $t$ due to projection. We have

$$\begin{aligned}
(\phi^\top(s)\theta_t - r(s,a) - \gamma\phi^\top(s')\theta_t)^2 &\leq 2[(\phi(s) - \gamma\phi(s'))^\top \theta_t]^2 + 2r^2(s,a) \\
&\leq 2\|\phi(s) - \gamma\phi(s')\|_2^2 \|\theta_t\|_2^2 + 2r_{max}^2 \\
&\leq 4(\|\phi(s)\|_2^2 + \gamma^2\|\phi(s')\|_2^2)\|\theta_t\|_2^2 + 2r_{max}^2 \\
&\leq 4(1+\gamma^2)B_\phi^2 B_\theta^2 + 2r_{max}^2.
\end{aligned} \tag{30}$$

We also have

$$\sum_{a \in \mathcal{A}} \sum_{s' \in \mathcal{S}} \frac{\pi^2(a|s)}{\mu(a|s)} \mathsf{P}(s'|s,a) \leq \rho_{max} \cdot \sum_{a \in \mathcal{A}} \sum_{s' \in \mathcal{S}} \pi(a|s)\mathsf{P}(s'|s,a) = \rho_{max}.$$

Substituting the above two inequalities into eq. (29) yields

$$\mathbb{E}\left[\left\|\widehat{\mathcal{T}}_t(\theta_t)\right\|^2 \bigg| \mathcal{F}_{t-1}\right]$$

$$\leq \rho_{max}\left(4(1+\gamma^2)B_\phi^2 B_\theta^2 + 2r_{max}^2\right) B_\phi^2 \sum_{s \in \mathcal{S}} \mathbb{P}\left(s_t^b = s \big| \mathcal{F}_{t-1}\right) \mathbb{E}\left[(F_t^b)^2 \big| \mathcal{F}_{t-1}, s_t^b = s\right]. \tag{31}$$

Define $r_b(s) := \mathbb{P}\left(s_t^b = s \big| \mathcal{F}_{t-1}\right) \mathbb{E}\left[(F_t^b)^2 \big| \mathcal{F}_{t-1}, s_t^b = s\right] = d_{\mu,b}(s)\mathbb{E}\left[(F_t^b)^2 \big| \mathcal{F}_{t-1}, s_t^b = s\right]$.

We have the following equations hold for $r_b(s)$:

$$\begin{aligned}
r_b(s) &= \mathbb{P}(s_t^b = s | \mathcal{F}_{t-1})\mathbb{E}\left[\left(\gamma\rho_t^{b-1}F_t^{b-1} + 1\right)^2 \Big| \mathcal{F}_{t-1}, s_t^b = s\right] \\
&= d_{\mu,b}(s)\mathbb{E}\left[1 + 2\gamma\rho_t^{b-1}F_t^{b-1} + \gamma^2(\rho_t^{b-1})^2(F_t^{b-1})^2 \big| \mathcal{F}_{t-1}, s_t^b = s\right] \\
&= d_{\mu,b}(s) + 2\gamma d_{\mu,b}(s)\mathbb{E}\left[\rho_t^{b-1}F_t^{b-1} \big| \mathcal{F}_{t-1}, s_t^b = s\right] \\
&\quad + \gamma^2 d_{\mu,b}(s)\mathbb{E}\left[(\rho_t^{b-1})^2(F_t^{b-1})^2 \big| \mathcal{F}_{t-1}, s_t^b = s\right].
\end{aligned} \tag{32}$$

For the second term in the RHS of eq. (32), we have

$$d_{\mu,b}(s)\mathbb{E}\left[\rho_t^{b-1}F_t^{b-1} \big| \mathcal{F}_{t-1}, s_t^b = s\right]$$

$$\overset{(i)}{=} d_{\mu,b}(s) \sum_{\tilde{s} \in \mathcal{S}, \tilde{a} \in \mathcal{A}} \mathbb{P}\left(s_t^{b-1} = \tilde{s}, a_t^{b-1} = \tilde{a} \big| s_t^b = s, \mathcal{F}_{t-1}\right)$$

$$\cdot \mathbb{E}\left[\rho_t^{b-1}F_t^{b-1} \big| \mathcal{F}_{t-1}, s_t^b = s, s_t^{b-1} = \tilde{s}, a_t^{b-1} = \tilde{a}\right]$$

$$\overset{(ii)}{=} d_{\mu,b}(s) \sum_{\tilde{s}, \tilde{a}} \frac{d_{\mu,b-1}(\tilde{s})\mu(\tilde{a}|\tilde{s})\mathsf{P}(s|\tilde{s}, \tilde{a})}{d_{\mu,b}(s)} \cdot \mathbb{E}\left[\rho_t^{b-1}F_t^{b-1} \big| \mathcal{F}_{t-1}, s_t^b = s, s_t^{b-1} = \tilde{s}, a_t^{b-1} = \tilde{a}\right]$$

$$\overset{(iii)}{=} \sum_{\tilde{s} \in \mathcal{S}, \tilde{a} \in \mathcal{A}} d_{\mu,b-1}(\tilde{s})\mu(\tilde{a}|\tilde{s})\mathsf{P}(s|\tilde{s}, \tilde{a}) \cdot \frac{\pi(\tilde{a}|\tilde{s})}{\mu(\tilde{a}|\tilde{s})}\mathbb{E}\left[F_t^{b-1} \big| \mathcal{F}_{t-1}, s_t^{b-1} = \tilde{s}\right]$$

$$= \sum_{\tilde{s} \in \mathcal{S}} P_\pi(s|\tilde{s}) \cdot d_{\mu,b-1}(\tilde{s})\mathbb{E}\left[F_t^{b-1} \big| \mathcal{F}_{t-1}, s_t^{b-1} = \tilde{s}\right]$$

$$\overset{(iv)}{=} (P_\pi^\top f_{b-1})_s, \tag{33}$$

where $(i)$ follows from the law of total probability, $(ii)$ follows from the Bayes rule, $(iii)$ follows from Markov property and $(iv)$ follow from the definition of $f_b$ which is given above eq. (23).

For the third term on the RHS of eq. (32), we have

$$
\begin{aligned}
& d_{\mu,b}(s)\mathbb{E}\left[(\rho_t^{b-1}F_b^{b-1})^2\big|\mathcal{F}_{t-1},s_t^b=s\right] \\
& \stackrel{(i)}{=} d_{\mu,b}(s)\sum_{\tilde{s}\in\mathcal{S},\tilde{a}\in\mathcal{A}}\mathbb{P}\left(s_t^{b-1}=\tilde{s},a_t^{b-1}=\tilde{a}\big|s_t^b=s,\mathcal{F}_{t-1}\right) \\
& \qquad\qquad\cdot\mathbb{E}\left[(\rho_t^{b-1}F_b^{b-1})^2\big|\mathcal{F}_{t-1},s_t^b=s,s_t^{b-1}=\tilde{s},a_t^{b-1}=\tilde{a}\right] \\
& \stackrel{(ii)}{=} d_{\mu,b}(s)\sum_{\tilde{s},\tilde{a}}\frac{d_{\mu,b-1}(\tilde{s})\mu(\tilde{a}|\tilde{s})\mathsf{P}(s|\tilde{s},\tilde{a})}{d_{\mu,b}(s)}\cdot\mathbb{E}\left[(\rho_t^{b-1}F_t^{b-1})^2\big|\mathcal{F}_{t-1},s_t^b=s,s_t^{b-1}=\tilde{s},a_t^{b-1}=\tilde{a}\right] \\
& = \sum_{\tilde{s}\in\mathcal{S},\tilde{a}\in\mathcal{A}}d_{\mu,b-1}(\tilde{s})\mu(\tilde{a}|\tilde{s})\mathsf{P}(s|\tilde{s},\tilde{a})\cdot\frac{\pi^2(\tilde{a}|\tilde{s})}{\mu^2(\tilde{a}|\tilde{s})}\cdot\mathbb{E}\left[(F_t^{b-1})^2\big|\mathcal{F}_{t-1},s_t^{b-1}=\tilde{s}\right] \\
& \stackrel{(iii)}{=} \sum_{\tilde{s}\in\mathcal{S}}P_{\mu,\pi}(s|\tilde{s})r_{b-1}(\tilde{s}) \\
& = (P_{\mu,\pi}^\top r_{b-1})_s,
\end{aligned}
\tag{34}
$$

where $(i)$ follows from the law of total probability, $(ii)$ follows from the Bayes' rule, and in $(iii)$ we define $P_{\mu,\pi}\in\mathbb{R}^{|\mathcal{S}|\times|\mathcal{S}|}$ where $(P_{\mu,\pi})_{s,\tilde{s}}=\sum_{\tilde{a}\in\mathcal{A}}\frac{\pi^2(\tilde{a}|\tilde{s})}{\mu(\tilde{a}|\tilde{s})}\mathsf{P}(s|\tilde{s},\tilde{a})$ for each $(s,\tilde{s})\in\mathcal{S}\times\mathcal{S}$.

Substituting eqs. (33) and (34) into eq. (32) yields

$$
r_b = d_{\mu,b}+2\gamma P_\pi^\top f_{b-1}+\gamma^2 P_{\mu,\pi}^\top r_{b-1}.
$$

We also have the following inequality holds

$$
\begin{aligned}
\mathbf{1}^\top r_b &= \mathbf{1}^\top d_{\mu,b}+2\gamma\mathbf{1}^\top P_\pi^\top f_{b-1}+\gamma^2\mathbf{1}^\top P_{\mu,\pi}^\top r_{b-1} \\
& \stackrel{(i)}{=} 1+2\gamma\mathbf{1}^\top f_{b-1}+\gamma^2\mathbf{1}^\top P_{\mu,\pi}^\top r_{b-1} \\
& \stackrel{(ii)}{\leq} 1+2\gamma\mathbf{1}^\top f_{b-1}+\gamma^2\rho_{max}\mathbf{1}^\top r_{b-1},
\end{aligned}
\tag{35}
$$

where $(i)$ follows from $\mathbf{1}^\top P_\pi^\top=(P_\pi\mathbf{1})^\top=\mathbf{1}^\top$, and $(ii)$ follows from the facts that $r_{b-1}\succeq\mathbf{0}$ and

$$
\mathbf{1}^\top P_{\mu,\pi}^\top=(P_{\mu,\pi}\mathbf{1})^\top=\mathrm{vec}\left(\sum_{s\in\mathcal{S}}\sum_{\tilde{a}\in\mathcal{A}}\frac{\pi^2(\tilde{a}|\tilde{s})}{\mu(\tilde{a}|\tilde{s})}\mathsf{P}(s|\tilde{s},\tilde{a})\right)=\mathrm{vec}\left(\sum_{\tilde{a}\in\mathcal{A}}\frac{\pi^2(\tilde{a}|\tilde{s})}{\mu(\tilde{a}|\tilde{s})}\right)\preceq\rho_{max}\mathbf{1}^\top.
$$

Recursively applying eq. (35) yields

$$
\begin{aligned}
\mathbf{1}^\top r_b &\leq \sum_{\tau=0}^{b-1}(\gamma^2\rho_{max})^\tau(1+2\gamma\mathbf{1}^\top f_{b-\tau-1})+(\gamma^2\rho_{max})^b\mathbf{1}^\top r_0 \\
& \stackrel{(i)}{=} \sum_{\tau=0}^{b-1}(\gamma^2\rho_{max})^\tau(1+2\gamma\mathbf{1}^\top f_{b-\tau-1})+(\gamma^2\rho_{max})^b,
\end{aligned}
\tag{36}
$$

where $(i)$ follows from the fact that $\mathbf{1}^\top r_0=1$.

Recall that $f_b=d_{\mu,b}+\gamma P_\pi^\top f_{b-1}$ and $f=d_\mu+\gamma P_\pi^\top f$. We have

$$
\mathbf{1}^\top(f_\tau-f)=\mathbf{1}^\top(d_{\mu,\tau}-d_\mu)+\gamma\mathbf{1}^\top P_\pi^\top(f_{\tau-1}-f)\stackrel{(i)}{=}\gamma\mathbf{1}^\top(f_{\tau-1}-f)\stackrel{(ii)}{=}\gamma^\tau\mathbf{1}^\top(f_0-f),
$$

where $(i)$ follows from the facts that $d_{\mu,\tau}$ and $d_\mu$ are both probability distributions and $\mathbf{1}^\top d_{\mu,\tau}=\mathbf{1}^\top d_\mu=1$ and $\mathbf{1}^\top P_\pi^\top=\mathbf{1}^\top$ and $(ii)$ follows from recursively applying $(i)$.

Thus, we have

$$
|\mathbf{1}^\top f_\tau|=|(1-\gamma^\tau)\mathbf{1}^\top f+\gamma^\tau\mathbf{1}^\top f_0|\leq|(1-\gamma^\tau)\mathbf{1}^\top f|+\gamma^\tau\leq\|f\|_1+1.
\tag{37}
$$

Substituting eq. (37) into eq. (36) yields

$$\mathbf{1}^\top r_b \leq \sum_{\tau=0}^{b-1}(\gamma^2\rho_{max})^\tau(3+2\gamma\|f\|_1) + (\gamma^2\rho_{max})^b.$$

Under different conditions of $\rho_{max}$, the term $\mathbf{1}^\top r_b$ is upper bounded differently as following:

(a). $\gamma^2\rho_{max} > 1$

$$\mathbf{1}^\top r_b \leq \left(\frac{3+2\gamma\|f\|_1}{\gamma^2\rho_{max}-1} + 1\right)\gamma^{2b}\rho_{max}^b. \tag{38}$$

(b). $\gamma^2\rho_{max} = 1$

$$\mathbf{1}^\top r_b \leq (3+2\gamma\|f\|_1)\,b + 1. \tag{39}$$

(c). $\gamma^2\rho_{max} < 1$

$$\mathbf{1}^\top r_b \leq \frac{3+2\gamma\|f\|_1}{1-\gamma^2\rho_{max}} + 1. \tag{40}$$

Substituting the above inequalities into eq. (31), we can upper-bound the term $\mathbb{E}\left[\left\|\widehat{\mathcal{T}}_t(\theta_t)\right\|_2^2\middle|\mathcal{F}_{t-1}\right]$ under different conditions accordingly:

(a). $\gamma^2\rho_{max} > 1$

$$\mathbb{E}\left[\left\|\widehat{\mathcal{T}}_t(\theta_t)\right\|_2^2\middle|\mathcal{F}_{t-1}\right] \leq \rho_{max}\left(4(1+\gamma^2)B_\phi^2 B_\theta^2 + 2r_{max}^2\right)B_\phi^2\left(\frac{3+2\gamma\|f\|_1}{\gamma^2\rho_{max}-1}+1\right)\gamma^{2b}\rho_{max}^b$$
$$= C_{\sigma,1}\gamma^{2b}\rho_{max}^b,$$

where we specify $C_{\sigma,1} = \rho_{max}\left(4(1+\gamma^2)B_\phi^2 B_\theta^2 + 2r_{max}^2\right)B_\phi^2\left(\frac{3+2\gamma\|f\|_1}{\gamma^2\rho_{max}-1}+1\right)$.

(b). $\gamma^2\rho_{max} = 1$

$$\mathbb{E}\left[\left\|\widehat{\mathcal{T}}_t(\theta_t)\right\|_2^2\middle|\mathcal{F}_{t-1}\right] \leq \rho_{max}\left(4(1+\gamma^2)B_\phi^2 B_\theta^2 + 2r_{max}^2\right)B_\phi^2\left((3+2\gamma\|f\|_1)\,b+1\right)$$
$$= C_{\sigma,2}b,$$

where we specify $C_{\sigma,2} = \rho_{max}\left(4(1+\gamma^2)B_\phi^2 B_\theta^2 + 2r_{max}^2\right)B_\phi^2\left(4+2\gamma\|f\|_1\right)$.

(c). $\gamma^2\rho_{max} < 1$

$$\mathbb{E}\left[\left\|\widehat{\mathcal{T}}_t(\theta_t)\right\|_2^2\middle|\mathcal{F}_{t-1}\right] \leq \rho_{max}\left(4(1+\gamma^2)B_\phi^2 B_\theta^2 + 2r_{max}^2\right)B_\phi^2\left(\frac{3+2\gamma\|f\|_1}{1-\gamma^2\rho_{max}}+1\right)$$
$$:= C_{\sigma,3}.$$

where we specify $C_{\sigma,3} = \rho_{max}\left(4(1+\gamma^2)B_\phi^2 B_\theta^2 + 2r_{max}^2\right)B_\phi^2\left(\frac{3+2\gamma\|f\|_1}{1-\gamma^2\rho_{max}}+1\right)$.

To summarize, the variance term $\left\|\widehat{\mathcal{T}}_t(\theta_t)\right\|_2^2$ can be bounded as following

$$\mathbb{E}\left[\left\|\widehat{\mathcal{T}}_t(\theta_t)\right\|_2^2\middle|\mathcal{F}_{t-1}\right] \leq \sigma^2,$$

where

$$\sigma^2 = \begin{cases} \mathcal{O}(1), & \text{if } \gamma^2\rho_{max} < 1. \\ \mathcal{O}(b), & \text{if } \gamma^2\rho_{max} = 1. \\ \mathcal{O}((\gamma^2\rho_{max})^b), & \text{if } \gamma^2\rho_{max} > 1. \end{cases}$$

### D.3 PROOF OF THEOREM 1

**Theorem 3** (Formal Statement of Theorem 1). *Suppose Assumptions 1 and 2 hold. Consider PER-ETD(0) specified in Algorithm 1. Let the stepsize $\eta_t = \frac{2}{\mu_0(t+t_0)}$, where $t_0 = \frac{8L_0^2}{\mu_0^2}$, $\mu_0$ is defined in Lemma 7, and $L_0$ is defined in Lemma 8 in Appendix C. Let the projection set $\Theta = \{\theta \in \mathbb{R}^d : \|\theta\|_2 \le B_\theta\}$, where $B_\theta = \frac{\|\Phi^\top\|_2 r_{max}}{(1-\gamma)\mu_0}$ (which implies $\theta^* \in \Theta$). Then the convergence guarantee falls into the following two cases depending on the value of $\rho_{max}$.*

*(a) If $\gamma^2 \rho_{max} \le 1$, let $b = \max\left\{\left\lceil \frac{\log(\mu_0)-\log(5C_bB_\phi)}{\log(\xi)} \right\rceil, \frac{\log T}{\log(1/\xi)}\right\}$, where $C_b$ is a constant defined in the proof of Proposition 1 in Appendix D.1, $B_\phi := \max_{s\in\mathcal{S}} \|\phi(s)\|_2$, and $\xi := \max\{\gamma, \chi\}$. Then the output $\theta_T$ satisfies*

$$\mathbb{E}\left[\|\theta_T - \theta^*\|_2^2\right] \le \tilde{\mathcal{O}}\left(\frac{1}{T}\right).$$

*(b) If $\gamma^2 \rho_{max} > 1$, let $b = \max\left\{\left\lceil \frac{\log(\mu_0)-\log(5C_bB_\phi)}{\log(\xi)} \right\rceil, \frac{\log(T)}{\log(\gamma^2\rho_{max})+\log(1/\xi)}\right\}$, where $C_b$ is a constant whose definition could be found in the proof of Proposition 1 in Appendix D.1, $B_\phi := \max_{s\in\mathcal{S}} \|\phi(s)\|_2$, and $\xi := \max\{\gamma, \chi\}$. Then the output $\theta_T$ satisfies*

$$\mathbb{E}\left[\|\theta_T - \theta^*\|_2^2\right] \le \mathcal{O}\left(\frac{1}{T^a}\right),$$

*where $a = \frac{1}{\log_{1/\xi}(\gamma^2\rho_{max})+1} < 1$.*

*Thus, PER-ETD(0) attains an $\epsilon$-accurate solution with $\tilde{\mathcal{O}}\left(\frac{1}{\epsilon}\right)$ samples if $\gamma^2\rho_{max} \le 1$, and with $\tilde{\mathcal{O}}\left(\frac{1}{\epsilon^{1/a}}\right)$ samples if $\gamma^2\rho_{max} > 1$.*

*Proof.* Note the $\theta$ update specified in Algorithm 1 is the closed form solution of the following maximization problem.

$$\theta_{t+1} = \operatorname*{argmax}_{\theta\in\Theta} \eta_t \left\langle \widehat{\mathcal{T}}_t(\theta_t), \theta \right\rangle + \frac{1}{2}\|\theta - \theta_t\|_2^2.$$

Applying Lemma 9 with $\theta^* = \theta_{t+1}$, $\eta = \eta_t$, $G = \widehat{\mathcal{T}}_t(\theta_t)$, and $\theta_0 = \theta_t$ yields, for any $\theta \in \Theta$,

$$\eta_t \left\langle \widehat{\mathcal{T}}_t(\theta_t), \theta_{t+1} - \theta \right\rangle + \frac{1}{2}\|\theta_t - \theta_{t+1}\|_2^2 \le \frac{1}{2}\|\theta_t - \theta\|_2^2 - \frac{1}{2}\|\theta_{t+1} - \theta\|_2^2. \qquad (41)$$

Proceed with the first term in the above inequality as follows

$$\left\langle \widehat{\mathcal{T}}_t(\theta_t), \theta_{t+1} - \theta \right\rangle$$
$$= \langle \mathcal{T}(\theta_{t+1}), \theta_{t+1} - \theta \rangle + \langle \mathcal{T}(\theta_t) - \mathcal{T}(\theta_{t+1}), \theta_{t+1} - \theta \rangle + \left\langle \widehat{\mathcal{T}}_t(\theta_t) - \mathcal{T}(\theta_t), \theta_{t+1} - \theta \right\rangle$$
$$\overset{(i)}{\ge} \langle \mathcal{T}(\theta_{t+1}), \theta_{t+1} - \theta \rangle - L_0\|\theta_t - \theta_{t+1}\|_2\|\theta_{t+1} - \theta\|_2 + \left\langle \widehat{\mathcal{T}}_t(\theta_t) - \mathcal{T}(\theta_t), \theta_{t+1} - \theta \right\rangle$$
$$= \langle \mathcal{T}(\theta_{t+1}), \theta_{t+1} - \theta \rangle - L_0\|\theta_t - \theta_{t+1}\|_2\|\theta_{t+1} - \theta\|_2 + \left\langle \widehat{\mathcal{T}}_t(\theta_t) - \mathcal{T}(\theta_t), \theta_{t+1} - \theta_t \right\rangle$$
$$\quad + \left\langle \widehat{\mathcal{T}}_t(\theta_t) - \mathcal{T}(\theta_t), \theta_t - \theta \right\rangle$$
$$\ge \langle \mathcal{T}(\theta_{t+1}), \theta_{t+1} - \theta \rangle - L_0\|\theta_t - \theta_{t+1}\|_2\|\theta_{t+1} - \theta\|_2 - \left\|\widehat{\mathcal{T}}_t(\theta_t) - \mathcal{T}(\theta_t)\right\|_2 \cdot \|\theta_{t+1} - \theta_t\|_2$$
$$\quad + \left\langle \widehat{\mathcal{T}}_t(\theta_t) - \mathcal{T}(\theta_t), \theta_t - \theta \right\rangle,$$

where $(i)$ follows from the Cauchy-Schwartz inequality and Lemma 8.

Substituting the above inequality into eq. (41) yields

$$\eta_t \langle \mathcal{T}(\theta_{t+1}), \theta_{t+1} - \theta \rangle - \eta_t L_0\|\theta_t - \theta_{t+1}\|_2\|\theta_{t+1} - \theta\|_2 - \eta_t \left\|\widehat{\mathcal{T}}_t(\theta_t) - \mathcal{T}(\theta_t)\right\|_2 \cdot \|\theta_{t+1} - \theta_t\|_2$$

$$+ \eta_t \left\langle \widehat{\mathcal{T}}_t(\theta_t) - \mathcal{T}(\theta_t), \theta_t - \theta \right\rangle + \frac{1}{2} \|\theta_t - \theta_{t+1}\|_2^2 \leq \frac{1}{2} \|\theta_t - \theta\|_2^2 - \frac{1}{2} \|\theta_{t+1} - \theta\|_2^2. \qquad (42)$$

Applying Young's inequality to $\eta_t \left\| \widehat{\mathcal{T}}_t(\theta_t) - \mathcal{T}(\theta_t) \right\|_2 \cdot \|\theta_{t+1} - \theta_t\|_2$ yields

$$\eta \left\| \widehat{\mathcal{T}}_t(\theta_t) - \mathcal{T}(\theta_t) \right\|_2 \cdot \|\theta_{t+1} - \theta_t\|_2 \leq \frac{1}{4} \|\theta_{t+1} - \theta_t\|_2^2 + \eta_t^2 \left\| \widehat{\mathcal{T}}_t(\theta_t) - \mathcal{T}(\theta_t) \right\|_2^2,$$

and applying Young's inequality to $\eta_t L_0 \|\theta_t - \theta_{t+1}\|_2 \|\theta_{t+1} - \theta\|_2$ yields

$$\eta_t L_0 \|\theta_t - \theta_{t+1}\|_2 \|\theta_{t+1} - \theta\|_2 \leq \frac{1}{4} \|\theta_t - \theta_{t+1}\|_2^2 + \eta_t^2 L_0^2 \|\theta_{t+1} - \theta\|_2^2.$$

Substituting the above two inequalities into eq. (42) yields

$$\frac{1}{2} \|\theta_t - \theta\|_2^2 \geq \eta_t \left\langle \mathcal{T}(\theta_{t+1}), \theta_{t+1} - \theta \right\rangle + \left( \frac{1}{2} - \eta_t^2 L_0^2 \right) \|\theta_{t+1} - \theta\|_2^2$$

$$+ \eta_t \left\langle \widehat{\mathcal{T}}_t(\theta_t) - \mathcal{T}(\theta_t), \theta_t - \theta \right\rangle - \eta_t^2 \left\| \widehat{\mathcal{T}}_t(\theta_t) - \mathcal{T}(\theta_t) \right\|_2^2.$$

Taking expectation conditioned on $\mathcal{F}_{t-1}$ on the both sides of the above inequality, we obtain

$$\frac{1}{2} \|\theta_t - \theta\|_2^2 \geq \eta_t \mathbb{E}\left[ \langle \mathcal{T}(\theta_{t+1}), \theta_{t+1} - \theta \rangle | \mathcal{F}_{t-1} \right] + \left( \frac{1}{2} - \eta_t^2 L_0^2 \right) \mathbb{E}\left[ \|\theta_{t+1} - \theta\|_2^2 | \mathcal{F}_{t-1} \right]$$

$$+ \eta_t \left\langle \mathbb{E}\left[ \widehat{\mathcal{T}}_t(\theta_t) - \mathcal{T}(\theta_t) \Big| \mathcal{F}_{t-1} \right], \theta_t - \theta \right\rangle - \eta_t^2 \mathbb{E}\left[ \left\| \widehat{\mathcal{T}}_t(\theta_t) - \mathcal{T}(\theta_t) \right\|_2^2 \Big| \mathcal{F}_{t-1} \right]. \qquad (43)$$

Letting $\theta = \theta^*$ and applying Lemma 7 to eq. (43) yields

$$\frac{1}{2} \|\theta_t - \theta^*\|_2^2 \geq \left( \frac{1}{2} + \mu_0 \eta_t - \eta_t^2 L_0^2 \right) \mathbb{E}\left[ \|\theta_{t+1} - \theta^*\|_2^2 | \mathcal{F}_{t-1} \right]$$

$$- \eta_t C_b B_\phi \xi^b \|\theta_t - \theta^*\|_2^2 - \eta_t C_b \epsilon_{approx} \xi^b \|\theta_t - \theta^*\|_2$$

$$- \eta_t^2 \mathbb{E}\left[ \left\| \widehat{\mathcal{T}}_t(\theta_t) - \mathcal{T}(\theta_t) \right\|_2^2 \Big| \mathcal{F}_{t-1} \right]$$

$$\overset{(i)}{\geq} \left( \frac{1}{2} + \mu_0 \eta_t - \eta_t^2 L_0^2 \right) \mathbb{E}\left[ \|\theta_{t+1} - \theta^*\|_2^2 | \mathcal{F}_{t-1} \right]$$

$$- \eta_t C_b B_\phi \xi^b \|\theta_t - \theta^*\|_2^2 - \eta_t C_b \epsilon_{approx} \xi^b \|\theta_t - \theta^*\|_2 - 4\eta_t^2 C_b^2 B_\phi^2 \xi^{2b} \|\theta_t - \theta^*\|_2^2$$

$$- 4\eta_t^2 C_b^2 \xi^{2b} \epsilon_{approx}^2 - 2\sigma^2 \eta_t^2, \qquad (44)$$

where $(i)$ follows from Propositions 1 and 2, and the facts that $(x + y)^2 \leq 2x^2 + 2y^2$ and

$$\left\| \widehat{\mathcal{T}}_t(\theta_t) - \mathcal{T}(\theta_t) \right\|_2^2 \leq 2 \left\| \mathbb{E}\left[ \widehat{\mathcal{T}}_t(\theta_t) \Big| \mathcal{F}_{t-1} \right] - \mathcal{T}(\theta_t) \right\|_2^2 + 2 \left\| \mathbb{E}\left[ \widehat{\mathcal{T}}_t(\theta_t) \Big| \mathcal{F}_{t-1} \right] - \widehat{\mathcal{T}}_t(\theta_t) \right\|_2^2$$

$$\leq 2 \left\| \widehat{\mathcal{T}}(\theta_t) \right\|_2^2 + 2 \left\| \mathbb{E}\left[ \widehat{\mathcal{T}}_t(\theta_t) \Big| \mathcal{F}_{t-1} \right] - \widehat{\mathcal{T}}_t(\theta_t) \right\|_2^2.$$

Taking expectation on both sides of the above inequality yields

$$\left( \frac{1}{2} + \mu_0 \eta_t - \eta_t^2 L_0^2 \right) \mathbb{E}\left[ \|\theta_{t+1} - \theta^*\|_2^2 \right]$$

$$\leq \left( \frac{1}{2} + C_b B_\phi \xi^b \eta_t + 4 C_b^2 B_\phi^2 \xi^{2b} \eta_t^2 \right) \mathbb{E}\left[ \|\theta_t - \theta^*\|_2^2 \right] + \eta_t C_b B_\theta \epsilon_{approx} \xi^b$$

$$+ 4\eta_t^2 C_b^2 \xi^{2b} \epsilon_{approx}^2 + 2\sigma^2 \eta_t^2.$$

Recall that we set $t_0 = \frac{8L_0^2}{\mu_0^2}$. Let $\alpha_t = (t + t_0 + 1)(t + t_0 + 2)$. Multiplying $2\alpha_t$ on both sides of the above inequality and telescoping from $t = 0, 1, 2, \ldots, T - 1$ yields

$$\sum_{t=0}^{T-1} \alpha_t \left( 1 + 2\mu_0 \eta_t - 2\eta_t^2 L_0^2 \right) \mathbb{E}\left[ \|\theta_{t+1} - \theta^*\|_2^2 \right]$$

$$\leq \sum_{t=0}^{T-1} \alpha_t \left(1 + 2C_b B_\phi \xi^b \eta_t + 8C_b^2 B_\phi^2 \xi^{2b} \eta_t^2\right) \mathbb{E}\left[\|\theta_t - \theta^*\|_2^2\right]$$

$$+ \left(4\sigma^2 + 8C_b^2 \xi^{2b} \epsilon_{approx}^2\right) \sum_{t=0}^{T-1} \alpha_t \eta_t^2 + 2C_b B_\theta \epsilon_{approx} \xi^b \sum_{t=0}^{T-1} \alpha_t \eta_t. \tag{45}$$

Recall the setting of $\eta_t$, we have

$$1 + 2\mu_0 \eta_t - 2\eta_t^2 L_0^2 = 1 + \frac{3\mu_0 \eta_t}{2}\left(\frac{4}{3} - \frac{4}{3\mu_0}\eta_t L_0^2\right)$$

$$= 1 + \frac{3\mu_0 \eta_t}{2}\left(1 + \frac{1}{3}\left(1 - \frac{4}{\mu_0}\eta_t L_0^2\right)\right)$$

$$\overset{(i)}{\geq} 1 + \frac{3\mu_0 \eta_t}{2},$$

where $(i)$ follows from the fact that $1 - \frac{4\eta_t L_0^2}{\mu_0} \geq 1 - \frac{8L_0^2}{\mu_0^2 t_0} \geq 0$. Multiplying $\alpha_t$ on both sides of the above inequality yields

$$\alpha_t(1 + 2\mu_0 \eta_t - 2\eta_t^2 L_0^2) \geq (t + t_0 + 1)(t + t_0 + 2)\left(1 + \frac{3\mu_0}{2}\frac{2}{\mu_0(t + t_0)}\right)$$

$$= (t + t_0 + 1)(t + t_0 + 2)(t + t_0 + 3)/(t + t_0). \tag{46}$$

Under appropriate value of $b$, we have $C_b B_\phi \xi^b \leq \frac{\mu_0}{5}$. Which implies that

$$2C_b B_\phi \xi^b \eta_t + 8C_b^2 B_\phi^2 \xi^{2b} \eta_t^2 = \frac{\mu_0 \eta_t}{2} + \frac{\mu_0 \eta_t}{2}\left(\frac{4C_b B_\phi \xi^b}{\mu_0} + \frac{16C_b^2 B_\phi^2 \xi^{2b} \eta_t}{\mu_0} - 1\right)$$

$$\leq \frac{\mu_0 \eta_t}{2} + \frac{\mu_0 \eta_t}{2}\left(\frac{4}{5} + \frac{16\mu_0 \eta_t}{25} - 1\right)$$

$$\leq \frac{\mu_0 \eta_t}{2} + \frac{\mu_0 \eta_t}{2}\left(\frac{4\mu_0^2}{25L_0^2} - \frac{1}{5}\right)$$

$$\leq \frac{\mu_0 \eta_t}{2}.$$

Multiplying $\alpha_{t+1}$ on both sides of the above inequality yields

$$\alpha_{t+1}(1 + 2C_b B_\phi \xi^b \eta_{t+1} + 8C_b^2 B_\phi^2 \xi^{2b} \eta_{t+1}^2) \leq \alpha_{t+1}\left(1 + \frac{\mu_0 \eta_{t+1}}{2}\right)$$

$$\leq \alpha_{t+1}\left(1 + \frac{\mu_0 \eta_{t+1}}{2}\right)$$

$$= (t + t_0 + 2)(t + t_0 + 3)\left(1 + \frac{\mu_0}{2}\frac{2}{\mu_0(t + t_0 + 1)}\right)$$

$$= (t + t_0 + 2)^2(t + t_0 + 3)/(t + t_0 + 1). \tag{47}$$

Equations (46) and (47) together imply that

$$\alpha_t(1 + 2\mu_0 \eta_t - 2\eta_t^2 L_0^2) - \alpha_{t+1}(1 + 2C_b B_\phi \xi^b \eta_t + 8C_b^2 B_\phi^2 \xi^{2b} \eta_t^2)$$

$$\geq \frac{(t + t_0 + 1)(t + t_0 + 2)(t + t_0 + 3)}{t + t_0} - \frac{(t + t_0 + 2)^2(t + t_0 + 3)}{t + t_0 + 1}$$

$$= \frac{(t + t_0 + 2)(t + t_0 + 3)}{(t + t_0)(t + t_0 + 1)}\left((t + t_0 + 1)^2 - (t + t_0)(t + t_0 + 2)\right)$$

$$= \frac{(t + t_0 + 2)(t + t_0 + 3)}{(t + t_0)(t + t_0 + 1)}$$

$$> 0.$$

The above inequality shows that the $\|\theta_t - \theta^*\|_2^2, t = 1, \ldots, T-1$, terms on both sides of eq. (45) can be canceled, which indicates the following

$$(T + t_0)(T + t_0 + 1) \left(1 + 2\mu_0\eta_{T-1} - 2\eta_{T-1}^2 L_0^2\right) \mathbb{E}\left[\|\theta_T - \theta^*\|_2^2\right]$$
$$\leq (t_0 + 1)(t_0 + 2)\left(1 + 2C_b B_\phi \xi^b \eta_0 + 8C_b^2 B_\phi^2 \xi^{2b} \eta_0^2\right)\|\theta_0 - \theta^*\|_2^2$$
$$+ \left(4\sigma^2 + 8C_b^2 \xi^{2b}\epsilon_{approx}^2\right)\sum_{t=0}^{T-1}\alpha_t\eta_t^2 + 2C_b B_\theta\epsilon_{approx}\xi^b\sum_{t=0}^{T-1}\alpha_t\eta_t. \quad (48)$$

Note that $\sum_{t=0}^{T-1}\alpha_t\eta_t^2 \leq \sum_{t=0}^{T-1}\frac{6}{\mu_0^2} \leq \frac{6T}{\mu_0^2}, 1 + 2\mu_0\eta_{T-1} - 2\eta_{T-1}^2 L_0^2 \geq 1$, and

$$\sum_{t=0}^{T-1}\alpha_t\eta_t \leq \frac{4}{\mu_0}\sum_{t=0}^{T-1}(t + t_0 + 2) \leq \frac{2}{\mu_0}(T + t_0 + 2)^2.$$

Dividing $(T + t_0)(T + t_0 + 1)\left(1 + 2\mu_0\eta_{T-1} - 2\eta_{T-1}^2 L_0^2\right)$ on both sides of eq. (48) yields

$$\mathbb{E}\left[\|\theta_T - \theta^*\|_2^2\right]$$
$$\leq \frac{(t_0 + 1)(t_0 + 2)}{(T + t_0)(T + t_0 + 1)}\left(1 + \frac{\mu_0\eta_0}{2}\right)\|\theta_0 - \theta^*\|_2^2$$
$$+ \frac{24\sigma^2 + 48C_b^2\xi^{2b}\epsilon_{approx}^2}{\mu_0^2}\frac{1}{T + t_0 + 1} + \frac{4C_b B_\theta\epsilon_{approx}\xi^b}{\mu_0}\frac{(T + t_0 + 2)^2}{(T + t_0 + 1)(T + t_0)}$$
$$= \mathcal{O}\left(\frac{\|\theta_0 - \theta^2\|_2^2}{T^2}\right) + \mathcal{O}\left(\frac{\sigma^2}{T}\right) + \mathcal{O}\left(\frac{C_b^2\xi^{2b}}{T}\right) + \mathcal{O}\left(C_b\xi^b\right). \quad (49)$$

Based on different conditions of $\sigma^2$, we pick different $b$ and the convergence rate is as follows.

$(a)$. $\gamma^2\rho_{max} \leq 1$, Proposition 2 show that $\sigma^2 \leq \mathcal{O}(b)$. We specify

$$b = \max\left\{\left\lceil\frac{\log(\mu_0) - \log(5C_b B_\phi)}{\log(\xi)}\right\rceil, \frac{\log T}{\log(1/\xi)}\right\} \leq \mathcal{O}(\log(T)).$$

Equation (49) yields,

$$\mathbb{E}\left[\|\theta_T - \theta^*\|_2^2\right] = \mathcal{O}\left(\frac{\|\theta_0 - \theta^2\|_2^2}{T^2}\right) + \mathcal{O}\left(\frac{\log(T)}{T}\right) + \mathcal{O}\left(\frac{1}{T^2}\right) + \mathcal{O}\left(\frac{1}{T}\right) = \tilde{\mathcal{O}}\left(\frac{1}{T}\right).$$

$(b)$. $\gamma^2\rho_{max} > 1$, Proposition 2 show that $\sigma^2 = \mathcal{O}\left((\gamma^2\rho_{max})^b\right)$. We specify

$$b = \max\left\{\left\lceil\frac{\log(\mu_0) - \log(5C_b B_\phi)}{\log(\xi)}\right\rceil, \frac{\log(T)}{\log(\gamma^2\rho_{max}) + \log(1/\xi)}\right\}.$$

Equation (49) yields

$$\mathbb{E}\left[\|\theta_T - \theta^*\|_2^2\right] = \mathcal{O}\left(\frac{\|\theta_0 - \theta^2\|_2^2}{T^2}\right) + \mathcal{O}\left(\frac{T^{1-a}}{T}\right) + \mathcal{O}\left(\frac{C_b^2}{T^{1+a}}\right) + \mathcal{O}\left(\frac{C_b}{T^a}\right) = \mathcal{O}\left(\frac{1}{T^a}\right).$$

$\square$

# E    PROOFS OF PROPOSITIONS AND THEOREM FOR PER-ETD($\lambda$)

## E.1    PROOF OF PROPOSITION 3

Define the matrix $A_t := \rho_t^b e_t^b(\phi_t^b - \gamma\phi_t^{b+1})$ and $c_t := r_t^b\rho_t^b e_t^b$. We have

$$\widehat{\mathcal{T}}_t^\lambda(\theta_t) = A_t\theta_t - c_t. \quad (50)$$

Recall that $\theta_t$ is $\mathcal{F}_{t-1}$-measurable. We have

$$\mathbb{E}\left[\widehat{\mathcal{T}}_t^\lambda(\theta_t)\Big|\mathcal{F}_{t-1}\right] = \mathbb{E}\left[A_t|\mathcal{F}_{t-1}\right]\theta_t - \mathbb{E}\left[c_t|\mathcal{F}_{t-1}\right].$$

To bound the bias error term $\left\| \mathbb{E}\left[\widehat{\mathcal{T}}_t^\lambda(\theta_t)\big|\mathcal{F}_{t-1}\right] - \mathcal{T}^\lambda(\theta_t) \right\|_2$, we first take conditional expectations on $A_t$ and $c_t$, respectively, as following

$$
\begin{aligned}
&\mathbb{E}\left[A_t|\mathcal{F}_{t-1}\right] \\
&= \mathbb{E}\left[\rho_t^b e_t^b(\phi_t^b - \gamma\phi_t^{b+1})^\top\big|\mathcal{F}_{t-1}\right] \\
&\stackrel{(i)}{=} \sum_{s\in\mathcal{S},a\in\mathcal{A},s'\in\mathcal{S}} \mathbb{P}\left(s_t^b = s, a_t^b = a, s_t^{b+1} = s'\big|\mathcal{F}_{t-1}\right) \\
&\qquad\qquad \cdot \mathbb{E}\left[\rho_t^b e_t^b(\phi_t^b - \gamma\phi_t^{b+1})^\top\big|\mathcal{F}_{t-1}, s_t^b = s, a_t^b = a, s_t^{b+1} = s'\right] \\
&\stackrel{(ii)}{=} \sum_{s,a,s'} \mathbb{P}\left(s_t^b = s|\mathcal{F}_{t-1}\right)\mu(a|s)\mathsf{P}(s'|s,a)\cdot\frac{\pi(a|s)}{\mu(a|s)}\mathbb{E}\left[e_t^b|s_t^b = s,\mathcal{F}_{t-1}\right](\phi(s) - \gamma\phi(s'))^\top \\
&= \sum_{s\in\mathcal{S}} \mathbb{P}\left(s_t^b = s|\mathcal{F}_{t-1}\right)\mathbb{E}\left[e_t^b|\mathcal{F}_{t-1}, s_t^b = s\right]\sum_{a\in\mathcal{A},s'\in\mathcal{S}}\pi(a|s)\mathsf{P}(s'|s,a)(\phi(s) - \gamma\phi(s'))^\top \\
&= \sum_{s\in\mathcal{S}} \mathbb{P}\left(s_t^b = s|\mathcal{F}_{t-1}\right)\mathbb{E}\left[e_t^b|\mathcal{F}_{t-1}, s_t^b = s\right]\cdot\left((\Phi)_{(s,\cdot)} - \gamma(P_\pi\Phi)_{(s,\cdot)}\right),
\end{aligned}
\tag{51}
$$

where $(i)$ follows from the law of total probability and $(ii)$ follows from the Markov property and the fact that $e_t^b$ only depends on $(s_t^0, a_t^0, s_t^1, \ldots, s_t^b)$.

Define $\beta_\tau(s) = \mathbb{P}\left(s_t^\tau = s|\mathcal{F}_{t-1}\right)\mathbb{E}\left[e_t^b|\mathcal{F}_{t-1}, s_t^\tau = s\right]$. We have

$$
\begin{aligned}
&\beta_b(s) = \mathbb{P}\left(s_t^b = s|\mathcal{F}_{t-1}\right)\mathbb{E}\left[e_t^b|s_t^b = s,\mathcal{F}_{t-1}\right] \\
&\stackrel{(i)}{=} \mathbb{P}\left(s_t^b = s|\mathcal{F}_{t-1}\right) \\
&\quad \cdot \sum_{\tilde{s}\in\mathcal{S},\tilde{a}\in\mathcal{A}} \mathbb{P}\left(s_t^{b-1} = \tilde{s}, a_t^{b-1} = \tilde{a}\big|s_t^b = s,\mathcal{F}_{t-1}\right) \\
&\quad \cdot \mathbb{E}\left[\gamma\lambda\rho_t^{b-1}e_t^{b-1} + (\lambda + (1-\lambda)(1 + \rho_t^{b-1}\gamma F_t^{b-1})\phi_t^b)\big|s_t^{b-1} = \tilde{s}, a_t^{b-1} = \tilde{a}, s_t^b = s,\mathcal{F}_{t-1}\right] \\
&\stackrel{(ii)}{=} \mathbb{P}\left(s_t^b = s|\mathcal{F}_{t-1}\right)\phi(s) \\
&\quad + \mathbb{P}\left(s_t^b = s|\mathcal{F}_{t-1}\right)\sum_{\tilde{s}\in\mathcal{S},\tilde{a}\in\mathcal{A}} \frac{\mathbb{P}\left(s_t^{b-1} = \tilde{s}|\mathcal{F}_{t-1}\right)\mu(\tilde{a}|\tilde{s})\mathsf{P}(s|\tilde{s},\tilde{a})}{\mathbb{P}\left(s_t^b = s|\mathcal{F}_{t-1}\right)} \\
&\qquad\qquad \cdot \frac{\pi(\tilde{a}|\tilde{s})}{\mu(\tilde{a}|\tilde{s})}\cdot\mathbb{E}\left[\gamma\lambda e_t^{b-1} + (1-\lambda)\gamma F_t^{b-1}\phi(s)\big|s_t^{b-1} = \tilde{s},\mathcal{F}_{t-1}\right] \\
&= \mathbb{P}\left(s_t^b = s|\mathcal{F}_{t-1}\right)\phi(s) + \sum_{\tilde{s}\in\mathcal{S}}\mathbb{P}\left(s_t^{b-1} = \tilde{s}|\mathcal{F}_{t-1}\right)P_\pi(s|\tilde{s}) \\
&\qquad\qquad \cdot \mathbb{E}\left[\gamma\lambda e_t^{b-1} + (1-\lambda)\gamma F_t^{b-1}\phi(s)\big|s_t^{b-1} = \tilde{s},\mathcal{F}_{t-1}\right] \\
&\stackrel{(iii)}{=} (\lambda d_{\mu,b}(s) + (1-\lambda)f_b(s))\cdot\phi(s) + \gamma\lambda(P_\pi^\top\beta_{b-1})_s,
\end{aligned}
\tag{52}
$$

where $(i)$ follows from the law of total probability, $(ii)$ follows from the Bayes rule and the Markov property, and $(iii)$ follows from the following definitions: $d_{\mu,b}(s) = \mathbb{P}\left(s_t^b = s|\mathcal{F}_{t-1}\right)$, $f_b(s) = d_{\mu,b}(s)\mathbb{E}\left[F_t^b = s|s_t^b = s,\mathcal{F}_{t-1}\right]$, $f_b = d_{\mu,b} + \gamma P_\pi^\top f_{b-1}$, and $\beta_\tau(s) = \mathbb{P}\left(s_t^\tau = s|\mathcal{F}_{t-1}\right)\mathbb{E}\left[e_t^b|\mathcal{F}_{t-1}, s_t^\tau = s\right]$.

Define the matrix $\beta_\tau \in \mathbb{R}^{d\times|\mathcal{S}|}$, where $\beta_\tau = (\beta_\tau(1), \beta_\tau(2), \ldots, \beta_\tau(|\mathcal{S}|))$. Then, eq. (52) implies that

$$
\beta_b = \lambda\Phi^\top D_{\mu,b} + (1-\lambda)\Phi^\top F_b + \gamma\lambda\beta_{b-1}P_\pi,
\tag{53}
$$

where $D_{\mu,b} := \operatorname{diag}(d_{\mu,b}(1), d_{\mu,b}(2), \ldots, d_{\mu,b}(|\mathcal{S}|))$ and $F_b = \operatorname{diag}(f_b)$.

Recursively applying the above equality yields

$$
\beta_b = (\gamma\lambda)^b\beta_0 P_\pi^b + \lambda\sum_{\tau=0}^{b-1}(\gamma\lambda)^\tau\Phi^\top D_{\mu,b-\tau}P_\pi^\tau + (1-\lambda)\sum_{\tau=0}^{b-1}(\gamma\lambda)^\tau\Phi^\top F_{b-\tau}P_\pi^\tau.
\tag{54}
$$

Taking expectation of $c_t$ conditioned on $\mathcal{F}_{t-1}$, we have

$$
\begin{aligned}
\mathbb{E}\left[c_t|\mathcal{F}_{t-1}\right] &= \mathbb{E}\left[\rho_t^b e_t^b r_t^b|\mathcal{F}_{t-1}\right] \\
&= \sum_{s\in\mathcal{S},a\in\mathcal{A}} \mathbb{P}\left(s_t^b = s, a_t^b = a|\mathcal{F}_{t-1}\right) \mathbb{E}\left[\rho_t^b e_t^b r_t^b|s_t^b = s, a_t^b = a, \mathcal{F}_{t-1}\right] \\
&= \sum_{s\in\mathcal{S},a\in\mathcal{A}} \mathbb{P}\left(s_t^b = s|\mathcal{F}_{t-1}\right)\mu(a|s)\cdot\frac{\pi(a|s)}{\mu(a|s)}r(s,a)\mathbb{E}\left[e_t^b|s_t^b = s, \mathcal{F}_{t-1}\right] \\
&= \sum_{s\in\mathcal{S}} r_\pi(s)\mathbb{P}\left(s_t^b = s|\mathcal{F}_{t-1}\right)\mathbb{E}\left[e_t^b|s_t^b = s, \mathcal{F}_{t-1}\right] \\
&= \sum_{s\in\mathcal{S}} r_\pi(s)\beta_b(s).
\end{aligned}
\tag{55}
$$

Substituting eqs. (51) and (55) into eq. (50) yields

$$
\mathbb{E}\left[\widehat{\mathcal{T}}_t^\lambda(\theta_t)\Big|\mathcal{F}_{t-1}\right] = \sum_{s\in\mathcal{S}}\beta_b(s)\left(\Phi\theta_t - \gamma P_\pi\Phi\theta_t - r_\pi\right)_s = \beta_b\left(\Phi\theta_t - \gamma P_\pi\Phi\theta_t - r_\pi\right).
$$

Recall the definition of $\mathcal{T}^\lambda(\theta)$. We have

$$
\mathcal{T}^\lambda(\theta_t) - \mathbb{E}\left[\widehat{\mathcal{T}}_t^\lambda(\theta_t)\Big|\mathcal{F}_{t-1}\right] = \left(\Phi^\top M(I-\gamma\lambda P_\pi)^{-1} - \beta_b\right)\left(\Phi\theta_t - \gamma P_\pi\Phi\theta_t - r_\pi\right).
\tag{56}
$$

We then proceed to bound the term $\Phi^\top M(I-\gamma\lambda P_\pi)^{-1} - \beta_b$

$$
\begin{aligned}
&\Phi^\top M(I-\gamma\lambda P_\pi)^{-1} - \beta_b \\
&\overset{(i)}{=} \Phi^\top M\left(\sum_{\tau=0}^\infty(\gamma\lambda)^\tau P_\pi^\tau\right) - \beta_b \\
&\overset{(ii)}{=} \Phi^\top\left(\lambda D_\mu + (1-\lambda)F\right)\left(\sum_{\tau=0}^\infty(\gamma\lambda)^\tau P_\pi^\tau\right) \\
&\quad - \left((\gamma\lambda)^b\beta_0 P_\pi^b + \lambda\sum_{\tau=0}^{b-1}(\gamma\lambda)^\tau\Phi^\top D_{\mu,b-\tau}P_\pi^\tau + (1-\lambda)\sum_{\tau=0}^{b-1}(\gamma\lambda)^\tau\Phi^\top F_{b-\tau}P_\pi^\tau\right) \\
&= \lambda\sum_{\tau=0}^{b-1}(\gamma\lambda)^\tau\Phi^\top\left(D_\mu - D_{\mu,b-\tau}\right)P_\pi^\tau + (1-\lambda)\sum_{\tau=0}^{b-1}(\gamma\lambda)^\tau\Phi^\top\left(F - F_{b-\tau}\right)P_\pi^\tau \\
&\quad + \sum_{\tau=b}^\infty(\gamma\lambda)^\tau\Phi^\top(\lambda D_\mu + (1-\lambda)F)P_\pi^\tau - \lambda(\gamma\lambda)^b\Phi^\top D_{\mu,0}P_\pi^b,
\end{aligned}
$$

where $(i)$ follows from the fact that $(I-\gamma P_\pi)^{-1} = \sum_{\tau=0}^\infty\gamma^\tau P_\pi^\tau$, and $(ii)$ follows from eq. (54). Substituting the above equality into eq. (56) and taking $\ell_2$ norm on the both sides yield

$$
\begin{aligned}
&\left\|\mathcal{T}^\lambda(\theta_t) - \mathbb{E}\left[\widehat{\mathcal{T}}_t^\lambda(\theta_t)\Big|\mathcal{F}_{t-1}\right]\right\|_2 \\
&= \left\|\left(\lambda\sum_{\tau=0}^{b-1}(\gamma\lambda)^\tau\Phi^\top\left(D_\mu - D_{\mu,b-\tau}\right)P_\pi^\tau + (1-\lambda)\sum_{\tau=0}^{b-1}(\gamma\lambda)^\tau\Phi^\top\left(F - F_{b-\tau}\right)P_\pi^\tau\right.\right. \\
&\quad \left.\left. + \sum_{\tau=b}^\infty(\gamma\lambda)^\tau\Phi^\top(\lambda D_\mu + (1-\lambda)F)P_\pi^\tau - \lambda(\gamma\lambda)^b\Phi^\top D_{\mu,0}P_\pi^b\right)\left(\Phi\theta_t - \gamma P_\pi\theta_t - r_\pi\right)\right\|_2 \\
&\leq \lambda\sum_{\tau=0}^{b-1}(\gamma\lambda)^\tau\left\|\Phi^\top\left(D_\mu - D_{\mu,b-\tau}\right)P_\pi^\tau\left(\Phi\theta_t - \gamma P_\pi\Phi\theta_t - r_\pi\right)\right\|_2 \\
&\quad + (1-\lambda)\sum_{\tau=0}^{b-1}(\gamma\lambda)^\tau\left\|\Phi^\top\left(F - F_{b-\tau}\right)P_\pi^\tau\left(\Phi\theta_t - \gamma P_\pi\Phi\theta_t - r_\pi\right)\right\|_2
\end{aligned}
$$

$$+ \sum_{\tau=b}^{\infty} (\gamma\lambda)^{\tau} \left\| \Phi^{\top} (\lambda D_{\mu} + (1-\lambda)F) P_{\pi}^{\tau} (\Phi\theta_t - \gamma P_{\pi}\Phi\theta_t - r_{\pi}) \right\|_2$$

$$+ \lambda(\gamma\lambda)^b \left\| \Phi^{\top} D_{\mu,0} P_{\pi}^b (\Phi\theta_t - \gamma P_{\pi}\Phi\theta_t - r_{\pi}) \right\|_2$$

$$\overset{(i)}{\leq} \lambda \sum_{\tau=0}^{b-1} (\gamma\lambda)^{\tau} B_{\phi} \|d_{\mu} - d_{\mu,b-\tau}\|_1 (1+\gamma) \left( B_{\phi}\|\theta_t - \theta_{\lambda}^*\|_2 + \epsilon_{approx} \right)$$

$$+ (1-\lambda) \sum_{\tau=0}^{b-1} (\gamma\lambda)^{\tau} B_{\phi} \|f - f_{b-\tau}\|_1 (1+\gamma) \left( B_{\phi}\|\theta_t - \theta_{\lambda}^*\|_2 + C\epsilon_{approx} \right)$$

$$+ \sum_{\tau=b}^{\infty} (\gamma\lambda)^{\tau} B_{\phi} \left( \lambda\|d_{\mu}\|_1 + (1-\lambda)\|f\|_1 \right) (1+\gamma) \left( B_{\phi}\|\theta_t - \theta_{\lambda}^*\|_2 + \epsilon_{approx} \right)$$

$$+ \lambda(\gamma\lambda)^b B_{\phi} \|d_{\mu,0}\|_1 (1+\gamma) \left( B_{\phi}\|\theta_t - \theta_{\lambda}^*\|_2 + \epsilon_{approx} \right)$$

$$\overset{(ii)}{\leq} \left( B_{\phi}^2 \|\theta_t - \theta_{\lambda}^*\|_2 + B_{\phi}\epsilon_{approx} \right)$$

$$\cdot \left( \sum_{\tau=0}^{b-1} (\gamma\lambda)^{\tau} \left( \lambda(1+\gamma)C_M\chi^{b-\tau} + (1-\lambda)(1+\gamma) \left( \frac{C_M}{|\gamma-\chi|}\xi^{b-\tau} + \gamma^{b-\tau}(1+\|f\|_1) \right) \right) \right)$$

$$+ (1+\gamma) \left( B_{\phi}^2 \|\theta_t - \theta_{\lambda}^*\|_2 + B_{\phi}\epsilon_{approx} \right) \left( \left( \frac{\lambda + (1-\lambda)\|f\|_1}{1-\gamma\lambda} + \lambda \right) (\gamma\lambda)^b \right)$$

$$\overset{(iii)}{\leq} \left( B_{\phi}^2 \|\theta_t - \theta_{\lambda}^*\|_2 + B_{\phi}\epsilon_{approx} \right)$$

$$\cdot (1+\gamma) \left( \frac{\lambda}{|\chi - \gamma\lambda|} C_M\xi^b + \frac{C_M(1-\lambda)}{|\gamma-\chi|(\xi-\gamma\lambda)}\xi^b + (1+\|f\|_1)\gamma^b + \left( \frac{\lambda+(1-\lambda)\|f\|_1}{1-\gamma\lambda} + \lambda \right)(\gamma\lambda)^b \right)$$

$$\overset{(iv)}{\leq} C_{b,\lambda} \left( B_{\phi}\|\theta_t - \theta_{\lambda}^*\|_2 + \epsilon_{approx} \right) \xi^b,$$

where $(i)$ follows from Lemma 5 and eq. (28), $(ii)$ follows from Lemma 1 and eq. (27), in $(iii)$ we define

$$C_{b,\lambda} := B_{\phi}(1+\gamma) \left( \frac{\lambda}{|\chi-\gamma\lambda|} C_M + \frac{C_M(1-\lambda)}{|\gamma-\chi|(\xi-\gamma\lambda)} + 1 + \|f\|_1 + \left( \frac{\lambda+(1-\lambda)\|f\|_1}{1-\gamma\lambda} + \lambda \right) \right),$$

and $(iv)$ follows from Lemma 3.

## E.2 PROOF OF PROPOSITION 4

According to the definition of $\widehat{\mathcal{T}}_t^{\lambda}$, we have

$$\mathbb{E} \left[ \left\| \widehat{\mathcal{T}}_t^{\lambda}(\theta_t) \right\|_2^2 \Big| \mathcal{F}_{t-1} \right]$$

$$= \mathbb{E} \left[ (\rho_t^b)^2 \left( r_t^b + \gamma\theta_t^{\top}\phi_t^{b+1} - \theta_t^{\top}\phi_t^b \right)^2 (e_t^b)^{\top} e_t^b \Big| \mathcal{F}_{t-1} \right]$$

$$\overset{(i)}{=} \sum_{s \in \mathcal{S}, a \in \mathcal{A}, s' \in \mathcal{S}} \mathbb{P} \left( s_t^b = s, a_t^b = a, s_t^{b+1} = s' \big| \mathcal{F}_{t-1} \right)$$

$$\cdot \mathbb{E} \left[ (\rho_t^b)^2 \left( r_t^b + \gamma\theta_t^{\top}\phi_t^{b+1} - \theta_t^{\top}\phi_t^b \right)^2 (e_t^b)^{\top} e_t^b \Big| s_t^b = s, a_t^b = a, s_t^{b+1} = s', \mathcal{F}_{t-1} \right]$$

$$\overset{(ii)}{=} \sum_{s \in \mathcal{S}, a \in \mathcal{A}, s' \in \mathcal{S}} \mathbb{P} \left( s_t^b = s \big| \mathcal{F}_{t-1} \right) \mu(a|s) \mathsf{P}(s'|s,a)$$

$$\frac{\pi^2(a|s)}{\mu^2(a|s)} (r(s,a) + \gamma\theta_t^{\top}\phi(s') - \theta_t^{\top}\phi(s))^2 \mathbb{E} \left[ (e_t^b)^{\top} e_t^b \big| s_t^b = s, \mathcal{F}_{t-1} \right]$$

$$\overset{(iii)}{\leq} \sum_{s \in \mathcal{S}} \mathbb{P} \left( s_t^b = s \big| \mathcal{F}_{t-1} \right) \mathbb{E} \left[ (e_t^b)^{\top} e_t^b \big| s_t^b = s, \mathcal{F}_{t-1} \right]$$

$$\cdot \sum_{s' \in \mathcal{S}} P_{\mu,\pi}(s'|s)(r(s,a) + \gamma\phi(s')^{\top}\theta_t - \phi(s)^{\top}\theta_t)^2$$

$$\overset{(iv)}{\leq} \rho_{max} \left(4(1+\gamma^2)B_\phi^2 B_\theta^2 + 2r_{max}^2\right) B_\phi^2 \sum_{s\in\mathcal{S}} \mathbb{P}\left(s_t^b = s\middle|\mathcal{F}_{t-1}\right) \mathbb{E}\left[(e_t^b)^\top e_t^b \middle| s_t^b = s, \mathcal{F}_{t-1}\right],$$

$$(57)$$

where $(i)$ follows from the law of total probability, $(ii)$ follows from the Markov property and the fact that $e_t^b$ only depends on $(s_t^0, a_t^0, \ldots, s_t^b)$, and $(iii)$ follows from eq. (30) and the fact $\sum_{s'} P_{\mu,\pi}(s'|s) \leq \rho_{max}$.

Define $\Delta_b(s) = \mathbb{P}\left(s_t^b = s\middle|\mathcal{F}_{t-1}\right) \mathbb{E}\left[(e_t^b)^\top e_t^b\middle|s_t^b = s, \mathcal{F}_{t-1}\right]$. We then proceed to bound the term $\Delta_b(s)$. We have

$$\Delta_b(s)$$

$$\overset{(i)}{=} \mathbb{P}\left(s_t^b = s\middle|\mathcal{F}_{t-1}\right)$$
$$\sum_{\tilde{s}\in\mathcal{S},\tilde{a}\in\mathcal{A}} \mathbb{P}\left(s_t^{b-1} = \tilde{s}, a_t^{b-1} = \tilde{a}\middle|s_t^b = s, \mathcal{F}_{t-1}\right) \mathbb{E}\left[(e_t^b)^\top e_t^b\middle|s_t^{b-1} = \tilde{s}, a_t^{b-1} = \tilde{a}, s_t^b = s, \mathcal{F}_{t-1}\right]$$

$$\overset{(ii)}{=} \mathbb{P}\left(s_t^b = s\middle|\mathcal{F}_{t-1}\right)$$
$$\sum_{\tilde{s}\in\mathcal{S},\tilde{a}\in\mathcal{A}} \frac{\mathbb{P}\left(s_t^{b-1} = \tilde{s}\middle|\mathcal{F}_{t-1}\right)\mu(\tilde{a}|\tilde{s})\mathsf{P}(s|\tilde{s},\tilde{a})}{\mathbb{P}\left(s_t^b = s\middle|\mathcal{F}_{t-1}\right)} \mathbb{E}\left[(e_t^b)^\top e_t^b\middle|s_t^{b-1} = \tilde{s}, a_t^{b-1} = \tilde{a}, s_t^b = s, \mathcal{F}_{t-1}\right]$$

$$\overset{(iii)}{=} \sum_{\tilde{s}\in\mathcal{S},\tilde{a}\in\mathcal{A}} \mathbb{P}\left(s_t^{b-1} = \tilde{s}\middle|\mathcal{F}_{t-1}\right)\mu(\tilde{a}|\tilde{s})\mathsf{P}(s|\tilde{s},\tilde{a})$$

$$\cdot \mathbb{E}\left[\left(\gamma\lambda\rho_t^{b-1}e_t^{b-1} + \left(\lambda + (1-\lambda)\left(\gamma\rho_t^{b-1}F_t^{b-1} + 1\right)\right)\phi_t^b\right)^\top\right.$$
$$\left.\cdot\left(\gamma\lambda\rho_t^{b-1}e_t^{b-1} + \left(\lambda + (1-\lambda)\left(\gamma\rho_t^{b-1}F_t^{b-1} + 1\right)\right)\phi_t^b\right)\middle|s_t^{b-1} = \tilde{s}, a_t^{b-1} = \tilde{a}, s_t^b = s, \mathcal{F}_{t-1}\right]$$

$$= \sum_{\tilde{s}\in\mathcal{S},\tilde{a}\in\mathcal{A}} \mathbb{P}\left(s_t^{b-1} = \tilde{s}\middle|\mathcal{F}_{t-1}\right)\mu(\tilde{a}|\tilde{s})\mathsf{P}(s|\tilde{s},\tilde{a})$$

$$\mathbb{E}\left[(\gamma\lambda)^2(\rho_t^{b-1})^2(e_t^{b-1})^\top e_t^{b-1} + (1-\lambda)^2\gamma^2(\rho_t^{b-1}F_t^{b-1})^2\phi(s)^\top\phi(s) + \phi(s)^\top\phi(s)\right.$$
$$+2\gamma^2\lambda(1-\lambda)(\rho_t^{b-1})^2 F_t^{b-1}\phi(s)^\top e_t^{b-1} + 2\gamma\lambda\rho_t^{b-1}\phi(s)^\top e_t^{b-1}$$
$$\left.+2(1-\lambda)\gamma\rho_t^{b-1}F_t^{b-1}\phi(s)^\top\phi(s)\middle|s_t^{b-1} = \tilde{s}, a_t^{b-1} = \tilde{a}, s_t^b = s, \mathcal{F}_{t-1}\right]$$

$$= \mathbb{P}\left(s_t^b = s\middle|\mathcal{F}_{t-1}\right)\|\phi(s)\|_2^2$$
$$+ \phi^\top(s)\sum_{\tilde{s}\in\mathcal{S}} P_\pi(s|\tilde{s})\mathbb{P}\left(s_t^{b-1} = \tilde{s}\middle|\mathcal{F}_{t-1}\right)\mathbb{E}\left[2\gamma\lambda e_t^{b-1} + 2\gamma(1-\lambda)F_t^{b-1}\phi(s)\middle|s_t^{b-1} = \tilde{s}, \mathcal{F}_{t-1}\right]$$

$$+ \lambda^2\gamma^2\sum_{\tilde{s}\in\mathcal{S}} P_{\mu,\pi}(s|\tilde{s})\mathbb{P}\left(s_t^{b-1} = \tilde{s}\middle|\mathcal{F}_{t-1}\right)\mathbb{E}\left[(e_t^{b-1})^\top e_t^{b-1}\middle|s_t^{b-1} = \tilde{s}, \mathcal{F}_{t-1}\right]$$

$$+ 2\gamma^2\lambda(1-\lambda)\sum_{\tilde{s}\in\mathcal{S}} P_{\mu,\pi}(s|\tilde{s})\mathbb{P}\left(s_t^{b-1} = \tilde{s}\middle|\mathcal{F}_{t-1}\right)\mathbb{E}\left[F_t^{b-1}\phi(s)^\top e_t^{b-1}\middle|s_t^{b-1} = \tilde{s}, \mathcal{F}_{t-1}\right]$$

$$+ \|\phi(s)\|_2^2\gamma^2(1-\lambda)^2\sum_{\tilde{s}\in\mathcal{S}} P_{\mu,\pi}(s|\tilde{s})\mathbb{P}\left(s_t^{b-1} = \tilde{s}\middle|\mathcal{F}_{t-1}\right)\mathbb{E}\left[(F_t^{b-1})^2\middle|s_t^{b-1} = \tilde{s}, \mathcal{F}_{t-1}\right]$$

$$\leq B_\phi^2 d_{\mu,b}(s) + 2\gamma\lambda\phi^\top(s)(\beta_{b-1}P_\pi)_{(\cdot,s)} + 2\gamma(1-\lambda)B_\phi^2(P_\pi^\top f_{b-1})_s$$
$$+ \lambda^2\gamma^2(P_{\mu,\pi}^\top\Delta_{b-1})_s + \gamma^2(1-\lambda)^2 B_\phi^2(P_{\mu,\pi}^\top r_{b-1})_s$$
$$+ 2\gamma^2\lambda(1-\lambda)\sum_{\tilde{s}\in\mathcal{S}} P_{\mu,\pi}(s|\tilde{s})\mathbb{P}\left(s_t^{b-1} = \tilde{s}\middle|\mathcal{F}_{t-1}\right)\mathbb{E}\left[F_t^{b-1}\phi(s)^\top e_t^{b-1}\middle|s_t^{b-1} = \tilde{s}, \mathcal{F}_{t-1}\right]$$

$$\overset{(iv)}{=} B_\phi^2 d_{\mu,b}(s) + \lambda^2\gamma^2(P_{\mu,\pi}^\top\Delta_{b-1})_s + 2\gamma(1-\lambda)B_\phi^2(P_\pi^\top f_{b-1})_s + 2\gamma\lambda\phi^\top(s)(\beta_{b-1}P_\pi)_{(\cdot,s)}$$
$$+ \gamma^2(1-\lambda)^2 B_\phi^2(P_{\mu,\pi}^\top r_{b-1})_s + 2\gamma^2\lambda(1-\lambda)\phi(s)^\top(\delta_{b-1}P_{\mu,\pi})_{(\cdot,s)}, \tag{58}$$

where $(i)$ follows from the law of total probability, $(ii)$ follows from the Bayes rule, $(iii)$ follows the update rule of $e_t^b$ and in $(iv)$ we define

$$\delta_\tau(s) = \mathbb{P}\left(s_t^\tau = s\middle|\mathcal{F}_{t-1}\right)\mathbb{E}\left[F_t^\tau e_t^\tau\middle|s_t^\tau = s, \mathcal{F}_{t-1}\right].$$

Summing eq. (58) over $\mathcal{S}$ yields

$$\mathbf{1}^\top \Delta_b = \sum_s \Delta_b(s)$$

$$\leq B_\phi^2 \mathbf{1}^\top d_{\mu,b}(s) + \lambda^2 \gamma^2 \mathbf{1}^\top P_{\mu,\pi}^\top \Delta_{b-1} + 2\gamma(1-\lambda) B_\phi^2 \mathbf{1}^\top P_\pi^\top f_{b-1} + 2\gamma\lambda \text{trace}\left(\Phi\beta_{b-1} P_\pi\right)$$

$$+ \gamma^2(1-\lambda)^2 B_\phi^2 \mathbf{1}^\top P_{\mu,\pi}^\top r_{b-1} + 2\gamma^2\lambda(1-\lambda)\text{trace}\left(\Phi\delta_{b-1} P_{\mu,\pi}\right)$$

$$\overset{(i)}{\leq} \lambda^2\gamma^2\rho_{max} \mathbf{1}^\top \Delta_{b-1} + B_\phi^2 + 2\gamma(1-\lambda) B_\phi^2 \mathbf{1}^\top f_{b-1} + 2\gamma\lambda\text{trace}\left(\Phi\beta_{b-1} P_\pi\right)$$

$$+ \gamma^2(1-\lambda)^2 B_\phi^2 \rho_{max} \mathbf{1}^\top r_{b-1} + 2\gamma^2\lambda(1-\lambda)\text{trace}\left(\Phi\delta_{b-1} P_{\mu,\pi}\right),$$

where $(i)$ follows from Lemma 4 with $P = P_{\mu,\pi}$.

Recursively applying the above inequality, we have

$$\mathbf{1}^\top \Delta_b \leq (\lambda^2\gamma^2)^b \rho_{max}^b \mathbf{1}^\top \Delta_0 + \sum_{\tau=1}^b (\lambda^2\gamma^2)^{b-\tau} (\rho_{max})^{b-\tau} \left(B_\phi^2 + 2\gamma(1-\lambda) B_\phi^2 \mathbf{1}^\top f_{\tau-1}\right.$$

$$\left. + 2\gamma\lambda\text{trace}\left(\Phi\beta_{\tau-1} P_\pi\right) + \gamma^2(1-\lambda)^2 B_\phi^2 \rho_{max} \mathbf{1}^\top r_{\tau-1} + 2\gamma^2\lambda(1-\lambda)\text{trace}\left(\Phi\delta_{\tau-1} P_{\mu,\pi}\right)\right).$$
$$(59)$$

Substituting eq. (54) into $\text{trace}\left(\Phi\beta_\tau P_\pi\right)$ with $b$ and $\tau$ replaced by $\tau$ and $m$ respectively yields

$$\text{trace}\left(\Phi\beta_\tau P_\pi\right)$$

$$\overset{(i)}{=} (\gamma\lambda)^\tau \text{trace}\left(\Phi\beta_0 P_\pi^{\tau+1}\right)$$

$$+ \lambda \sum_{m=0}^{\tau-1} (\gamma\lambda)^m \left(\text{trace}\left(\Phi\Phi^\top D_{\mu,\tau-m} P_\pi^{m+1}\right) + (1-\lambda)\text{trace}\left(\Phi\Phi^\top F_{\tau-m} P_\pi^{m+1}\right)\right)$$

$$\overset{(ii)}{=} (\gamma\lambda)^\tau \text{trace}\left(P_\pi^{\tau+1}\Phi\Phi^\top D_{\mu,0}\right)$$

$$+ \lambda \sum_{m=0}^{\tau-1} (\gamma\lambda)^m \left(\text{trace}\left(P_\pi^{m+1}\Phi\Phi^\top D_{\mu,\tau-m}\right) + (1-\lambda)\text{trace}\left(P_\pi^{m+1}\Phi\Phi^\top F_{\tau-m}\right)\right)$$

$$\overset{(iii)}{\leq} (\gamma\lambda)^\tau B_\phi^2 \mathbf{1}^\top d_{\mu,0} + B_\phi^2 \sum_{m=0}^{\tau-1} (\gamma\lambda)^m \left(\lambda\|d_{\mu,\tau-m}\|_1 + (1-\lambda)\|f_{\tau-m}\|_1\right)$$

$$\overset{(iv)}{\leq} (\gamma\lambda)^\tau B_\phi^2 + \frac{1-(\gamma\lambda)^\tau}{1-\gamma\lambda}\left(\lambda + (1-\lambda)(1+\|f\|_1)\right) B_\phi^2$$

$$\leq \frac{B_\phi^2}{1-\gamma\lambda}\left(1 + (1-\lambda)\|f\|_1\right),$$
$$(60)$$

where $(i)$ follows from eq. (54), $(ii)$ follows from the facts that $\text{trace}(AB) = \text{trace}(BA)$ and $\delta_0 = \Phi^\top D_{\mu,0}$, $(iii)$ follows from Lemma 6 with $P = P_\pi$, and $(iv)$ follows from eq. (37).

Next, we proceed to bound the term $\text{trace}\left(\Phi^\top \delta_{\tau-1} P_{\mu,\pi}\right)$.

$$\delta_{b-1}(\tilde{s}) \overset{(i)}{=} \mathbb{P}\left(s_t^{b-1} = \tilde{s}\big|\mathcal{F}_{t-1}\right) \mathbb{E}\left[F_t^{b-1}\left(\gamma\lambda\rho_t^{b-2} e_t^{b-2} + (\lambda + (1-\lambda)F_t^{b-1})\phi(\tilde{s})\right)\big|s_t^{b-1} = \tilde{s}, \mathcal{F}_{t-1}\right]$$

$$\overset{(ii)}{=} (1-\lambda)\mathbb{P}\left(s_t^{b-1} = \tilde{s}\big|\mathcal{F}_{t-1}\right) \mathbb{E}\left[(F_t^{b-1})^2\big|s_t^{b-1} = \tilde{s}, \mathcal{F}_{t-1}\right]\phi(\tilde{s})$$

$$+ \lambda\mathbb{P}\left(s_t^{b-1} = \tilde{s}\big|\mathcal{F}_{t-1}\right) \mathbb{E}\left[F_t^{b-1}\big|s_t^{b-1} = \tilde{s}, \mathcal{F}_{t-1}\right]\phi(\tilde{s})$$

$$+ \gamma\lambda\mathbb{P}\left(s_t^{b-1} = \tilde{s}\big|\mathcal{F}_{t-1}\right) \mathbb{E}\left[\rho_t^{b-2} e_t^{b-2}\big|s_t^{b-1} = \tilde{s}, \mathcal{F}_{t-1}\right]\phi(\tilde{s})$$

$$+ \gamma^2\lambda\mathbb{P}\left(s_t^{b-1} = \tilde{s}\big|\mathcal{F}_{t-1}\right) \mathbb{E}\left[(\rho_t^{b-2})^2 F_t^{b-2} e_t^{b-2}\big|s_t^{b-1} = \tilde{s}, \mathcal{F}_{t-1}\right]\phi(\tilde{s})$$

$$\overset{(iii)}{=} (1-\lambda)r_{b-1}(\tilde{s})\phi(\tilde{s}) + \lambda f_{b-1}(\tilde{s})\phi(\tilde{s})$$

$$+ \mathbb{P}\left(s_t^{b-1} = \tilde{s}\big|\mathcal{F}_{t-1}\right) \sum_{s'',a''} \mathbb{P}\left(s_t^{b-2} = s'', a_t^{b-2} = a''\big|s_t^{b-1} = \tilde{s}, \mathcal{F}_{t-1}\right)$$

$$\cdot \mathbb{E}\left[\gamma\lambda\rho_t^{b-2}e_t^{b-2} + \gamma^2\lambda(\rho_t^{b-2})^2 F_t^{b-2}e_t^{b-2}\big|s_t^{b-2}=s'', a_t^{b-2}=a'', s_t^{b-1}=\tilde{s}, \mathcal{F}_{t-1}\right]$$

$$= (1-\lambda)r_{b-1}(\tilde{s})\phi(\tilde{s}) + \lambda f_{b-1}(\tilde{s})\phi(\tilde{s})$$

$$+ \mathbb{P}\left(s_t^{b-1}=\tilde{s}\big|\mathcal{F}_{t-1}\right)\sum_{s'',a''}\frac{\mathbb{P}\left(s_t^{b-2}=s''\big|\mathcal{F}_{t-1}\right)\mu(a''|s'')\mathsf{P}(\tilde{s}|s'',a'')}{\mathbb{P}\left(s_t^{b-1}=\tilde{s}\big|\mathcal{F}_{t-1}\right)}$$

$$\cdot \mathbb{E}\left[\gamma\lambda\frac{\pi(a''|s'')}{\mu(a''|s'')}e_t^{b-2} + \gamma^2\lambda\frac{\pi^2(a''|s'')}{\mu^2(a''|s'')}F_t^{b-2}e_t^{b-2}\bigg|s_t^{b-2}=s'', \mathcal{F}_{t-1}\right]$$

$$= (1-\lambda)r_{b-1}(\tilde{s})\phi(\tilde{s}) + \lambda f_{b-1}(\tilde{s})\phi(\tilde{s}) + \gamma\lambda(\beta_{b-2}P_\pi)_{(\cdot,s)} + \gamma^2\lambda(\delta_{b-2}P_{\mu,\pi})_{(\cdot,s)},$$

where $(i)$ follows from the update of $e_t^{b-1}$, $(ii)$ follows from the update rule of $F_t^{b-1}$, and $(iii)$ follows from the law of total probability.

The above equality implies that

$$\delta_{b-1} = (1-\lambda)\Phi^\top\mathrm{diag}(r_{b-1}) + \lambda\Phi^\top\mathrm{diag}(f_{b-1}) + \gamma\lambda\beta_{b-2}P_\pi + \gamma^2\lambda\delta_{b-2}P_{\pi,\mu}.$$

Recursively applying the above equality yields

$$\delta_{b-1} = \sum_{m=1}^{b-1}(\Phi^\top((1-\lambda)\mathrm{diag}(r_m) + \lambda\mathrm{diag}(f_m)) + \gamma\lambda\beta_{m-1}P_\pi)(P_{\pi,\mu})^{b-1-m}$$

$$+ (\gamma^2\lambda)^{b-1}\delta_0(P_{\pi,\mu})^{b-1}.$$

Note that the above inequality holds for any fixed $b >= 2$. As a result, by changing of notation, for all $\tau \geq 1$, we have

$$\delta_\tau = \sum_{m=1}^{\tau}(\Phi^\top((1-\lambda)\mathrm{diag}(r_m) + \lambda\mathrm{diag}(f_m)) + \gamma\lambda\beta_{m-1}P_\pi)(P_{\pi,\mu})^{\tau-m} + (\gamma^2\lambda)^\tau\delta_0(P_{\pi,\mu})^\tau.$$

$$(61)$$

Substituting eq. (61) into $\mathrm{trace}(\Phi\delta_\tau P_{\mu,\pi})$, we have

$$\mathrm{trace}(\Phi\delta_\tau P_{\mu,\pi})$$

$$= \mathrm{trace}\Bigg(\Phi\Bigg(\sum_{m=1}^{\tau}(\Phi^\top((1-\lambda)\mathrm{diag}(r_m) + \lambda\mathrm{diag}(f_m)) + \gamma\lambda\beta_{m-1}P_\pi)(P_{\pi,\mu})^{\tau-m}$$

$$+ (\gamma^2\lambda)^\tau\delta_0(P_{\pi,\mu})^\tau\Bigg)P_{\mu,\pi}\Bigg)$$

$$\stackrel{(i)}{=} \sum_{m=1}^{\tau}\mathrm{trace}\left((P_{\pi,\mu})^{\tau-m+1}\left(\Phi\Phi^\top((1-\lambda)\mathrm{diag}(r_m) + \lambda\mathrm{diag}(f_m)) + \gamma\lambda\Phi\beta_{m-1}P_\pi\right)\right)$$

$$+ \mathrm{trace}\left((\gamma^2\lambda)^\tau(P_{\pi,\mu})^{\tau+1}\Phi\delta_0\right),$$

$$(62)$$

where $(i)$ follows from the fact that $\mathrm{trace}(AB) = \mathrm{trace}(BA)$ and $\mathrm{trace}(A+B) = \mathrm{trace}(A) + \mathrm{trace}(B)$.

Applying Lemma 6 with $Q = P_{\mu,\pi}$, $C = \rho_{max}$, $P = P_\pi$ and $D = (1-\lambda)\mathrm{diag}(r_m)$, we have

$$\mathrm{trace}\left((P_{\pi,\mu})^{\tau-m+1}\Phi\Phi^\top(1-\lambda)\mathrm{diag}(r_m)\right) \leq (1-\lambda)\rho_{max}^{\tau-m+1}B_\phi^2\|r_m\|_1.$$

$$(63)$$

Applying Lemma 6 with $Q = P_{\mu,\pi}$, $C = \rho_{max}$, $P = P_\pi$ and $D = \lambda\mathrm{diag}(f_m)$, we have

$$\mathrm{trace}\left((P_{\pi,\mu})^{\tau+1-m}\Phi\Phi^\top\lambda\mathrm{diag}(f_m)\right) \leq \lambda\rho_{max}^{\tau+1-m}B_\phi^2\|f_m\|_1 \stackrel{(i)}{\leq} \lambda\rho_{max}^{\tau+1-m}B_\phi^2(1 + \|f\|_1), \quad (64)$$

where $(i)$ follows from eq. (37).

For the term $\mathrm{trace}\left((P_{\pi,\mu})^{\tau+1-m}\Phi\beta_{m-1}P_\pi\right)$, we have

$$\mathrm{trace}\left((P_{\pi,\mu})^{\tau+1-m}\Phi\beta_{m-1}P_\pi\right)$$

$$\overset{(i)}{=} (\gamma\lambda)^{m-1}\text{trace}\left(P_\pi^m (P_{\pi,\mu})^{\tau+1-m}\Phi\Phi^\top D_{\mu,0}\right)$$

$$+ \sum_{l=0}^{m-2}(\gamma\lambda)^l\left(\lambda\text{trace}\left(P_\pi^{l+1}(P_{\pi,\mu})^{\tau+1-m}\Phi\Phi^\top D_{\mu,m-1-l}\right)\right.$$

$$\left.+(1-\lambda)\text{trace}\left(P_\pi^{l+1}(P_{\pi,\mu})^{\tau-m+1}\Phi\Phi^\top F_{\pi,m-1-l}\right)\right)$$

$$\overset{(ii)}{\leq} B_\phi^2\rho_{max}^{\tau+1-m}\left((\gamma\lambda)^{m-1}\mathbf{1}^\top d_{\mu,0} + \sum_{l=0}^{m-2}(\gamma\lambda)^l\left(\lambda\mathbf{1}^\top d_{\mu,m-1-l}+(1-\lambda)\mathbf{1}^\top f_{m-1-l}\right)\right)$$

$$\overset{(iii)}{\leq} \frac{B_\phi^2\rho_{max}^{\tau+1-m}}{1-\gamma\lambda}\left(1+(1-\lambda)\|f\|_1\right), \tag{65}$$

where $(i)$ follows from eq. (54) and the facts that $\text{trace}(A+B) = \text{trace}(A)+\text{trace}(B)$ and $\text{trace}(AB) = \text{trace}(BA)$, $(ii)$ follows from Lemma 6 with $Q = P_{\mu,\pi}$, $P = P_\pi$, and $D = F_{m-1-l}$ and $D_{\pi,m-1-l}$ respectively, and $(iii)$ follow from the eq. (37).

Recall $\delta_0 = \Phi^\top D_{\mu,0}$. Applying Lemma 6 with $Q = P_{\mu,\pi}$ and $D = D_{\mu,0}$ yields

$$\text{trace}\left((\gamma^2\lambda)^\tau (P_{\pi,\mu})^{\tau+1}\Phi\delta_0\right) \leq (\gamma^2\lambda)^\tau B_\phi^2\rho_{max}^{\tau+1}. \tag{66}$$

Substituting eqs. (63) to (66) into eq. (62), we have

$$\text{trace}\left(\Phi\delta_\tau P_{\mu,\pi}\right)$$

$$\leq (\gamma^2\lambda)^\tau B_\phi^2\rho_{max}^{\tau+1}$$

$$+ \sum_{m=1}^\tau B_\phi^2\rho_{max}^{\tau-m+1}\left(\lambda(\|f\|_1+1)+\frac{1+(1-\lambda)\|f\|_1}{1-\gamma\lambda}\right) + \sum_{m=1}^\tau (1-\lambda)\rho_{max}^{\tau-m+1}B_\phi^2\|r_m\|_1. \tag{67}$$

Substituting eqs. (60) and (67) into eq. (59), we have

$$\mathbf{1}^\top\Delta_b \leq \lambda^{2b}\gamma^{2b}\rho_{max}^b B_\phi^2 + (1+2\gamma(1-\lambda)(1+\|f\|_1)+2\gamma\lambda C_\beta)B_\phi^2\sum_{\tau=1}^b\lambda^{2(b-\tau)}\gamma^{2(b-\tau)}\rho_{max}^{b-\tau}$$

$$+\gamma^2(1-\lambda)^2\rho_{max}^{b+1}B_\phi^2\sum_{\tau=1}^b\lambda^{2b-2\tau}\gamma^{2b-2\tau}\rho_{max}^{-\tau}\|r_{\tau-1}\|_1 + 2\lambda^b\gamma^{2b}(1-\lambda)\rho_{max}^b B_\phi^2\sum_{\tau=1}^b\lambda^{b-\tau}$$

$$+2\gamma^2\lambda(1-\lambda)B_\phi^2\left(\lambda\|f\|_1+1+\frac{1+(1-\lambda)\|f\|_1}{1-\gamma\lambda}\right)\rho_{max}^b\sum_{\tau=1}^b\gamma^{2b-2\tau}\lambda^{2b-2\tau}\sum_{m=1}^{\tau-1}\rho_{max}^{-m}$$

$$+2\gamma^2\lambda(1-\lambda)^2\rho_{max}^b B_\phi^2\sum_{\tau=1}^b\gamma^{2b-2\tau}\lambda^{2b-2\tau}\sum_{m=1}^{\tau-1}\rho_{max}^{-m}\|r_m\|_1, \tag{68}$$

where we let $C_\beta := \frac{B_\phi^2}{1-\gamma\lambda}\left(1+(1-\lambda)\|f\|_1\right)$.

Under different conditions of $\rho_{max}$, the term $\mathbf{1}^\top\Delta_b$ and $\mathbb{E}\left[\left\|\widehat{\mathcal{T}}_t^\lambda(\theta_t)\right\|_2^2 \Big| \mathcal{F}_{t-1}\right]$ can be upper bounded differently as following:

$(a)$. $\gamma^2\rho_{max} < 1$, substituting eq. (40) into eq. (68) yields

$$\mathbf{1}^\top\Delta_b \leq \lambda^{2b}\gamma^{2b}\rho_{max}^b B_\phi^2 + (1+2\gamma(1-\lambda)(1+\|f\|_1)+2\gamma\lambda C_\beta)B_\phi^2\sum_{\tau=1}^b\lambda^{2(b-\tau)}\gamma^{2(b-\tau)}\rho_{max}^{b-\tau}$$

$$+\gamma^2(1-\lambda)^2\rho_{max}^{b+1}B_\phi^2\sum_{\tau=1}^b\lambda^{2b-2\tau}\gamma^{2b-2\tau}\rho_{max}^{-\tau}\left(\frac{3+2\gamma\|f\|_1}{1-\gamma^2}+1\right)$$

$$+2\lambda^b\gamma^{2b}(1-\lambda)\rho_{max}^b B_\phi^2\sum_{\tau=1}^b\lambda^{b-\tau}$$

$$+ 2\gamma^2\lambda(1-\lambda)B_\phi^2 \left( \lambda\|f\|_1 + 1 + \frac{1+(1-\lambda)\|f\|_1}{1-\gamma\lambda} \right) \rho_{max}^b$$

$$\cdot \sum_{\tau=1}^{b} \gamma^{2b-2\tau}\lambda^{2b-2\tau} \sum_{m=1}^{\tau-1} \rho_{max}^{-m}$$

$$+ 2\gamma^2\lambda(1-\lambda)^2 \rho_{max}^b B_\phi^2 \sum_{\tau=1}^{b} \gamma^{2b-2\tau}\lambda^{2b-2\tau} \sum_{m=1}^{\tau-1} \rho_{max}^{-m} \left( \frac{3+2\gamma\|f\|_1}{1-\gamma^2} + 1 \right)$$

$$\leq \lambda^{2b}\gamma^{2b}\rho_{max}^b B_\phi^2 + \frac{(1+2\gamma(1-\lambda)(1+\|f\|_1)+2\gamma\lambda C_\beta)B_\phi^2}{1-\gamma^2\rho_{max}\lambda}$$

$$+ \frac{\gamma^2(1-\lambda)^2\rho_{max}B_\phi^2}{1-\gamma^2\rho_{max}\lambda}\left( \frac{3+2\gamma\|f\|_1}{1-\gamma^2} + 1 \right) + 2\lambda^{2b}\gamma^b\rho_{max}^b B_\phi^2$$

$$+ \frac{2\gamma^2\lambda(1-\lambda)B_\phi^2}{(1-\gamma^2\lambda^2)(1-\rho_{max}^{-1})}\left( \lambda\|f\|_1 + 1 + \frac{1+(1-\lambda)\|f\|_1}{1-\gamma\lambda} \right)\rho_{max}^{b+1}$$

$$+ \frac{2\gamma^2\lambda(1-\lambda)^2 B_\phi^2}{(1-\gamma^2\lambda^2)(1-\rho_{max}^{-1})}\left( \frac{3+2\gamma\|f\|_1}{1-\gamma^2} + 1 \right)\rho_{max}^{b+1}, \tag{69}$$

where the last two terms of the above inequality are of the order $\mathcal{O}\left(\rho_{max}^b\right)$. Therefore, we have $\mathbf{1}^\top\Delta_b \leq C\rho_{max}^b$ for some $C > 0$. Substituting eq. (69) into eq. (57) yields

$$\mathbb{E}\left[ \left\| \widehat{\mathcal{T}}_t^\lambda(\theta_t) \right\|_2^2 \Big| \mathcal{F}_{t-1} \right] \leq C_{\sigma,\lambda,1}\rho_{max}^b,$$

where $C_{\sigma,\lambda,1} > 0$ is a constant and is determined by eq. (69).

$(b)$. $\gamma^2\rho_{max} = 1$, substituting eq. (39) into eq. (68) yields

$$\mathbf{1}^\top\Delta_b \leq \lambda^{2b}B_\phi^2 + (1+2\gamma(1-\lambda)(1+\|f\|_1)+2\gamma\lambda C_\beta)B_\phi^2 \sum_{\tau=1}^{b} \lambda^{2(b-\tau)}$$

$$+ \gamma^2(1-\lambda)^2\rho_{max}B_\phi^2 \sum_{\tau=1}^{b} \lambda^{2b-2\tau}(4+2\gamma\|f\|_1)\tau + 2\lambda^b(1-\lambda)B_\phi^2 \sum_{\tau=1}^{b} \lambda^{b-\tau}$$

$$+ 2\gamma^2\lambda(1-\lambda)B_\phi^2 \left( \lambda\|f\|_1 + 1 + \frac{1+(1-\lambda)\|f\|_1}{1-\gamma\lambda} \right)\rho_{max}^b \sum_{\tau=1}^{b} \gamma^{2b-2\tau}\lambda^{2b-2\tau} \sum_{m=1}^{\tau} \rho_{max}^{-m}$$

$$+ 2\gamma^2\lambda(1-\lambda)^2(4+2\gamma\|f\|_1)B_\phi^2 \sum_{\tau=1}^{b} \lambda^{2b-2\tau} \sum_{m=1}^{\tau} \rho_{max}^{\tau-m}m$$

$$\leq \lambda^{2b}B_\phi^2 + \frac{(1+2\gamma(1-\lambda)(1+\|f\|_1)+2\gamma\lambda C_\beta)B_\phi^2}{1-\lambda^2} + 2\lambda^b B_\phi^2$$

$$+ \left( \frac{\gamma^2(1-\lambda)^2\rho_{max}B_\phi^2}{1-\lambda^2}(4+2\gamma\|f\|_1) + \frac{2\gamma^2\lambda(1-\lambda)^2(4+2\gamma\|f\|_1)B_\phi^2}{(1-\lambda^2)(1-\rho_{max}^{-1})} \right) b$$

$$+ \frac{2\gamma^2\lambda(1-\lambda)B_\phi^2}{(1-\gamma^2\lambda^2)(1-\rho_{max}^{-1})}\left( \lambda\|f\|_1 + 1 + \frac{1+(1-\lambda)\|f\|_1}{1-\gamma\lambda} \right)\rho_{max}^{b+1}, \tag{70}$$

where the last term of the above inequality is of the order $\mathcal{O}\left(\rho_{max}^b\right)$. Therefore, we have $\mathbf{1}^\top\Delta_b \leq C\rho_{max}^b$ for some $C > 0$. Substituting eq. (70) into eq. (57) yields

$$\mathbb{E}\left[ \left\| \widehat{\mathcal{T}}_t^\lambda(\theta_t) \right\|_2^2 \Big| \mathcal{F}_{t-1} \right] \leq C_{\sigma,\lambda,2}\rho_{max}^b,$$

where $C_{\sigma,\lambda,2} > 0$ is a constant and is determined by eq. (70).

$(c)$. $\gamma^2\rho_{max} > 1$, substituting eq. (38) into eq. (68) yields

$$\mathbf{1}^\top\Delta_b \leq \lambda^{2b}\gamma^{2b}\rho_{max}^b B_\phi^2 + (1+2\gamma(1-\lambda)(1+\|f\|_1)+2\gamma\lambda C_\beta)B_\phi^2 \sum_{\tau=1}^{b} \lambda^{2(b-\tau)}\gamma^{2(b-\tau)}\rho_{max}^{b-\tau}$$

$$+ \gamma^{2b+2}(1-\lambda)^2 \rho_{max}^{b+1} B_\phi^2 \left( \frac{3 + 2\gamma \|f\|_1}{\gamma^2 \rho_{max} - 1} + 1 \right) \sum_{\tau=1}^{b} \lambda^{2b-2\tau} + 2\lambda^b \gamma^{2b}(1-\lambda)\rho_{max}^b B_\phi^2 \sum_{\tau=1}^{b} \lambda^{b-\tau}$$

$$+ 2\gamma^2 \lambda(1-\lambda) B_\phi^2 \left( \lambda \|f\|_1 + 1 + \frac{1 + (1-\lambda)\|f\|_1}{1 - \gamma\lambda} \right) \rho_{max}^b \sum_{\tau=1}^{b} \gamma^{2b-2\tau} \lambda^{2b-2\tau} \sum_{m=1}^{\tau-1} \rho_{max}^{-m}$$

$$+ 2\gamma^2 \lambda(1-\lambda)^2 \rho_{max}^b B_\phi^2 \left( \frac{3 + 2\gamma \|f\|_1}{\gamma^2 \rho_{max} - 1} + 1 \right) \sum_{\tau=1}^{b} \gamma^{2b-2\tau} \lambda^{2b-2\tau} \sum_{m=1}^{\tau-1} \gamma^{2m}$$

$$\leq \lambda^{2b} \gamma^{2b} \rho_{max}^b B_\phi^2 + \frac{(1 + 2\gamma(1-\lambda)(1 + \|f\|_1) + 2\gamma\lambda C_\beta) B_\phi^2}{\gamma^2 \rho_{max} - 1} \cdot \gamma^{2b} \rho_{max}^b$$

$$+ \frac{\gamma^{2b+2}(1-\lambda)^2 \rho_{max}^{b+1} B_\phi^2}{1 - \lambda^2} \left( \frac{3 + 2\gamma \|f\|_1}{\gamma^2 \rho_{max} - 1} + 1 \right) + 2\lambda^b \gamma^{2b} \rho_{max}^b B_\phi^2$$

$$+ \frac{2\gamma^2 \lambda(1-\lambda) B_\phi^2}{(1 - \gamma^2 \lambda^2)(1 - \rho_{max}^{-1})} \left( \lambda \|f\|_1 + 1 + \frac{1 + (1-\lambda)\|f\|_1}{1 - \gamma\lambda} \right) \rho_{max}^{b+1}$$

$$+ \frac{2\gamma^2 \lambda(1-\lambda)^2}{(1 - \gamma^2)(1 - \gamma^2 \lambda^2)} B_\phi^2 \left( \frac{3 + 2\gamma \|f\|_1}{\gamma^2 \rho_{max} - 1} + 1 \right) \rho_{max}^b, \tag{71}$$

where the last term of the above inequality is of the order $\mathcal{O}\left( \rho_{max}^b \right)$. Therefore, we have $\mathbf{1}^\top \Delta_b \leq C\rho_{max}^b$ for some $C > 0$. Substituting eq. (71) into eq. (57) yields

$$\mathbb{E}\left[ \left\| \widehat{\mathcal{T}}_t^\lambda(\theta_t) \right\|_2^2 \Big| \mathcal{F}_{t-1} \right] \leq C_{\sigma,\lambda,3} \rho_{max}^b,$$

where $C_{\sigma,\lambda,3} > 0$ is a constant and is determined by eq. (71).

To summarize, the variance term $\mathbb{E}\left[ \left\| \widehat{\mathcal{T}}_t(\theta_t) \right\|_2^2 \Big| \mathcal{F}_{t-1} \right]$ can be bounded by $\sigma_\lambda^2 = \mathcal{O}(\rho_{max}^b)$.

### E.3  PROOF OF THEOREM 2

**Theorem 4** (Formal Statement of Theorem 2). *Suppose Assumptions 1 and 2 hold. Consider PER-ETD($\lambda$) specified in Algorithm 2. Let the stepsize $\eta_t = \frac{2}{\mu_\lambda(t+t_\lambda)}$, $t_\lambda = \frac{8L_\lambda^2}{\mu_\lambda^2}$, where $\mu_\lambda$ is defined in Lemma 7 and $L_\lambda$ is defined in Lemma 8 in Appendix C. Further let*

$$b = \max\left\{ \left\lceil \frac{\log(\mu_\lambda) - \log(5C_{b,\lambda} B_\phi)}{\log(\xi)} \right\rceil, \frac{\log(T)}{\log(\rho_{max}) + \log(1/\xi)} \right\},$$

*where $C_{b,\lambda}$ is a constant defined in the proof of Proposition 3 in Appendix E.1, $B_\phi := \max_{s \in \mathcal{S}} \|\phi(s)\|_2$, and $\xi := \max\{\gamma, \chi\}$. Let the projection set $\Theta = \left\{ \theta \in \mathbb{R}^d : \|\theta\|_2 \leq B_\theta \right\}$, where $B_\theta = \frac{\|\Phi^\top\|_2 r_{max}}{(1-\gamma)\mu_\lambda}$ (which implies $\theta_\lambda^* \in \Theta$). Then the output $\theta_T$ of PER-ETD($\lambda$) satisfies*

$$\mathbb{E}\left[ \|\theta_T - \theta_\lambda^*\|_2^2 \right] \leq \mathcal{O}\left( \frac{1}{T^{a_\lambda}} \right),$$

*where $a_\lambda = \frac{1}{\log_{1/\xi}(\rho_{max})+1}$. Further, PER-ETD($\lambda$) attains an $\epsilon$-accurate solution with $\tilde{\mathcal{O}}\left( \frac{1}{\epsilon^{1/a_\lambda}} \right)$ samples.*

*Proof.* The proof follows the same steps as Theorem 1 by replacing the terms $\widehat{\mathcal{T}}_t$, $\mathcal{T}$, $t_0$, $L_0$, $\mu_0$, $C_b$, $\sigma^2$ and $\theta^*$ with $\widehat{\mathcal{T}}_t^\lambda$, $\mathcal{T}^\lambda$, $t_\lambda$, $L_\lambda$, $\mu_\lambda$, $C_{b,\lambda}$, $\sigma_\lambda^2$ and $\theta_\lambda^*$ respectively. Specifically, Propositions 3 and 4 are applied to bound the bias and variance over the steps similarly to eq. (44). We then have the convergence as follows.

$$\mathbb{E}\left[ \|\theta_T - \theta_\lambda^*\|_2^2 \right] \leq \mathcal{O}\left( \frac{\|\theta_0 - \theta_\lambda^*\|_2^2}{T^2} \right) + \mathcal{O}\left( \frac{\sigma_\lambda^2}{T} \right) + \mathcal{O}\left( \frac{\xi^{2b}}{T} \right) + \mathcal{O}\left( \xi^b \right)$$

$$= \mathcal{O}\left( \frac{\|\theta_0 - \theta_\lambda^*\|_2^2}{T^2} \right) + \mathcal{O}\left( \frac{\rho_{max}^b}{T} \right) + \mathcal{O}\left( \frac{\xi^{2b}}{T} \right) + \mathcal{O}\left( \xi^b \right). \tag{72}$$

We further specify $b = \left\{ \left\lceil \frac{\log(\mu_0) - \log(5 C_b B_\phi)}{\log(\xi)} \right\rceil, \frac{\log(T)}{\log(\rho_{max}) + \log(1/\xi)} \right\}$. Then Equation (72) yields

$$\mathbb{E}\left[\|\theta_T - \theta_\lambda^*\|_2^2\right] \leq \mathcal{O}\left(\frac{\|\theta_0 - \theta_\lambda^*\|_2^2}{T^2}\right) + \mathcal{O}\left(\frac{T^{1-a_\lambda}}{T}\right) + \mathcal{O}\left(\frac{1}{T^{1+a_\lambda}}\right) + \mathcal{O}\left(\frac{1}{T^{a_\lambda}}\right) = \mathcal{O}\left(\frac{1}{T^{a_\lambda}}\right).$$

$\square$

