# OpenReview forum: "PER-ETD: A Polynomially Efficient Emphatic Temporal Difference Learning Method"
_ICLR.cc/2022/Conference — ICLR 2022 Poster_

### Official Review · Reviewer_M48z · 2021-11-02

**Correctness:** 4
**Technical Novelty And Significance:** 3
**Empirical Novelty And Significance:** 3
**Recommendation:** 8
**Confidence:** 3

**Main Review:**

The paper addresses a fundamental problem in reinforcement learning, and proposes a simple solution to this problem, with theoretical guarantees. The paper discusses the issues with off-policy evaluation, specifically the problem of distribution correction, for which emphatic algorithms have been introduced. It proposes that previous methods like GTD deal with this poorly, due to this distribution mismatch, though it is not clear why that is the case, or is it specified. The paper would benefit from a clear explanation on what objective function these class of methods optimize and how that differs, and how one is preferable to the other, since both these class of methods have been algorithms that can deal with off-policy learning. Page 3 -> "cardinality of the state", maybe you mean "states" or "state space"? The vector of values for all the states has the same notation as the value function, and capitalized like random variables, which is a bit confusing. $B_\phi$ is not introduced. Page 5, "helps to stabilize" -> "helps stabilize". Lemma 1 (geometric ergodicity) -- can you explain what this lemma says in words, it is not clear why this will be useful or you need it in your analysis. There is no row space above "Proposition 1". There are a lot of constants introduced that it's easy to lose sight of what they mean and get confused, if ever they were introduced at the beginning of the paper. One way to make the paper more clear would be to hide all these details and explain high level the steps of the analysis or proof. In proposition 1, $\zeta$ should be $\zeta^b$ everywhere? Is $\rho_max$ defined anywhere? What is $T_a$? The explanations below Theorem 1 are confusing because I'm not sure how to interpret $a$, and I don't understand where the threshold comes from. Below proposition 4., should that be "PER-ETD" instead of "ETD" ?  Above "Experiments", in the "impact of $\lambda$ on error bound, do you mean "eligibile traces" or "the eligible trace" maybe? Figure 3. what is the space in which these are plotted? The first sub-figure is also missing the y-axis. How confident are we that these methods are scalable in the general spaces we use these methods in? Is there such a way of increasing the period still applicable?

**Summary Of The Paper:**

This paper presents a technique of reducing the variance of emphatic algorithms by resetting the trace periodically, and that increasing the period logarithmicaly can result in an optimal way of trading off bias and variance of the resulting learned value function. The authors prove such a method obtains the optimal tradeoff and results in convergence in settings where predecessor variants fails (TD and ETD). The paper proves these claims theoretically, and show an illustration on the Baird domain for empirical support.

**Summary Of The Review:**

The paper addresses a fundamental problem in RL is a simple way and proves that such a method achieves an optimal tradeoff between bias and variance in emphatic algorithms, leading to polynomial sample complexity. The claims of the paper are also illustrated in a toy domain, showing great potential. The paper could benefit from some more explanations and clarity.

---

> ### Author Response · Authors · 2021-11-20
> **Authors' response (part 2)**
>
> Q9: Is $\rho_{max}$ defined anywhere?
>
> A9: Thank you for pointing it out! $\rho_{max}$ denotes the maximum of the distribution mismatch, i.e., $\rho_{max}:=\max_{s,a}\frac{\pi(a|s)}{\mu(a|s)}$. We have added the definition in the revision in Section 2.4.
>
> Q10: What is $T^a$? How to interpret $a$? Where does the threshold come from?
>
> A10: $T^a$ means $T$ to the $a$-th power. Here, the constant $a$ captures the polynomial convergence rate. The threshold arises as follows. The update of the emphatic weight $F_t^\tau$ depends on the discount factor $\gamma$ and the distribution mismatch ratio $\rho_{max}$. For mild mismatch, such an update satisfies the contraction property, thus yielding bounded emphatic weights. The threshold arises when the mismatch ratio $\rho_{max}$ becomes large enough so that the contraction no longer holds, which then yields a slower convergence rate.
>
> Q11:Below Proposition 4, should that be “PER-ETD” instead of “ETD”?
>
> A11: Yes, thank you for pointing this out. We made the change in the revision.
>
> Q12: Above "Experiments", in the "impact of $\lambda$” on error bound, do you mean "eligible traces" or "the eligible trace" maybe?
>
> A12: Thank you for the question. We meant “impact of the eligible trace (via the parameter $\lambda$)” because by introducing eligible traces the performance of PER-ETD is affected by the parameter $\lambda$. We clarified this in the paper.
>
> Q13:Figure 3, what is the space in which these are plotted? Does the first plot miss the y-axis?
>
> A13: Good question! They are plotted in the space that the parameter $\theta$ lies in. The first plot is in 1-dimensional Euclidean space, i.e., $\mathbb{R}^1$ (and this is why we did not mark y-axis). The second and third plots are in two-dimensional Euclidean space, i.e., $\mathbb{R}^2$. We have explained this in the caption of Figure 3 in the revision.
>
> Q14: How confident are we that these methods are scalable in the general spaces we use these methods in? Is there such a way of increasing the period still applicable?
>
> A14: The two algorithms PER-ETD(0) and PER-ETD($\lambda$) involve only vector addition and inner-product computations and hence should scale well with larger state and action spaces. Increasing the period length will cause large variance. Then we need to decrease the learning rate to reduce such a variance. In such a case, the training will take a longer time.

---

> > ### Comment · Reviewer_M48z · 2021-11-30
> > **Thanks for the response.**
> >
> > Thanks for the response. Explanations and comments make sense to me.

---

> ### Author Response · Authors · 2021-11-20
> **Authors’ response (part 1)**
>
> Many thanks for providing the review!  In our revision of the paper, we added new experiments in Appendix B, and made various revisions throughout the paper based on all reviewers’ comments. All our changes are highlighted with blue-colored texts. New comments on these changes are very welcome!
>
> Q1: The paper discusses the issues with off-policy evaluation, specifically the problem of distribution correction, for which emphatic algorithms have been introduced. It proposes that previous methods like GTD deal with this poorly, due to this distribution mismatch, though it is not clear why that is the case, or is it specified. The paper would benefit from a clear explanation on what objective function these class of methods optimize and how that differs, and how one is preferable to the other, since both these class of methods have been algorithms that can deal with off-policy learning.
>
> A1: Many thanks for the suggestion. Regarding GTD, it adjusts only the distribution mismatch of the action and does not adjust the distribution mismatch of the state. In this case, the state distribution mismatch can still cause unbounded error (with respect to ground-truth value function) as shown in (Kolter, 2011). Regarding ETD, it adjusts both state and action distribution mismatch. As a result, the error (with respect to the ground truth value function) is guaranteed to be bounded and decreases as the expressive power of the linear function class increases as shown in (Hallak et al., 2016). But ETD suffers from a large variance and may not always converge. Our algorithm PER-ETD enjoys the advantage of ETD (smaller convergence error) and overcomes the large variance issue, and hence can converge with polynomial sample complexity. We have added this explanation to the introduction of the paper.
>
> (Kolter, 2011): Kolter, J. "The fixed points of off-policy TD." Advances in Neural Information Processing Systems 24 (2011): 2169-2177.
> (Hallak et al., 2016): Hallak, Assaf, Aviv Tamar, Rémi Munos, and Shie Mannor. "Generalized emphatic temporal difference learning: Bias-variance analysis." In Thirtieth AAAI Conference on Artificial Intelligence. 2016.
>
> Q2: Page 3 -> "cardinality of the state", maybe you mean "states" or "state space"?
>
> A2: Thank you for pointing this out. Here we mean the cardinality of the state space. It has been revised in the paper.
>
> Q3: The vector of values for all the states has the same notation as the value function, and is capitalized like random variables, which is a bit confusing.
>
> A3: Thank you for pointing this out. We agree about the confusion. Since some of these notations follow the convention in the literature, we decided to add some explanations in the revision to clarify these notations, whenever there might be confusion.
>
> Q4: $B_\phi$ is not introduced on Page 5.
>
> A4: Thank you for pointing this out. In the revision, we defined $B_\phi = \max_{s} ||\phi(s)||_2$ in the end of Section 2.4. We also mentioned its definition on pages 6,7 where we use it.
>
> Q5: Can you explain what this Lemma (Lemma 1) says in words, it is not clear why this will be useful, or do you need it in your analysis.
>
> A5:  Lemma 1 shows that the distribution mismatch between the stationary distribution and the instantaneous state distribution at time $t$ decays fast with a geometric rate $\mathcal{O}(\chi^t)$. Our analyses in Propositions 1 and 3 decouple the biases of PER-ETD into the sum of distribution mismatches and initial errors. Lemma 1 is applied to yield the exponential decay rate.
>
> Q6: There is no row space above “Proposition 1”.
>
> A6: Thank you for pointing this out! We fixed this in the revision.
>
> Q7: There are a lot of constants introduced that it's easy to lose sight of what they mean and get confused if ever they were introduced at the beginning of the paper. One way to make the paper more clear would be to hide all these details and explain high-level steps of the analysis or proof.
>
> A7: Thank you for pointing this out. We have rewritten Theorems 1 and 2 in the revision, where we have hidden the constants into the big-O notation to make the results more concise. Our comments and interpretation about these theorems in the main body of the paper are all based on the order terms (which do not need the information about those constants). We then give the formal statements of these theorems with the constants in their precise forms in the appendices (See Appendices D.3 and E.3).
>
> Q8: $\xi$ should be $\xi^b$ everywhere?
>
> A8: Thanks for the question. This depends on the context. $\xi$ is a constant between $0$ and $1$, and $\xi^b$ represents its $b$-th power.

---

### Official Review · Reviewer_RfbH · 2021-11-02

**Correctness:** 3
**Technical Novelty And Significance:** 3
**Empirical Novelty And Significance:** 3
**Recommendation:** 8
**Confidence:** 3

**Main Review:**

Strengths:
- The paper is well written and clear to follow.
- Strong theoretical analysis
- The proposed tricks are simple and elegant.

Some suggestion:
- Adding an experiment to discuss the choice of behavior policy and target policy would be appreciated.
- Figure 1&2 is hard to read, perhaps you could plot the line with variance bar


**Summary Of The Paper:**

This paper proposed a new off-policy evaluation successor method of ETD. The method has reduced variance by leveraging a simple and effective way that restarting follow-on trace iteration every couple of updates. The authors also provide theoretical analysis that shows that the proposed method improves the converge rate from an exponential one to a polynomial one with the guarantee of the same fixed converging points. The empirical results show that the proposed method could converge in the case that neither TD nor ETD does.

**Summary Of The Review:**

Overall, I think the paper is solid and sound, it is marginally above the acceptance threshold. However, I would consider changing the scores if my concerns are addressed.

------------
After Rebuttal:
After reading the author's rebuttal and the comments of other reviewers, I decided to increase the score to 8. I appreciate the author's thorough and conscientious rebuttal to address all my concerns.

---

> ### Author Response · Authors · 2021-11-20
> **Authors’ response**
>
> Many thanks for providing the review!  In our revision of the paper, we added new experiments in Appendix B and made various revisions throughout the paper based on all reviewers’ comments. All our changes are highlighted with blue-colored texts. New comments on these changes are very welcome!
>
> Q1: Adding an experiment to discuss the choice of behavior policy and target policy would be appreciated.
>
> A1. Great point! We have added the suggested experiments in Appendix B. Please check the details in the revised paper. In summary, our results demonstrate the following points: (1) Under large mismatch, our PER-ETD has a significant advantage over TD and ETD. TD does not converge, and ETD experiences a substantial variance. But our PER-ETD converges fast with small variance and convergence error (with respect to the ground truth value function) as long as the period length is properly chosen. (2) Under only a slight mismatch, all TD, ETD and our PER-ETD converge. Our PER-ETD achieves a tradeoff between convergence rate, convergence error, and variance. Specifically, TD converges slowly and suffers from large convergence error, and ETD converges fast with small convergence error but suffers from large variance. Our PER-ETD achieves a tradeoff; i.e., converges faster than TD with smaller error, and attains a smaller variance than ETD. (3) Focusing on PER-ETD, larger mismatch (caused either by varying behavior or target policies) results in slower convergence rate, larger convergence error, and larger variance, which agrees with our theorem.
>
> Q2: Figure 1&2 is hard to read, perhaps you could plot the line with variance bar.
>
> A2: We have replotted our figures with variance bars in Appendix A.4 in the revision. It appears to us that for some plots (e.g., Figure 6) variance bars are more illustrative, and for some other plots (e.g., Figure 1) error bands are more illustrative. So we will provide both options to readers.

---

### Official Review · Reviewer_vCDh · 2021-11-02

**Correctness:** 4
**Technical Novelty And Significance:** 4
**Empirical Novelty And Significance:** 2
**Recommendation:** 8
**Confidence:** 3

**Main Review:**

The paper is well written and easy to follow. The analysis make the improvements of PER-ETD over ETD evident and convincing, and the empirical results, while limited, illustrate well the theoretical contributions when dealing with a significant mismatch.

I think the experiments should have contained some results in settings that aren't constructed to diverge. Specifically, I would have liked to see how PER-ETD performs compared to ETD and TD a bit more broadly. I think small illustrative domains are sufficient for this kind of work, but having more than a single domain/setting to either show that we see the same predicted effects in different settings or to give a more complete picture would have improved this work. For instance, an experiment which varies mismatch could have helped illustrate how mismatch affects PER-ETD, ETD, and TD, and possibly help visualize the two-phase behavior discussed in the paper.

That being said, I find the overall contributions sufficient for acceptance. The work is well motivated, novel, and provides a notable improvement over ETD. A more complete comparison would have been nice, but, as it stands, this is likely to be of interest to parts of RL community and is worth sharing.

Questions:
================

- Is setting the same step size really "fair"? I can understand keeping the same step-size for different $b$ for the purpose of illustrating the theoretical contributions, but why is it fair to use a fixed step size across different algorithms?
- What does the shaded area represent?

**Summary Of The Paper:**

The authors propose an improved variant of emphatic temporal difference (ETD) aimed at addressing issues of high variance when faced with a large mismatch between behavior and target policies. The main improvement of PER-ETD, the proposed algorithm, comes from periodically clearing the follow-on trace at logarthmically increasing periods. The authors present a finite-time analysis for both the PER-ETD(0) and PER-ETD($\lambda$) case showing how bias and variance depends on the time between clearing the trace. This is used to derive a schedule that effectively minimizes the variance and bias to achieve polynomial sample complexity. The authors conclude by illustrating these improved properties in Baird's counter-example MDP. The results confirm the effect on bias and variance of the period parameter and highlight's that $\lambda=1$ no longer results in the closest fixed-point to the optimal solution.

**Summary Of The Review:**

The paper is well written and easy to follow. The analysis make the improvements of PER-ETD over ETD evident and convincing, and the empirical results, while limited, illustrate well the theoretical contributions when dealing with a significant mismatch.

---

> ### Author Response · Authors · 2021-11-20
> **Authors’ response**
>
> Many thanks for providing the review! In our revision of the paper, we added new experiments in Appendix B, and made various revisions throughout the paper based on all reviewers’ comments. All our changes are highlighted with blue-colored texts. New comments on these changes are very welcome!
>
> Q1: Could the author compare PER-ETD, ETD, and TD in more domains?   For instance, an experiment that varies mismatch could have helped illustrate how it affects.
>
> A1: Great point! We have added the suggested experiments in Appendix B. Please check the details in the revised paper. In summary, our results demonstrate the following points: (1) Under large mismatch, our PER-ETD has a significant advantage over TD and ETD. TD does not converge, and ETD experiences a substantial variance. But our PER-ETD converges fast with small variance and convergence error (with respect to the ground truth value function) as long as the period length is properly chosen. (2) Under only a slight mismatch, all TD, ETD and our PER-ETD converge. Our PER-ETD achieves a tradeoff between convergence rate, convergence error, and variance. Specifically, TD converges slow and suffers from large convergence error, and ETD converges fast with small convergence error but suffers from large variance. Our PER-ETD achieves a tradeoff; i.e., converges faster than TD with smaller error, and attains a smaller variance than ETD. (3) Focusing on PER-ETD, larger mismatch (caused either by varying behavior or target policies) results in slower convergence rate, larger convergence error, and larger variance, which agrees with our theorem.
>
> Q2: Is setting the same step size really "fair"?
>
> A2: Great question! The reason that we set the stepsize to be the same for all algorithms is that we are here interested in comparing the variance and bias errors in the convergence, and the same stepsize for all algorithms can fairly compare how sensitive these errors are with respect to the stepsize. Otherwise, an algorithm can have arbitrarily small variance if the stepsize is set to be small enough. On the other hand, if we were interested in comparing the convergence rate, then we would tune the stepsize for each algorithm to achieve their corresponding best convergence rates.
>
> Q3: What does the shaded area represent?
>
> A3: Thanks for the question! In our experiments, we repeat the training of each algorithm with 20 independent initializations. The shaded area reflects the actual variations of the algorithm induced by the randomness of sampled trajectories over these experiments runs, and the solid line indicates the average over all these experiments runs.

---

### Official Review · Reviewer_TgfL · 2021-11-07

**Correctness:** 4
**Technical Novelty And Significance:** 4
**Empirical Novelty And Significance:** 2
**Recommendation:** 8
**Confidence:** 3

**Main Review:**

Pros:
1. In my opinion, this paper does answer an important question in off-policy value evaluation. The contribution is solid, and the proposed algorithm intuitively makes sense.
2. The analysis in this paper is quite novel. The trade-off between variance and bias is very interesting.
3. Theorem 1 is interesting in capturing how the sample complexity depends on the mismatch level.
4. The advantage of this work over the concurrent paper (Zhang2021) is clear.



Cons:
1. It seems that the algorithm is not using all the data in an efficient manner. It seems to me that the update for the theta parameter mostly depends on the the last data point within the last period. Is that right? Most data points in the period are only used to update F? Is there a better way to make full use of the data? Maybe some variant similar to least square type methods?
2. This paper considers a projected version of ETD.  Is it possible to remove the projection operator? In addition, there are a few papers on how to remove projection for the analysis of stochastic approximation under Markovian noise which should be mentioned in this paper.

   R. Srikant and L. Ying,  Finite-time error bounds for linear stochastic approximation and TD learning, COLT 2018.

    B. Hu and U. Syed, Characterizing the exact behaviors of temporal difference learning algorithms using Markov jump linear system theory, NeurIPS 2019.

   M. Kaledin, E. Moulines, A. Naumov, V. Tadic, H. Wai. Finite Time Analysis of Linear Two-timescale Stochastic Approximation with Markovian Noise, COLT 2020.

   C. Z. Chen, S. Zhang, T. T. Doan, S. T. Maguluri, and J.-P. Clarke, Performance of Q-Learning with Linear Function Approximation: Stability and Finite-Time Analysis.




**Summary Of The Paper:**

The variance of ETD can grow up exponentially so this paper proposes to use a periodically restarted variant (namely, the PER-ETD) as an improvement.  The authors show that the proposed method has a polynomial sample complexity, and provide some experimental validation.

**Summary Of The Review:**

Overall I am still positive on this paper. The analysis is very interesting and insightful in characterizing the trade-off between variance and bias. Currently my score is 6 but I am willing to increase my score if all my comments are addressed.


---------------------------------------------------
I think the authors' response to my comments are very reasonable. I have increased my score to 8.

---

> ### Author Response · Authors · 2021-11-20
> **Authors’ response**
>
> Many thanks for providing the review! In our revision of the paper, we added new experiments in Appendix B, and made various revisions throughout the paper based on all reviewers’ comments. All our changes are highlighted with blue-colored texts. New comments on these changes are very welcome!
>
> Q1: It seems that the algorithm is not using all the data in an efficient manner. It seems to me that the update for the theta parameter mostly depends on the last data point within the last period. Is that right? Most data points in the period are only used to update $F$? Is there a better way to make full use of the data? Maybe some variant similar to least square type methods?
>
> A1: Great comments and questions! Yes, all the data points except the last within a period are used to update $F$, and those data were not reused for updating the main variable $\theta$. The reason is that the data here are sampled as a Markovian trajectory, and samples within the same period are highly correlated. Hence, reusing data samples in one period will cause a high-correlation bias between the $F$ estimator and $\theta$ estimator, which can be proven to hurt the convergence rate. Such phenomena have already been observed in the literature (Kotsalis et al. 2020), and it has been shown that not reusing data has a better sample complexity than reusing data (which causes correlation bias).
>
> (Kotsalis et al. 2020): Kotsalis Georgios, Guanghui Lan, and Tianjiao Li. "Simple and optimal methods for stochastic variational inequalities, II: Markovian noise and policy evaluation in reinforcement learning." arXiv preprint arXiv:2011.08434 (2020).
>
> Q2: Could the author remove the projection? And there are related papers on how to remove projection for SA with Markovian noise which should be mentioned.
>
> A2: Great point! Up to our best understanding, removing projection here does not lead to a finite-time convergence guarantee. The main reason is that for our off-policy algorithm PER-ETD, the variance scales with $T$, due to the presence of the importance sampling ratio for adjusting the distribution mismatch. Thus, projection helps to keep the variance bounded. The references such as (Srikant and Ying, 2018, Hu and Syed, Kaledin et al., 2020 and Chen et al., 2020) consider on-policy TD, in which the variance is bounded without projection and the projection is not needed. This is also the reason that the techniques in these studies cannot be used in our analysis. We have also added these references in Section 1.2 on related works.

---

> > ### Comment · Reviewer_TgfL · 2021-11-24
> > **Re: Authors' response**
> >
> > Thanks for the response. The explanations make sense to me. I have increased my score accordingly.

---

> > > ### Author Response · Authors · 2021-11-24
> > > **Many thanks!**
> > >
> > > We thank the reviewer very much for the prompt and positive response!

---

### Decision · Program_Chairs · 2022-01-20

**Decision:**

Accept (Poster)

**Comment:**

This paper investigates TD-based off-policy policy evaluation. This topic is of interest as most SOTA DRL methods are built upon unsound algorithms, whereas more sound variants are difficult to use in practice and have not been widely adopted. This paper introduces a new variant of ETD that addresses the variance issue with the existing algorithm, along with theory characterizing sample efficiency. The paper includes a well done illustrative empirical study to support the theory. The reviewers all scored the paper highly.

The AC pointed out several minor issues in the presentation that the authors should address for camera ready.  In addition the grammar and word usage is rough in some places. Please take time to improve the text.